# TMEM16A in smooth muscle cells acts as a pacemaker channel in the internal anal sphincter

Ping Lu[1], Lawrence M. Lifshitz [2], Karl Bellve [2] & Ronghua ZhuGe [1]✉

Maintenance of fecal continence requires a continuous or basal tone of the internal anal sphincter (IAS). Paradoxically, the basal tone results largely from high-frequency rhythmic contractions of the IAS smooth muscle. However, the cellular and molecular mechanisms that initiate these contractions remain elusive. Here we show that the IAS contains multiple pacemakers. These pacemakers spontaneously generate propagating calcium waves that drive rhythmic contractions and establish the basal tone. These waves are myogenic and act independently of nerve, paracrine or autocrine signals. Using cell-specific gene knockout mice, we further found that TMEM16A Cl⁻ channels in smooth muscle cells (but not in the interstitial cells of Cajal) are indispensable for pacemaking, rhythmic contractions, and basal tone. Our results identify TMEM16A in smooth muscle cells as a critical pacemaker channel that enables the IAS to contract rhythmically and continuously. This study provides cellular and molecular insights into fecal continence.

[1] Department of Microbiology and Physiological Systems, University of Massachusetts Chan Medical School, Worcester, MA, USA. [2] Program in Molecular Medicine, University of Massachusetts Chan Medical School, Worcester, MA, USA. ✉email: Ronghua.zhuge@umassmed.edu

Humans typically defecate one to three times daily and do so only in socially acceptable settings. Historically, controlled defecation has transformed society by promoting better hygiene and reducing infections from pathogens. Many mammals defecate in the same place, creating communal latrines[1]. Feces containing pheromones make communal latrines vital for reproduction, defense against predators, intra- and inter-species communication in the animal kingdom[2,3].

Controlled defecation requires mechanisms to regulate the opening and closing of the anus, ensuring the maintenance of fecal continence. The internal anal sphincter (IAS), a ring-shaped smooth muscle located at the inner layer of the anus, plays a crucial role in this continence. Physiologically, the IAS contracts continuously, producing a basal tone that contributes to over 70% to the resting anal pressure necessary for fecal continence[4–6]. Weakened IAS can lead to fecal incontinence, which affects the physical well-being and mental health of 2–24% of the U.S. population[7,8]. Conversely, excessively contracted IAS has been associated with conditions such as recurrent anal fissure[7,9–13]. Nitroglycerin, which reduces sphincter tone and resting anal pressure, is the only FDA-approved drug for chronic anal fissure. To date, there is no FDA-approved drug for fecal incontinence, highlighting the need for a deeper understanding of the cellular and molecular mechanisms underlying basal tone in IAS.

It has long been recognized that IAS basal tone is associated with rhythmic contractions[14–16]. A prevailing hypothesis is that, akin to other GI regions, IAS rhythmicity relies on interstitial cells of Cajal (ICC) with TMEM16A $Ca^{2+}$-activated $Cl^-$ ($Cl_{Ca}$) channels acting as pacemaker channels. According to this hypothesis, the IAS-ICC periodically releases $Ca^{2+}$ from the endoplasmic reticulum, thereby activating TMEM16A. This activation leads to the generation of electrical slow waves that propagate through gap junctions to smooth muscle cells (SMCs), resulting in the activation of L-type voltage-gated $Ca^{2+}$ channels and subsequent contraction[17]. The basis for this hypothesis is based on several observations: (1) $Cl^-$ channel blockers or L-type $Ca^{2+}$ channels blockers inhibit slow waves and tone recorded in IAS tissue[18]; (2) TMEM16A mRNA level is higher in isolated ICC than in other IAS cells[17], and (3) spontaneous $Ca^{2+}$ oscillations in ICC are inhibited by a TMEM16A blocker[19]. Yet, there is no report of TMEM16A $Cl_{Ca}$ currents in the IAS-ICC, nor direct proof of a coupling between TMEM16A channels in ICC and L-type $Ca^{2+}$ channels in SMCs. Significantly, the specific TMEM16A deletion in ICC or even the lack of ICC in the IAS showed no effect on IAS basal tone[20–23], underscoring the need for further studies to fully understand the role of ICC in IAS rhythmicity.

Another important question about IAS rhythmic contractions is their spatial origin. The exact origin—whether from a single pacemaker, multiple pacemakers, or an undirected distribution—remains unclear. This uncertainty arises due to the limitations of the commonly used isometric force measurements of IAS strips, which do not provide spatial information. Similarly, electrical recordings of slow waves suffer from limited spatial resolution. A recent imaging study reveals the spatial variations of spontaneous $Ca^{2+}$ transients within IAS-SMCs under isometric conditions[24]. Interestingly, these $Ca^{2+}$ events predominantly originate from the IAS distal end and can propagate longitudinally from the anal to the oral direction. However, the initiation sites and circumferential propagation mechanisms remain unresolved. These issues are critical because $Ca^{2+}$ signals and their circumferential propagation are expected to generate IAS basal tone and high anal canal pressure. Given the spatial limitations of previous methods, innovative approaches are urgently needed to elucidate this aspect.

One technique we have introduced is the preparation of precision-cut mouse IAS slices, which offer distinct advantages[20]. These cross-sectional slices preserve the cell arrangements in situ,

mimicking the natural contractions and relaxations of the IAS in vivo. With a thickness of approximately 250 μm, these IAS slices provide an ideal platform for studying function and dynamics at the single cell or tissue level using light fluorescence microscopy. The use of mice as a tissue source facilitates the exploration of the molecular mechanisms governing IAS contractility through genetic approaches. Here, we used the precision-cut IAS slices to address the following questions: (1) whether the IAS contains pacemakers, (2) whether $Ca^{2+}$ signaling controls basal tone at the IAS tissue level, and (3) whether TMEM16A in SMCs or ICC is critical for IAS pacemaking. Rather than relying on ambiguous TMEM16A pharmacology[25–27], we utilized cell-specific TMEM16A deletion to define its role in IAS rhythmicity. We found that each IAS slice houses approximately three pacemakers that orchestrate the generation of $Ca^{2+}$ waves responsible for basal tone. These waves are intrinsically myogenic, acting independently of neural, paracrine or autocrine signals. Importantly, our discovery underscores the critical role of TMEM16A $Cl_{Ca}$ channels in SMCs—not ICC—for pacemaking and maintenance of basal tone. This study identifies TMEM16A in SMCs as the primary pacemaker channel, driving rhythmic contractions of the IAS, providing insight into fecal continence.

## Results

**Spontaneous intercellular calcium waves in IAS-SMCs generate rhythmic contractions, resulting in IAS basal tone**. To directly establish the relationship between $[Ca^{2+}]_i$ and contractions at the IAS tissue level, we imaged $Ca^{2+}$ indicator Cal-520-loaded IAS slices at low magnification. Like contractions in the IAS slices without $Ca^{2+}$ indicators[20], those loaded with the indicators developed rhythmic contractions in 1.3 mM extracellular calcium ($[Ca^{2+}]_e$) solution (Fig. 1a). With $Ca^{2+}$ indicators present, these slices also spontaneously produced calcium oscillations (Fig. 1a). Removing extracellular $Ca^{2+}$ eliminated both $Ca^{2+}$ oscillations and rhythmic contractions, causing the IAS lumen to gradually expand until it reached a steady state (Figs. 1a–c). These findings suggest that, under physiological $[Ca^{2+}]_e$, IAS slices generate $Ca^{2+}$ oscillations and rhythmic contractions, thereby maintaining a basal tone.

We further observed that $Ca^{2+}$ oscillations propagated from one region to another within the smooth muscle (Fig. 1d). We have termed these as intercellular $Ca^{2+}$ waves. $Ca^{2+}$ waves can be categorized into two types. Type 1 $Ca^{2+}$ waves consisted of peaks with roughly equal amplitude and duration (Fig. 1d). These $Ca^{2+}$ waves were temporally associated with contractions in a one-to-one relationship. They preceded the contractions by $0.50 \pm 0.04$ s (Fig. 1d, inset and Fig. 1f) and returned to baseline faster than the contractions (Fig. 1d inset). Type 2 $Ca^{2+}$ waves displayed a complex waveform, often composed of peaks with variable duration and amplitude. However, they also preceded contractions, typically by $0.54 \pm 0.02$ s (Fig. 1e, inset and Fig. 1f). Type 2 $Ca^{2+}$ waves seemed to arise from the fusion of sequential $Ca^{2+}$ events, with one event being dominant. As depicted in Fig. 1e (inset), immediately after the first peak neared the basal level (image 4), a subsequent event began and coincided in time with a third event before its full decay (images 5 and 6). Consequently, the associated contractions also displayed an irregular shape. For instance, before the contraction induced by the first event had fully recovered, subsequent smaller events triggered new contractions (Fig. 1e).

Given that $Ca^{2+}$ waves continuously propagate across IAS slices and considering that SMCs represent a predominant cell type within the IAS, our analysis suggests that $Ca^{2+}$ waves occur in IAS-SMCs. These waves induce rhythmic contractions, resulting in the basal tone in IAS slices. This conclusion is consistent with our previous findings, which showed that synchronized $Ca^{2+}$ oscillations, as

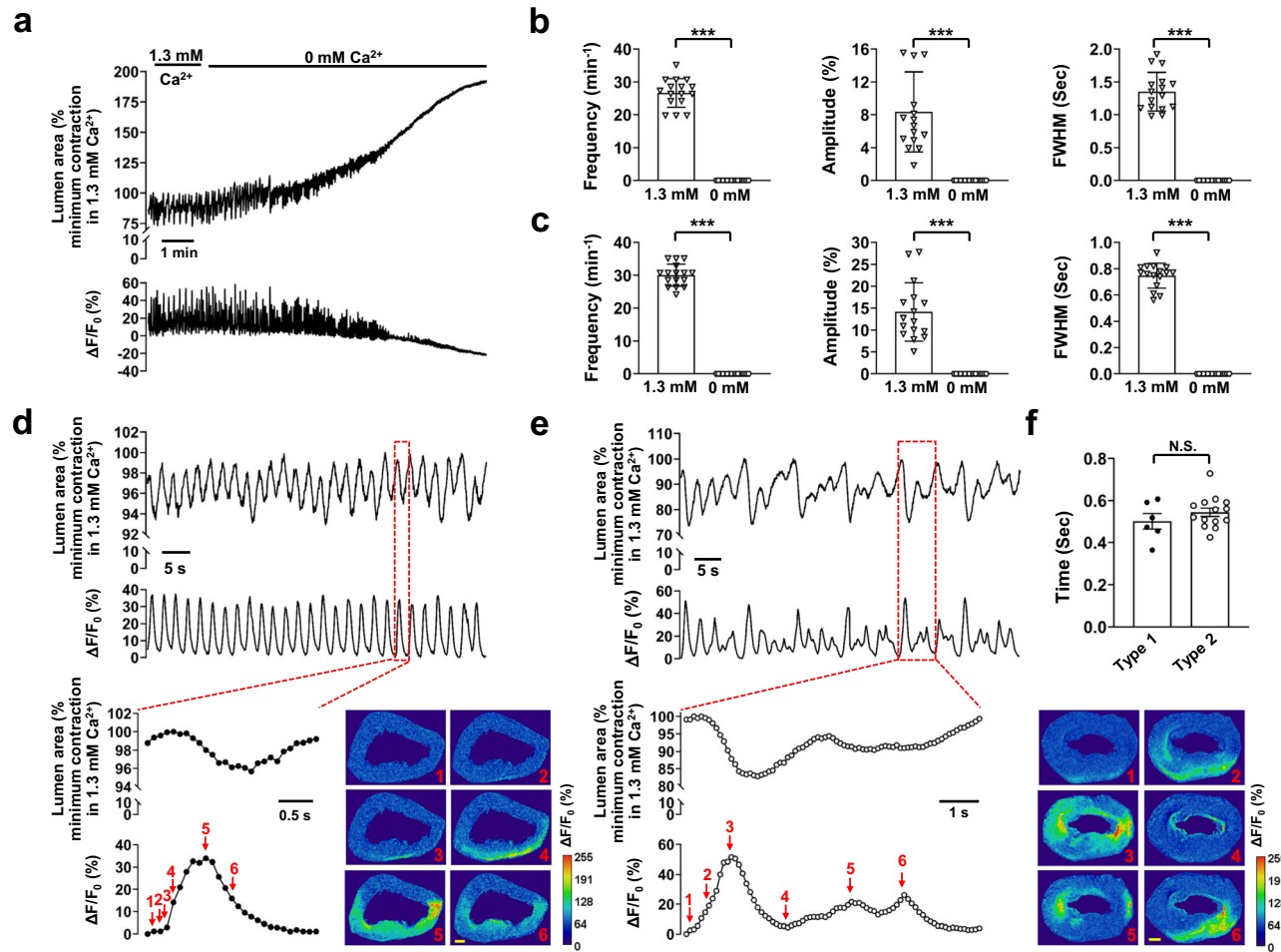

**Fig. 1 Intercellular Ca²⁺ waves induce rhythmic contractions, leading to basal tone, in IAS slices. a** $Ca^{2+}$ signals and associated contractions in the presence and absence of extracellular $Ca^{2+}$. The summed area values were normalized to the maximal relaxation or minimal contraction during rhythmic contractions in the presence of 1.3 mM extracellular $Ca^{2+}$ in this panel and throughout the other figures. **b** Summary of the effects of extracellular $Ca^{2+}$ removal on contractions. **c** Summary of the effects of extracellular $Ca^{2+}$ removal on $Ca^{2+}$ signals. FWHM: Full Width at Half Maximum. Data from the last 30 s in the 0 mM $Ca^{2+}$ medium were used to calculate the zero-calcium treatment. $n = 16$. **d** Type 1 intercellular calcium waves with regular amplitude and duration (lower trace) were temporally associated with rhythmic contractions (upper trace) in a one-to-one manner. Inset: a single event, indicated by the red dotted line box, is shown on an expanded scale. Images show changes in lumen area and $Ca^{2+}$ fluorescence ($\Delta F/F_0(\%)$); numbers in the images correspond to those marked near the $\Delta F/F_0(\%)$ trace, scale bar = 200 μm. **e** Type 2 intercellular calcium waves (lower trace) with irregular amplitude and duration were associated with rhythmic contractions (upper trace). Inset: a single event, indicated by the red dotted line box, is shown on an expanded scale. The images display temporal changes in the lumen area and $Ca^{2+}$ fluorescence ($\Delta F/F_0(\%)$); numbers in the images correspond to those marked near the $\Delta F/F_0(\%)$ trace, scale bar = 200 μm. **f** Delay of contraction onset relative to $Ca^{2+}$ wave onset. $n = 6$ for type 1, $n = 14$ for type 2; N.S. no significant difference, ***$p < 0.001$, as determined by paired two-tailed (**b**, **c**) or unpaired two-tailed (**f**) Student's $t$ test.

detected by genetically targeted R-CaMP1.07 in SMCs, lead to the shortening of SMCs at the single-cell level[20].

**Each IAS contains several fixed pacemaker sites that generate spontaneous intercellular Ca²⁺ waves.** It remains unknown whether $Ca^{2+}$ signals arise sporadically across the space or predominantly at certain fixed locations within the IAS. To address this, we mapped the initiation sites of $Ca^{2+}$ signals in IAS slices. We found that intercellular $Ca^{2+}$ waves consistently originated from several fixed points. These initiation points were mainly situated near the inner and outer edges or the distal end of the anus. From these origins, waves propagated either parallel or perpendicular to the circular SMCs. Owing to their consistent occurrence, we refer to these $Ca^{2+}$ active sites as "pacemakers." Figure 2 illustrates one such recording from an IAS slice. Within this slice, we identified three pacemakers. Pacemaker 1 was located around the 3 o'clock position along the outer edge

(Fig. 2a, d). Pacemakers 2 and 3 were approximately at the 7 o'clock position on the outer edge and around 10 o'clock near the inner edge, respectively (Fig. 2b–d). The $Ca^{2+}$ wave from pacemaker 1 spanned almost the entire ring of the slice (Fig. 2a). Frequently, $Ca^{2+}$ waves from distinct origins intersected and converged (Supplementary Movie 1). For instance, a wave from pacemaker 1 passed through pacemaker 2 (Fig. 2a), and the opposite was also observed (Fig. 2b). Occasionally, pacemaker 2 launched a $Ca^{2+}$ wave during an ongoing wave from pacemaker 3 (Fig. 2c). Such interactions between pacemakers could result in highly variable $Ca^{2+}$ signal amplitudes at the pacemaker sites. By plotting $Ca^{2+}$ waves from all detectable pacemakers over a duration of 2.5 min, we constructed a pacemaker distribution map for this IAS slice (Fig. 2d) and another IAS slice depicted in Fig. 1d (Supplementary Fig. 2). Notably, the region covered by these pacemakers encompassed only $0.69 \pm 0.13\%$ of the projected 2d area of the IAS slice. This suggests that pacemakers are distinctly localized functional domains within the IAS.

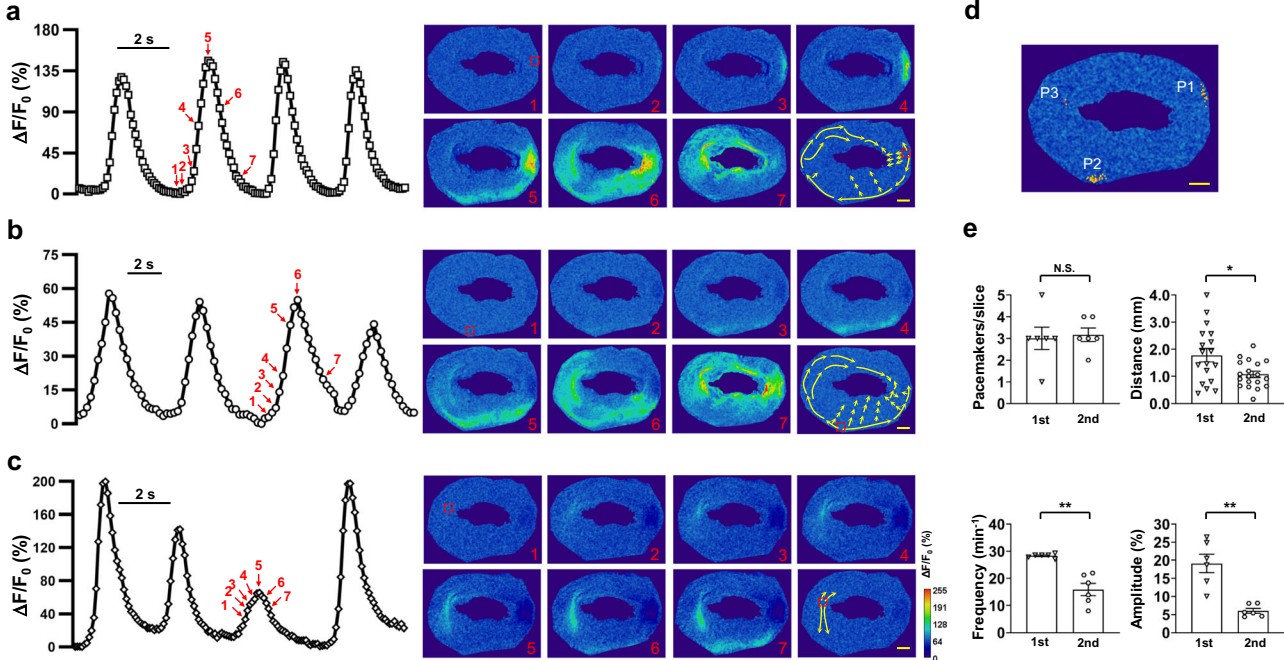

**Fig. 2 The IAS contains multiple pacemakers. a–c** Time courses of $Ca^{2+}$ oscillations from three pacemakers in an IAS slice. These time courses represent changes in $Ca^{2+}$ fluorescence from an area marked with a red square dotted box ($10 \times 10$ pixels), as shown in Image 1 on the right. Images illustrate the spatiotemporal evolution of $Ca^{2+}$ waves, and the numbers on the images correspond to those marked near the $\Delta F/F_0$ traces on the left. The arrows in the last image, which are not numbered, indicate the primary path of each $Ca^{2+}$ wave from pacemakers. Scale bar = 200 μm. **d** $Ca^{2+}$ pacemaker map of the IAS slice as shown in (**a–c**) is based on a 2.5-min recording. P1, P2, P3 represent pacemaker1, pacemaker2, and pacemaker3, respectively. Red indicates 2 repeats; yellow indicates 1 event. Scale bar = 200 μm. **e** Comparison of pacemaker parameters between the first and second slices from the anal verge ($n = 6$). There is no difference in the number of pacemakers between the two slices, while $Ca^{2+}$ wave propagation distance ($n = 6$; 18 and 19 events from the first and second slices, respectively), frequency, and amplitude are significantly different. N.S., no statistical significance, \*$p < 0.05$, \*\*$p < 0.01$, as determined by paired two-tailed Student's $t$ test.

To assess if pacemaker characteristics differ based on their IAS location, we analyzed pacemakers from the first and second slices near the anal verge (Fig. 2e). Both slices had similar pacemaker counts ($3.0 \pm 0.52$ vs. $3.16 \pm 0.31$; $p = 0.79$). Yet, the $Ca^{2+}$ travel distance was longer in the first slice ($1.78 \pm 0.25$ mm vs. $1.07 \pm 0.11$ mm, $p < 0.05$). Additionally, the first slice had greater $Ca^{2+}$ wave frequency ($28.31 \pm 0.31$ vs. $15.88 \pm 2.26$, $p < 0.01$) and amplitude ($19.11 \pm 2.55\%$ vs. $6.03 \pm 0.68\%$, $p < 0.01$). Thus, we exclusively used the first slice for subsequent imaging experiments.

**Pacemaking in the IAS is independent of nerve, and paracrine or autocrine signals.** While intercellular $Ca^{2+}$ waves traverse IAS-SMCs, the initiating signal may originate from non-SMCs in the IAS. We tested the role of tetrodotoxin, a $Na^+$ channel inhibitor, on these waves. Figure 3a shows 1 μM tetrodotoxin didn't alter the calcium wave properties in IAS slices with 1.3 mM $[Ca^{2+}]_e$. This indicates IAS neurons' action potential isn't crucial for the waves, especially as mouse gastrointestinal SMCs don't have voltage-gated $Na^+$ channels[28]. We then explored if IAS-produced substances affected $Ca^{2+}$ wave generation. If they were crucial, wave directions would change when solution flow reversed. However, Fig. 3b show flow reversal didn't impact pacemaker actions or wave spread, suggesting such substances likely don't initiate IAS's $Ca^{2+}$ waves.

**Calcium pacemaking may require $Cl_{Ca}$ channels in the IAS.** The TMEM16A $Cl_{Ca}$ channel serves as a pacemaker channel in parts of the GI tract[29,30]. TMEM16A inhibitors impact $Ca^{2+}$ transients in ICC, slow waves, and basal tone in the IAS[17,19,31]. Therefore, we investigated the effects of these inhibitors on pacemaker activity and $Ca^{2+}$ waves in IAS slices. Niflumic acid, a broad-spectrum $Cl_{Ca}$

channel inhibitor, abolished $Ca^{2+}$ waves in IAS slices (Fig. 4a). In contrast, T16Ainh-A01, a TMEM16A inhibitor at 10 μM mildly reduced them (Fig. 4b), showing a slight drop in frequency and amplitude, with stable FWHM.

Two additional TMEM16A inhibitors, CaCCinh-A01 and Ani9, have region-specific effects in the GI[32,33]. Both, at 10 μM, block $Cl_{Ca}$ channels[34,35]. Yet, they only reduced, not halted, $Ca^{2+}$ waves (Fig. 4c, d). With CaCCinh-A01, there was a decrease in frequency and amplitude, and a minor FWHM increase. Ani9 showed comparable outcomes.

These findings suggest that $Cl_{Ca}$ channel inhibitors could affect $Ca^{2+}$ waves in IAS slices. Given the mixed inhibitor outcomes and potential non-specific effects[25–27], a targeted gene knockout method is essential to determine the TMEM16A's roles in specific IAS cell types.

**Unique distribution pattern of TMEM16A in the IAS.** TMEM16A is expressed in ICC and SMCs in the IAS[17,31]. To better visualize its distribution in the IAS, we performed co-immunostaining of TMEM16A with c-Kit, a commonly used ICC marker, or Myh11, an SMC marker. As shown in Fig. 5a, b, TMEM16A was distributed as puncta largely in the center of the IAS and as bands at the edges and distal end of the IAS. Interestingly, TMEM16A puncta mostly colocalized with c-Kit immunostaining signals, and TMEM16A bands colocalized with Myh11 signals.

Considering the heterogeneity of TMEM16A distribution, we further analyzed its spatial pattern by dividing the IAS into three segments of equal length: proximal, middle, and distal (total length: $562.50 \pm 0.96$ μm) (Supplementary Fig. 3). Figure 5c shows that $28.76 \pm 2.70\%$ and $24.44 \pm 4.29\%$ of TMEM16A signals were located in ICC in the proximal and middle segments, respectively.

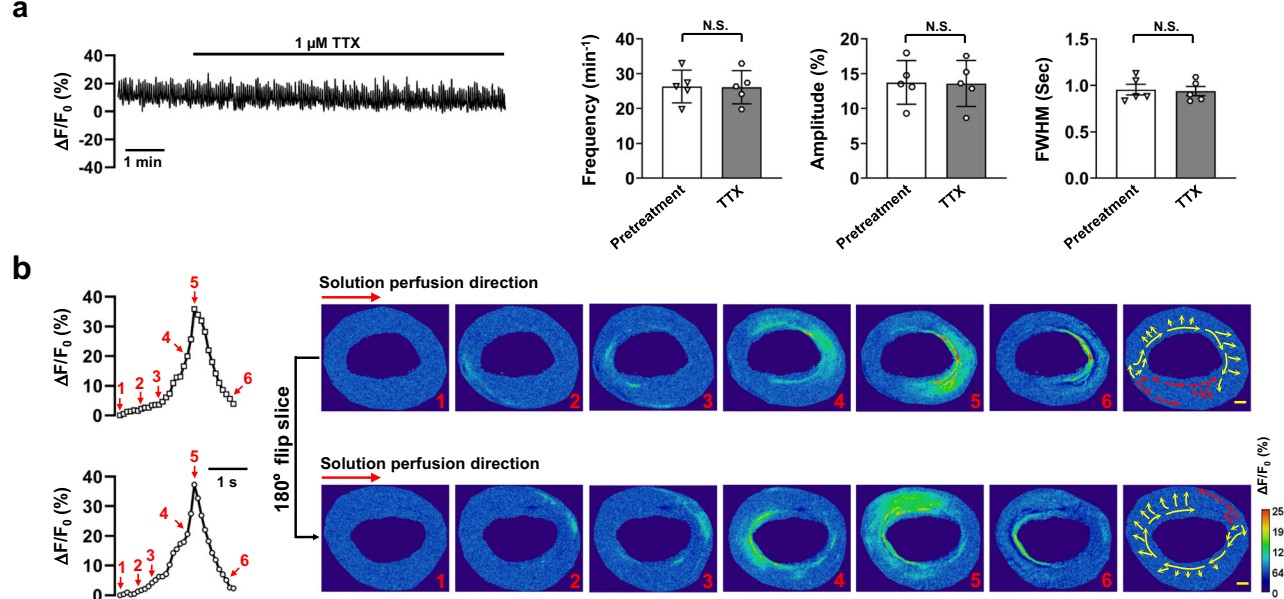

**Fig. 3 Pacemaking in the IAS is independent of nerve and paracrine/autocrine signals. a** Tetrodotoxin (TTX) had no effect on $Ca^{2+}$ waves in IAS slices. The trace displays a representative recording of the effect of $1\,\mu M$ TTX on $Ca^{2+}$ waves in the presence of 1.3 mM extracellular $Ca^{2+}$. Summarized effects on $Ca^{2+}$ wave frequency, amplitude, and FWHM are presented in the bar charts ($n = 5$). N.S., no statistical significance as determined by paired two-tailed Student's $t$ test. **b** The orientation of perfusion flow did not affect $Ca^{2+}$ waves and their propagation. Illustrated are a $Ca^{2+}$ wave's time course, represented by lines with open squares or circles, the spatiotemporal evolution, as shown in images 1–6, and its main propagation paths, depicted in the image without a number. The upper row demonstrates these aspects before a 180° horizontal rotation of the recording chamber, and the lower row displays them after the rotation ($n = 6$). Scale bar $= 200\,\mu m$.

This value decreased to $5.40 \pm 1.44\%$ in the distal segment. Conversely, $47.88 \pm 3.68\%$ and $53.90 \pm 3.84\%$ of TMEM16A signals were in SMCs in the proximal and middle segments, respectively, and this value increased to $84.32 \pm 4.77\%$ in the distal segment (Fig. 5d).

To further confirm the identity of TMEM16A-expressing cells, we genetically deleted TMEM16A in ICC (TMEM16A[ICCKO]) or in SMCs (TMEM16A[SMKO]) by crossing TMEM16A[flox/flox] mice with Kit[CreERT2] mice[20,36] or with SMA[Cre] mice[20,31], respectively. Strikingly, in the TMEM16A[ICCKO] IAS, only the TMEM16A puncta in the IAS disappeared, while TMEM16A bands at the edge and distal end remained (Fig. 5a). In contrast, in the TMEM16A[SMKO] IAS, the opposite pattern was observed: TMEM16A puncta remained, while TMEM16A bands at the edges and distal end were absent (Fig. 5b).

To quantify the efficiency of the deletion, we compared changes in TMEM16A expression in both knockout mice and their isogenic controls. In isogenic controls for TMEM16A[ICCKO] mice, the majority of c-Kit-expressing pixels contained TMEM16A signals in all three segments: $85.12 \pm 1.63\%$ for the proximal segment, $86.72 \pm 2.62\%$ for the middle segment, and $91.96 \pm 1.32\%$ for the distal segment (Fig. 5e). This percentage in TMEM16A[ICCKO] mice was dramatically reduced to about 5%: proximal at $5.54 \pm 1.46\%$, middle at $5.32 \pm 0.71\%$, and distal at $4.22 \pm 0.67\%$ (Fig. 5e). For the isogenic controls of TMEM16A[SMKO] mice, TMEM16A signals were found in more than 20% of Myh11-expressing pixels: $20.86 \pm 2.23\%$ in the proximal segment, $23.46 \pm 2.98\%$ in the middle segment, and $50.74 \pm 1.87\%$ in the distal segment (Fig. 5f). In TMEM16A[SMKO] mice, this percentage was reduced to nearly zero: $0.20 \pm 0.08\%$ in the proximal, $0.16 \pm 0.02\%$ in the middle, and $0.20 \pm 0.08\%$ in the distal segment (Fig. 5f).

These data clearly demonstrate that TMEM16A is expressed in both ICC and SMCs in the IAS in a spatially dependent manner. They further show that Kit[CreERT2] and SMA[Cre] can specifically and effectively delete TMEM16A in ICCs and SMCs, respectively.

**TMEM16A in ICCs is not required for pacemaking in the IAS.** Having established the spatial distribution pattern and knockout efficiency of TMEM16A in both ICC and SMCs, we next investigated their functional significance in each cell type. We found no significant differences in $Ca^{2+}$ waves in IAS slices between isogenic control mice and TMEM16A[ICCKO] mice under physiological $[Ca^{2+}]_e$ (Fig. 6a–e). For $Ca^{2+}$ waves, frequency was $30.73 \pm 0.93$ in controls versus $29.90 \pm 1.60$ in TMEM16A[ICCKO] mice ($p = 0.92$); amplitude was $14.65 \pm 2.60\%$ in controls versus $14.38 \pm 2.72\%$ in TMEM16A[ICCKO] mice ($p = 0.94$); and FWHM was $0.75 \pm 0.04\,s$ in controls versus $0.79 \pm 0.04\,s$ in TMEM16A[ICCKO] mice ($p = 0.41$).

We observed no difference in rhythmic contractions between TMEM16A[ICCKO] mice and their controls (Fig. 6a, b, g–i). The frequency of these contractions was $27.01 \pm 1.29$ events $min^{-1}$ in controls versus $28.26 \pm 1.05$ events $min^{-1}$ in TMEM16A[ICCKO] mice ($p = 0.46$); their amplitude was $7.02 \pm 1.33\%$ in controls versus $7.42 \pm 1.17\%$ in TMEM16A[ICCKO] ($p = 0.85$); and FWHM was $1.34 \pm 0.09\,s$ in controls versus $1.33 \pm 0.06\,s$ in TMEM16A[ICCKO] mice ($p = 0.93$).

After the removal of extracellular $Ca^{2+}$, we examined the effect of TMEM16A deletion in ICC on resting $[Ca^{2+}]_i$ and basal tone. A perfusion of zero $[Ca^{2+}]_e$ for ~4 min abolished $Ca^{2+}$ waves and rhythmic contractions in IAS slices from controls and TMEM16A[ICCKO] mice (Supplementary Fig. 4). Both groups displayed similar resting $[Ca^{2+}]_i$ ($24.27 \pm 4.54\%$ in controls vs. $23.84 \pm 4.62\%$ in TMEM16A[ICCKO] mice, $p = 0.95$) (Fig. 6f) and basal tone ($94.54 \pm 5.98\%$ in controls vs. $93.38 \pm 6.11\%$ in TMEM16A[ICCKO] mice, $p = 0.89$) (Fig. 6j).

We further quantified pacemaker numbers and $Ca^{2+}$ wave propagations in slices from both controls and TMEM16A[ICCKO] mice. We observed similar numbers of pacemaker sites ($2.88 \pm 0.48$ in controls vs. $2.75 \pm 0.45$ in TMEM16A[ICCKO] mice, $p = 0.85$) (Fig. 6k and Supplementary Fig. 5); propagation distances ($1.92 \pm 0.29$ mm in controls vs. $2.05 \pm 0.24$ mm in

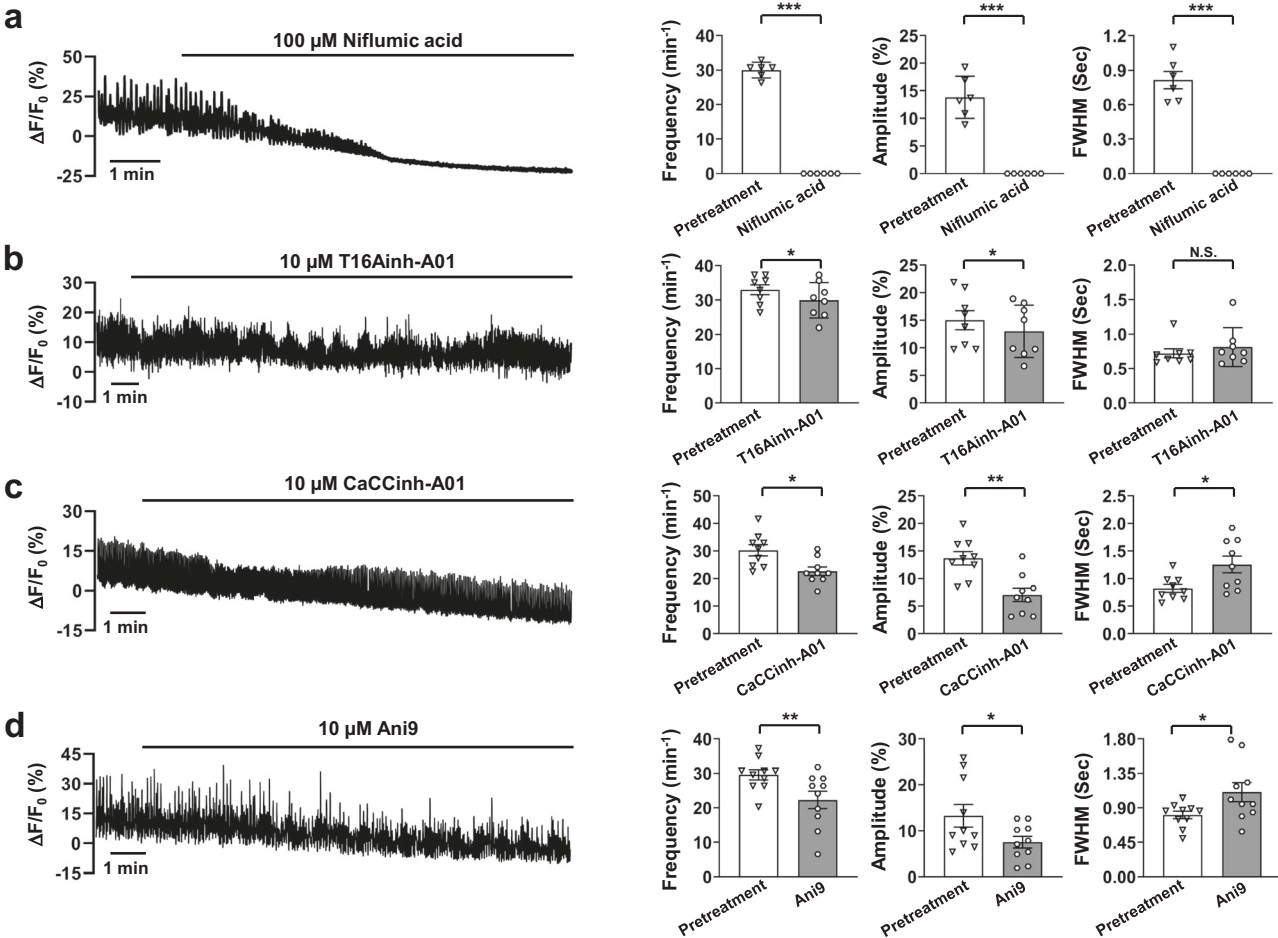

**Fig. 4 Ca²⁺ waves from pacemakers may require Cl$_{Ca}$ channels. a** Niflumic acid abolished Ca²⁺ waves in IAS slices. A representative recording of Ca²⁺ waves in 1.3 mM extracellular Ca²⁺ after niflumic acid treatment is shown, along with summarized results on Ca²⁺ wave frequency, amplitude, and FWHM ($n = 5$). **b** T16Ainh-A01 weakly inhibited Ca²⁺ waves in IAS slices. A representative recording of Ca²⁺ waves after treatment with T16Ainh-A01 is shown, along with summarized results of Ca²⁺ wave frequency, amplitude and FWHM ($n = 8$). **c** CaCCinh-A01 inhibited Ca²⁺ waves in IAS slices. A representative recording of Ca²⁺ waves after treatment with CaCCinh-A01 is shown, along with summarized results of Ca²⁺ wave frequency, amplitude and FWHM ($n = 9$). **d** Ani9 inhibited Ca²⁺ waves in IAS slices. A representative recording of Ca²⁺ waves after treatment with Ani9 is shown, along with summarized results of Ca²⁺ wave frequency, amplitude and FWHM ($n = 10$). N.S. indicates no statistical significance, *$p < 0.05$, **$p < 0.01$, ***$p < 0.001$, as determined by paired two-tailed Student's $t$ test.

TMEM16A^ICCKO mice, $p = 0.73$) (Fig. 6l); and propagation speeds (1.11 ± 0.10 mm s⁻¹ in controls vs 1.11 ± 0.11 in TMEM16A^ICCKO mice, $p = 0.996$).

Under isometric conditions, both controls and TMEM16A^ICCKO mice developed the same magnitude of basal tone in a time-dependent manner (Fig. 6m). The basal tone was 2.87 ± 0.21 mN in control mice at a frequency of 50.9 ± 1.82 events min⁻¹, compared to 2.88 ± 0.20 mN at 50.7 ± 2.19 events min⁻¹ in TMEM16A^ICCKO mice (Fig. 6n, o) ($p = 0.95$ for basal tone and $p = 0.94$ for frequency). Taken together, our findings indicate that TMEM16A in ICC is not required for pacemaking, Ca²⁺ signaling, rhythmic contractions, or basal tone in the IAS.

**TMEM16A in SMCs is required for pacemaking in the IAS.** To study the role of TMEM16A in SMCs, we first compared Ca²⁺ waves and associated rhythmic contractions in TMEM16A^SMKO mice and their isogenic controls (Fig. 7a, b). The SMC-specific TMEM16A deletion led to the total cessation of Ca²⁺ waves: frequency dropped from 29.36 ± 1.43 events min⁻¹ in controls to 0 in TMEM16A^SMKO mice, amplitude decreased from 16.21 ± 2.49% to 0, and FWHM reduced from 0.74 ± 0.03 s to 0 (Fig. 7c–e). Similarly, rhythmic contractions were abolished:

frequency was 26.34 ± 1.85 events min⁻¹ in controls and 0 in TMEM16A^SMKO, amplitude changed from 9.69 ± 2.02% to 0, and FWHM from 1.36 ± 0.13 s to 0 (Fig. 7g–i).

We also examined the effect of TMEM16A in SMCs on resting [Ca²⁺]$_i$ and basal tone by removing extracellular Ca²⁺. In IAS slices from isogenic controls, the removal of extracellular Ca²⁺ abolished Ca²⁺ waves (Fig. 7a and Supplementary Fig. 6). In contrast, in IAS slices from TMEM16A^SMKO mice, extracellular Ca²⁺ removal led to significantly less change in resting [Ca²⁺]$_i$ (26.71 ± 2.94% in controls vs. 17.31 ± 2.34% in TMEM16A^SMKO mice, $p < 0.05$; Fig. 7f) as well as in basal tone (86.18 ± 5.69% in controls vs. 18.01 ± 1.67% in TMEM16A^SMKO mice, $p < 0.001$; Fig. 7j). Intriguingly, IAS slices from TMEM16A^SMKO mice responded to zero [Ca²⁺]$_e$ with a transient increase in [Ca²⁺]$_i$ and contraction, suggesting that the slices were functional. To further assess the contractile ability of these IAS slices, we exposed them to high concentrations of KCl, which directly depolarizes the membrane. Supplementary Fig. 7 shows that 60 mM KCl led to a similar increase in calcium and contraction levels in both TMEM16A^SMKO mice and their isogenic controls (contraction: 41.30 ± 4.31% in the KO vs 39.24 ± 4.25% in the control, $p = 0.74$; calcium: 86.12 ± 8.55% in the KO vs

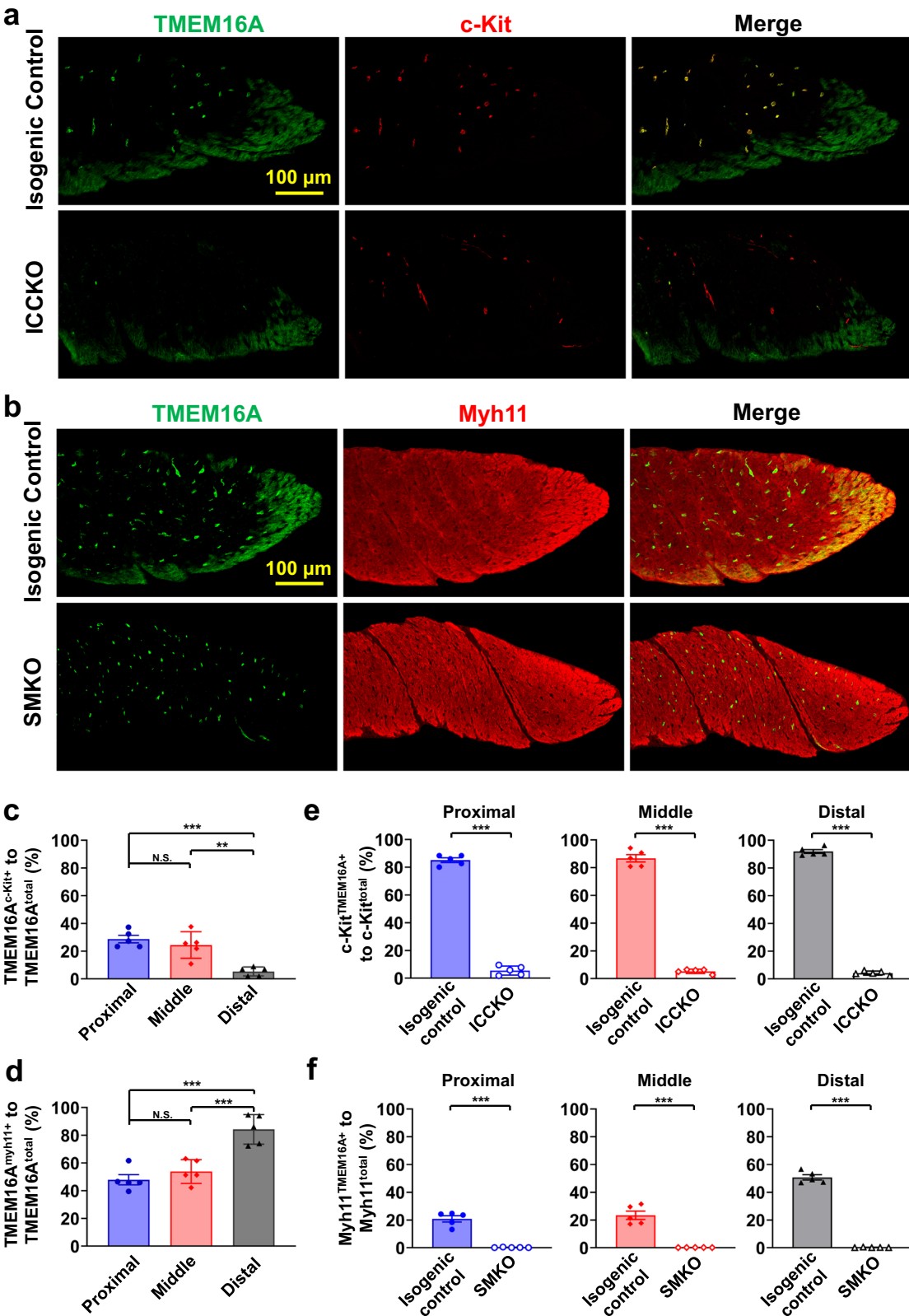

90.42 ± 10.25% in the control, $p = 0.76$), indicating that the deletion of TMEM16A in SMCs did not impair the contractile ability of the IAS.

Finally, we measured the number of pacemakers and $Ca^{2+}$ wave propagations in IAS slices from controls and TMEM16A$^{SMKO}$ mice. We detected 2.57 ± 0.37 pacemakers per slice with a propagation distance of 1.81 ± 0.21 mm at a speed of 1.17 ± 0.12 mm s$^{-1}$ in

controls, and zero pacemakers in TMEM16A$^{SMKO}$ mice (Fig. 7k, l). In summary, these results, in conjunction with our previous findings from isometric measurements of IAS strip tone from the same type of controls and TMEM16A$^{SMKO}$ mice[31], indicate TMEM16A in SMCs functions as a pacemaker channel essential for pacemaking, $Ca^{2+}$ waves, rhythmic contractions, and basal tone in the IAS.

**Fig. 5 Spatial pattern of TMEM16A expression in the IAS. a** Upper row: Immunostaining of TMEM16A (left) and c-kit protein (middle), and their colocalization (right) in a control IAS. Lower row: The same immunostaining in a TMEM16A$^{ICCKO}$ IAS. Maximum intensity projections are shown from a 20-plane 3D stacked image in both a and b below. Note that TMEM16A immunostaining signals are absent in c-kit-positive interstitial cells of Cajal (ICC) in the TMEM16A$^{ICCKO}$ IAS. **b** Upper row: Immunostaining of TMEM16A (left) and myh11 protein (middle), and their colocalization (right) in a control IAS. Lower row: The same immunostaining in a TMEM16A$^{SMKO}$ IAS. Note that TMEM16A immunostaining signals are absent in Myh11-expressing smooth muscle cells (SMCs) in the TMEM16A$^{SMKO}$ IAS. **c** Zonal quantification of TMEM16A-containing pixels in ICC relative to total TMEM16A-containing pixels using isogenic controls for TMEM16A$^{ICCKO}$ mice ($n = 5$). Zones are defined as one-third segments of the IAS; see Supplementary Fig. 3 for illustration. N.S. indicates no statistical significance, **$p < 0.01$, ***$p < 0.001$, as determined by one-way ANOVA analysis with Tukey's multiple comparisons test. **d** Zonal quantification of TMEM16A-containing pixels in SMCs relative to total TMEM16A- containing pixels using isogenic controls for TMEM16A$^{SMKO}$ mice ($n = 5$). N.S. indicates no statistical significance, ***$p < 0.001$, as determined by one-way ANOVA analysis with Tukey's multiple comparisons test. **e** Efficiency of TMEM16A deletion in ICCs in three different zones: proximal, middle, and distal regions of the IAS in TMEM16A$^{ICCKO}$ mice ($n = 5$). ***$p < 0.001$, as determined by unpaired two-tailed Student's $t$ test. **f** Efficiency of TMEM16A deletion in SMCs in three different zones: proximal, middle, and distal regions of the IAS in TMEM16A$^{SMKO}$ mice ($n = 5$). ***$p < 0.001$, as determined by unpaired two-tailed Student's $t$ test.

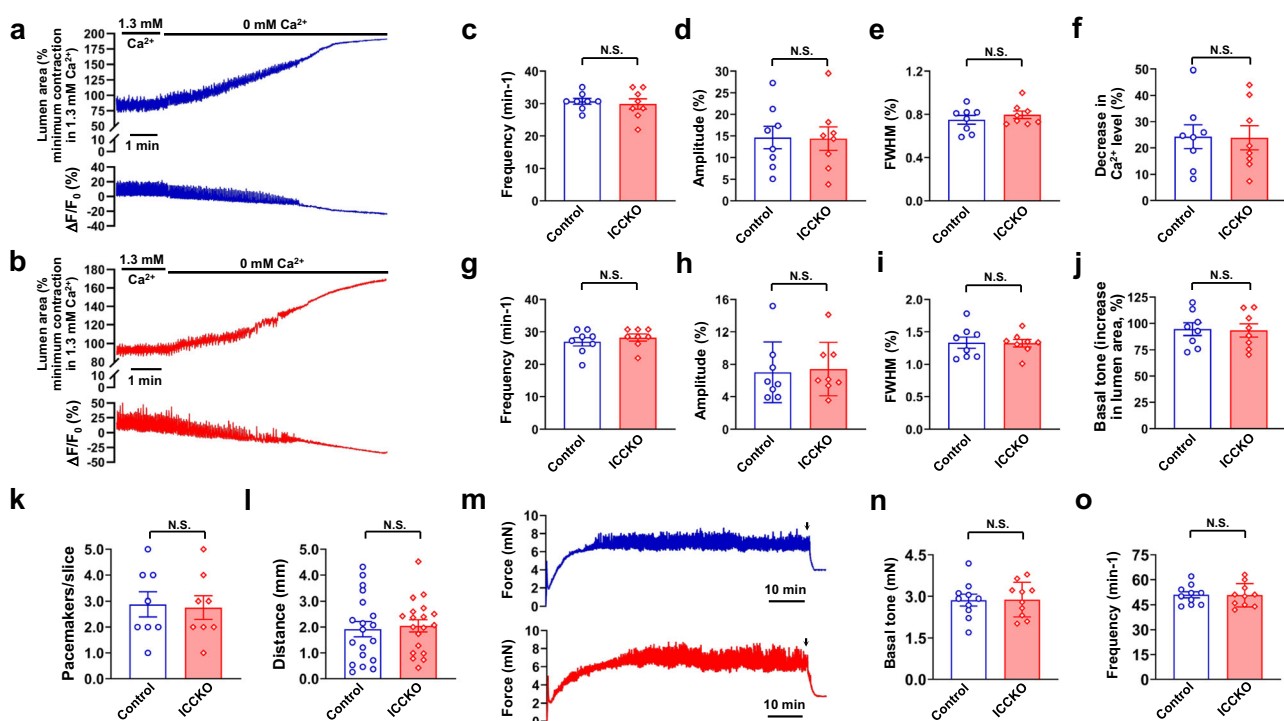

**Fig. 6 TMEM16A in the ICC is not required for pacemaking in the IAS. a, b** Representative recordings of contractions (changes in lumen area) and Ca$^{2+}$ waves ($\Delta F/F_0$(%)) with and without extracellular Ca$^{2+}$ in IAS slices loaded with Cal-520 AM from isogenic control mice (**a**) and TMEM16A$^{ICCKO}$ mice (**b**). **c–e** Summarized results of frequency, amplitude, and FWHM of Ca$^{2+}$ waves in IAS slices from control mice ($n = 8$) and TMEM16A$^{ICCKO}$ mice ($n = 8$) in the presence of 1.3 mM extracellular Ca$^{2+}$. **f** Summarized results of resting intracellular Ca$^{2+}$ levels upon the removal of extracellular Ca$^{2+}$. **g–i** Comparison of frequency, amplitude, and FWHM of rhythmic contractions in IAS slices from control mice ($n = 8$) and TMEM16A$^{ICCKO}$ mice ($n = 8$) in the presence of physiological extracellular Ca$^{2+}$. **j** Summarized results on the basal tone, as revealed by the removal of extracellular Ca$^{2+}$. **k, l** Comparison of the number of pacemakers per slice and the propagation distance of Ca$^{2+}$ waves in IAS slices from control mice ($n = 8$; 19 events) and TMEM16A$^{ICCKO}$ mice ($n = 8$; 19 events). **m** Representative tension recordings of IAS strips from control mice (blue trace) and TMEM16A$^{ICCKO}$ mice (red trace), as assessed by the isometric method. Arrows indicate the time of replacement of normal Ca$^{2+}$ medium with Ca$^{2+}$ free medium for estimation of basal tone. **n, o** Summarized results of basal tone and frequency of rhythmic contractions in IAS strips from control mice ($n = 10$) and TMEM16A$^{ICCKO}$ mice ($n = 10$). N.S. indicates no statistical significance, as determined by unpaired two-tailed Student's $t$ test.

## Discussion

Fecal continence requires a healthy IAS that consistently contracts to maintain high anal canal pressure. This study unveils a cellular and molecular mechanism enabling the IAS to exhibit this contractile capability. We found that the IAS harbors multiple pacemakers, primarily located at the edges or distal terminus of the anus. These pacemakers spontaneously initiate high-frequency calcium waves that propagate, driving rhythmic contractions and the resulting basal tone. Notably, TMEM16A in SMCs, but not in ICC, proves essential for these pacemaker activities in the IAS.

Traditionally, the IAS is classified as a tonic smooth muscle. However, prior seminal studies have indicated that the IAS

generates electrical slow waves and rhythmic contractions similar to other GI phasic smooth muscles, but at a higher frequency[14–16]. Simultaneous recording of Ca$^{2+}$ waves and contractions in our study revealed that these Ca$^{2+}$ waves drive the rhythmic contractions in the IAS. These findings suggest that the Ca$^{2+}$ waves identified in the current study may be the elusive link between slow waves and rhythmic contractions in the IAS. One point of uncertainty regarding this suggestion is that slow waves in mouse IAS have a frequency of approximately 60 min$^{-1}$[18], which is double the Ca$^{2+}$ wave frequency ($\sim$30 min$^{-1}$) we observed. This discrepancy could arise from differences in experimental conditions or the inherent nature of the two types of signals. For instance, in the slow

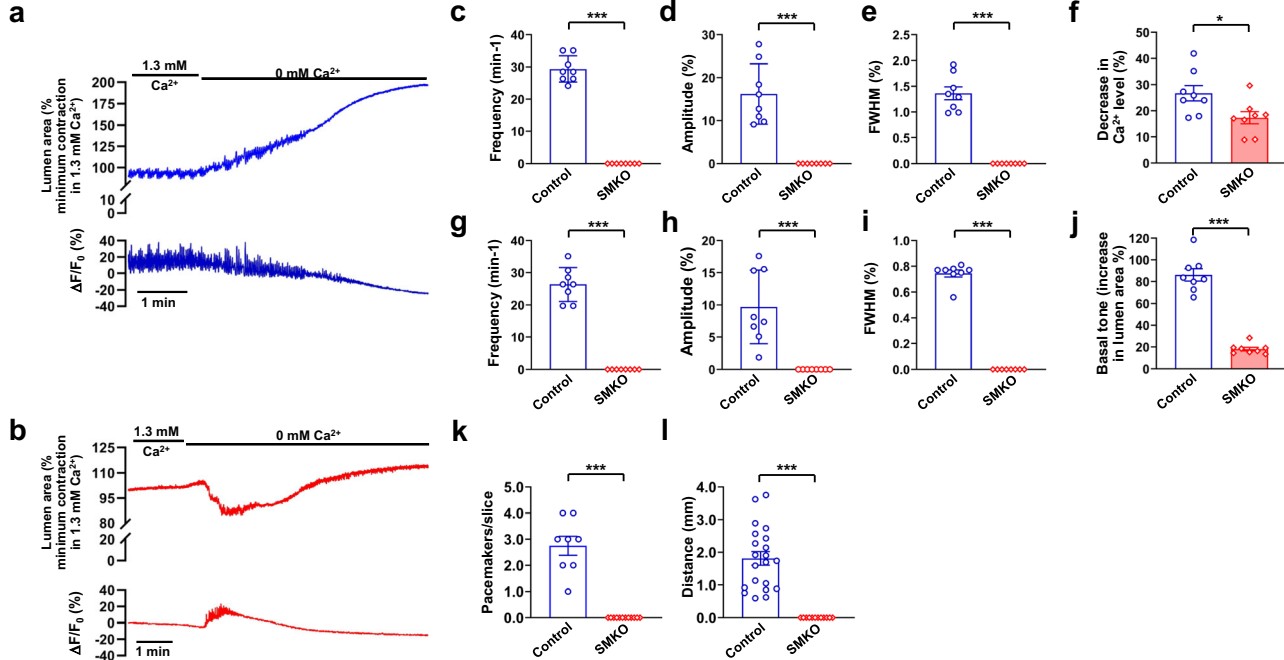

**Fig. 7 TMEM16A in SMCs is required for pacemaking in the IAS. a**, **b** Representative recordings of contractions (change in lumen area) and calcium waves ($\Delta F/F_0$(%)). These recordings were taken from IAS slices loaded with Cal-520 AM and derived from isogenic control mice (**a**) and TMEM16A$^{SMKO}$ mice (**b**). **c–e** Summary of frequency, amplitude, and FWHM of $Ca^{2+}$ waves in IAS slices from control mice ($n = 8$) and TMEM16A$^{SMKO}$ mice ($n = 8$) in the presence of 1.3 mM extracellular $Ca^{2+}$. **f** Summary data of the resting intracellular $Ca^{2+}$ levels after removal of extracellular $Ca^{2+}$. **g–i** Comparison of the frequency, amplitude, and FWHM of rhythmic contractions in IAS slices from control mice ($n = 8$) and TMEM16A$^{SMKO}$ mice ($n = 8$) in the presence of 1.3 mM extracellular $Ca^{2+}$. **j** Summary data showing basal tone changes when extracellular $Ca^{2+}$ was removed. **k**, **l** Comparisons of the number of pacemakers per slice and the distance over which $Ca^{2+}$ waves propagate in IAS slices from control mice ($n = 8$; 21 events for distance) and TMEM16A$^{SMKO}$ mice ($n = 12$; 0 event for distance). *$p < 0.05$, ***$p < 0.001$, as determined by unpaired two-tailed Student's $t$ test. Notes: (1) IAS slices from TMEM16A$^{SMKO}$ did not exhibit rhythmic contractions in the presence of 1.3 mM extracellular $Ca^{2+}$, as shown in (**b**). (2) These slices produced a transient increase in $Ca^{2+}$ concentration and contraction upon the removal of extracellular $Ca^{2+}$. The mechanisms underlying these observations remain to be determined.

wave studies, IAS tissues were bathed at in oxygenated (95% $O_2$/5% $CO_2$) solution at 37 °C, maintained under isometric conditions, and pre-treated with wortmannin[18] or treated with various receptor inhibitors[17–19]. In contrast, in our study, IAS slices were continuously perfused with a solution exposed to atmospheric air (21% $O_2$/0.04% $CO_2$) at 27 °C, allowed to contract isotonically, and loaded with $Ca^{2+}$ fluorescent indicators. Moreover, while slow waves are readings from a single cell impaled with a recording microelectrode, $Ca^{2+}$ waves reflect integrated $Ca^{2+}$ signals from the whole or a substantial area of the IAS slice. Nevertheless, we can't rule out the possibility that the observed frequency difference may be related to the creation of the thin slice preparation possibly causing an altered tissue physiology. Despite the differing frequencies of slow waves and $Ca^{2+}$ waves, both are high frequencies. This suggests they play a crucial role in driving rhythmic contractions and sustaining the basal tone in the IAS.

A notable finding in this study is that each IAS slice contains approximately three pacemakers situated at the IAS edges. These pacemakers are notably more active at the distal end of the IAS. This spatial pattern aligns with previous studies, which indicate a gradient in slow wave frequency— the highest frequency at the most distal end of the IAS and the lowest near the rectum[16,18]— and a higher frequency of $Ca^{2+}$ transients at the distal end of the IAS-SMCs[24]. Our study further shows that $Ca^{2+}$ waves from pacemakers can propagate circumferentially, encompassing a vast area of the slice, occasionally even the entire slice. Interestingly, slow waves tend to propagate further in the circumferential direction than in the longitudinal direction[18]. The observed dominant pacemakers suggest a possibility of IAS pacemaking

mirroring that of the heart. In the heart, while the sinoatrial node sets the pace under normal rhythm, the atrioventricular node, Purkinje fibers, and cardiac myocytes can generate spontaneous activity under abnormal conditions. Although we cannot dismiss this similarity, our findings lean more towards the notion that different pacemakers operate autonomously, given that they are rarely synchronized in phase within each IAS slice.

What do these multiple pacemakers imply functionally? The primary function of the IAS is to create a high-pressure zone in the anal canal and to maintain involuntarily fecal continence. Our data indicate that a single pacemaker seldom induces $Ca^{2+}$ waves that span the entire slice, suggesting that a single pacemaker may be insufficient to generate adequate anal pressure. Furthermore, these pacemakers differ not only in their spatial influence but also in their frequency and activation times. Hence, we propose that the overall basal tone of the IAS, and thus the pressure in the anal canal, is determined by the combined activities of various pacemakers, each contributing distinct kinetics to $Ca^{2+}$ wave generation. This multi-pacemaker system assures persistent IAS contraction, crucial for prolonged fecal continence and offers modulation flexibility, accommodating the movement of different contents.

Another major revelation in this study is that TMEM16A in ICC is neither necessary nor sufficient for IAS pacemaking and basal tone. Previous evidence has suggested that TMEM16A in the ICC may serve as a pacemaker channel in the IAS[17,19]. However, our experiments utilizing Kit$^{CreERT2}$ mice for specific and temporally controlled TMEM16A deletion in ICC clearly demonstrate that the absence of TMEM16A in ICC has no impact

on the number of pacemakers, $Ca^{2+}$ wave generation, rhythmic contractions, or basal tone. Additionally, the presence of TMEM16A in ICC in TMEM16A[SMKO] mice failed to generate these responses. These findings compellingly argue against the role of TMEM16A in ICC as a necessary component for spontaneous contractions in the IAS. Existing literature suggests that the ICC may not even be essential for the generation and maintenance of IAS basal tone. W/W[v] mice, which carry a c-kit mutation leading to an ICC deficiency or even absence of ICC in the GI tract[21,37–39], still maintain an IAS basal tone comparable to that of wild-type mice[20–23]. IAS slow waves are also indistinguishable between the two mouse types[22]. Using in vivo anorectal manometry, de Lorijn et al. observed no significant difference in anal canal pressure between wild-type and W/W[v] mice[21]. Thus, neither TMEM16A in the ICC nor the ICC themselves are pivotal for establishing IAS basal tone. However, both our study and previous research validate the expression of TMEM16A in the ICC of the IAS[17,20,31]. There is also substantial evidence for the involvement of the ICC in pacemaking, neurotransmission, and mechanosensing in other GI smooth muscles like the stomach and intestine[36,40,41]. It's plausible that, the ICC or TMEM16A within them, although not essential for generating the IAS basal tone, might still modulate the tone in response to neuronal activity or mechanical stimuli in the anorectum[21,22].

Bio-engineered "IAS rings," created from isolated IAS-SMCs, spontaneously generate a basal tone both in vitro and in vivo, mirroring that of the native IAS[42–46]. These pioneering studies suggest that IAS-SMCs inherently possess the molecular machinery responsible for rhythmic contractions. In our current study, we identified that TMEM16A in IAS-SMCs is essential for pacemaking within the IAS, as TMEM16[SMKO] IAS slices do not produce spontaneous $Ca^{2+}$ waves and rhythmic contractions. Moreover, TMEM16A is expressed exclusively in the SMCs located at the periphery and distal end of the IAS. The spatial distribution aligns with the anal canal pressure profile with a higher pressure zone at the distal end of the anus gradually decreasing toward the rectum[23]. Considering that pacemakers cover only $0.69 \pm 0.13\%$ of the IAS slice area, a value smaller than the area occupied by TMEM16A-expressing SMCs, we speculate that a subset of TMEM16A-expressing SMCs serves as pacemaker cells. Consequently, we've identified at least three classes of SMCs in the IAS: TMEM16A positive pacemaker SMCs, TMEM16A positive non-pacemaker SMCs, and TMEM16A negative SMCs. Exploring the factors that delineate these different SMC types in a spatially dependent manner would be an intriguing next step.

In light of the results from our current study, a discussion of TMEM16A expression and $Ca^{2+}$ oscillations in ICC[17,19], as studied by the Keef and Cobine group, is warranted. Firstly, these authors reported that, after sorting cells from the IAS, TMEM16A expression was 26.5-fold higher in ICC than in SMCs[17]. However, as shown in our study, TMEM16A is expressed only in a subset of SMCs located at the edge and the distal end of the IAS (Fig. 5). Therefore, spatial sampling variations may account for the reduced expression of TMEM16A in SMCs[17]. Secondly, the expression profile in the sorted cells may not accurately reflect the in situ expression. This is due to the potential effect of enzymatic digestion and the time elapsed between cell dissociation and sorting on TMEM16A expression in ICC and SMCs, as observed in studies of various other cell types[47–49]. Thirdly, these authors identified a subpopulation of ICC, termed "Type II ICC," that generate rhythmic, global $Ca^{2+}$ transients at the slow-wave frequency[19]. Could these $Ca^{2+}$ transients be identical to the $Ca^{2+}$ waves observed in the current study? Our assessment suggests that they are different. The $Ca^{2+}$ transients in ICC are confined to a smaller area and have a shorter propagation distance compared to the $Ca^{2+}$ waves we observed. This aligns with estimates

suggesting that ICC constitutes approximately 5% of the IAS volume[50]. One might wonder whether the $Ca^{2+}$ transients in type II ICC could trigger the $Ca^{2+}$ waves observed in the current study. Our TMEM16A[ICCKO] experiments suggest that this is unlikely, as the $Ca^{2+}$ waves remain unchanged after TMEM16A deletion in ICC. However, when TMEM16A is deleted in SMCs, $Ca^{2+}$ waves are abolished, suggesting that the $Ca^{2+}$ transients in type II ICC may originate from IAS-SMCs.

The development of TMEM16A antagonists has facilitated the identification of the biological functions of TMEM16A in various tissues and organs, including smooth muscle[51–55]. However, recent studies have also raised some uncertainties about the utility of these TMEM16A antagonists, primarily due to their potential off-target effects[25–27]. We observed that three TMEM16A antagonists had only marginal effects on inhibiting $Ca^{2+}$ waves and rhythmic contractions in IAS slices, suggesting weak inhibitory effects on TMEM16A activity. This finding is somewhat surprising, considering previous studies have demonstrated that 16Ainh-A01 and CaCCinh-A01 can abolish slow waves and basal tone under isometric conditions[17,31]. The reasons for this discrepancy are yet to be determined. Given the differences in experimental conditions, such as inhibitor administration, $O_2$ and $CO_2$ levels, temperature, and stretch, between these measurements, it is plausible that one or a combination of these factors may contribute to the inconsistency. Moreover, these uncertainties and inconsistencies highlight the benefits of a cell-specific genetic knockout approach in studying TMEM16A function in IAS and other smooth muscle tissues[31,56–58]. Nevertheless, it is important to consider the possibility of compensatory gene expression changes in any transgenic approach, which may influence the results.

In summary, using precision-cut IAS slices, cell-specific gene knockout techniques, and wide-field fluorescence microscopy, we determined that intercellular $Ca^{2+}$ waves, originating from multiple fixed pacemakers in SMCs, drive rhythmic contractions leading to basal tone in the IAS. These propagating calcium waves function independently of signals from nerves or other cells within the IAS. It is TMEM16A in SMCs, and not in ICC, that is essential for pacemaking. Our findings not only deepen our understanding of the mechanisms underlying IAS basal tone, but also pave the way for future studies on the role of TMEM16A in fecal incontinence and other anorectal motility disorders.

## Methods

**Mice.** All animal procedures were conducted with approval from the Institutional Animal Care and Use Committees at the University of Massachusetts Chan Medical School (protocol number A1473), following the guidelines set forth in the National Research Council Publication Guide for the Care and Use of Laboratory Animals and NIH Guide for the Care and Use of Laboratory Animals. Mice were housed under a standard 12-hour light/dark cycle (lights on at 07:00 AM) with ad libitum access to food and water at room temperature of $22 \pm 2.0\ °C$. Throughout the study, mice aged 8–12 weeks were used, and within each strain and experiment, they were aged-matched. Results are pooled from both male and female mice.

TMEM16A[SMKO] mice were bred and validated as described in our previous studies[20,31]. TMEM16A[SMKO] mice, as well as their isogenic controls, had a mixed genetic background comprised of C57BL/6 and Sv/129 mice. TMEM16A[ICCKO] mice, with a C57BL/6 genetic background, were generated and verified, following the methods outlined in our prior study[20]. The *Tmem16a* deletion in TMEM16A[ICCKO] was induced through intraperitoneal injection of tamoxifen (10 mg/mL) dissolved in 100 μL corn oil for 5 consecutive days. Functional tests were conducted 21 days after the final injection. In this study, a total number of 18

TMEM16A$^{SMKO}$ and 18 TMEM16A$^{ICCKO}$ mice were utilized. An equal number of isogenic controls were also employed for each respective strain. Additionally, 39 C57BL/6 mice were employed for pharmacological experiments.

**Isometric force measurement of IAS strips.** The anal canal and adjacent rectum (~1.5 cm in length) were promptly removed and immersed in ice-cold and oxygenated Krebs physiological buffer (KPS), comprising the following concentrations (in mM): 118.07 NaCl, 4.69 KCl, 2.52 CaCl$_2$, 1.16 MgSO$_4$, 1.01 NaH$_2$PO$_4$, 25 NaHCO$_3$, and 11.10 glucose. Skeletal muscle fibers and other extraneous tissues were carefully removed and discarded, leaving the anal canal intact. The IAS, recognizable as a thickened, circular smooth muscle at the lowermost part of the anal canal, was preserved. The IAS rings were inverted to facilitate the removal of the mucosal layer. Following removal, the rings were trimmed to approximately 1 mm in width and then longitudinally cut into two halves for subsequent force measurement.

The IAS strips were transferred to 37 °C oxygenated (95% O$_2$/5% CO$_2$) KPS, with each end connected to the force transducer of a myograph system (610-M, Danish Myo Technology, Aarhus, Denmark) and a 5 mN load applied immediately. Typically, the force initially declined to approximately 2 mN and then ascended, reaching a sustained level with superimposed rhythmic contractions.

To assess the IAS strips' basal tone, the KPS buffer was replaced with calcium-free KPS (in mM: 120.85 NaCl, 4.69 KCl, 1.16 MgSO$_4$, 1.01 NaH$_2$PO$_4$, 25 NaHCO$_3$, 1.0 EGTA, and 11.10 glucose), and the tension was recorded over a sufficient duration to achieve a stable minimum force (i.e., complete relaxation) (see Fig. 6m). The basal tone was calculated as the difference between Tension$_{Ca}$ and Tension$_{0Ca}$, where Tension$_{Ca}$ represents the mean tension of the last 1 min in the presence of normal KPS buffer before switching to the calcium-free KPS, and Tension$_{0Ca}$ is the mean tension of the last 1 min during calcium-free KPS treatment.

**Calcium and contraction measurements in precision-cut IAS slices.** The anal canal and the adjacent rectum were dissected in ice-cold Hanks' balanced salt solution (HBSS, Sigma Aldrich), supplemented with 20 mM HEPES buffer and adjusted to pH 7.4, hereafter referred to as sHBSS. Following the removal of skeletal muscle, the mucosal layer, and other extraneous tissues as detailed above, intact IAS rings were reinverted to their original orientation and embedded in 5% low melting point agarose gel. After solidifying at 4 °C, the rings were sectioned into 250 μm thick slices using a compresstome vibratome (VF-300-0Z; Precisionary Instruments, Greenville, NC, USA). Consecutive slices were sequentially placed into separate wells of a 24-well plate containing sHBSS buffer and then examined under a light microscope. The first two slices with intact smooth muscle rings from the direction of anal verge were chosen for testing. This criterion ensured that only the circular smooth muscle bundles from the IAS, free from any contamination by the adjacent rectum, were used.

IAS slices were loaded in darkness at 31 °C for approximately 30 min with sHBSS containing 20 μM Cal-520 AM (AAT Bioquest, Inc., Sunnyvale, CA, United States), 0.1% Pluronic F-127 and 200 μM sulfobromophthalein. Subsequently, they were kept in sHBSS containing 200 μM sulfobromophthalein for an additional 30 min at room temperature to facilitate the de-esterification of Cal-520 AM. For contraction and calcium measurements, IAS slices, with the agarose gel removed from within the lumen but still attached to their outer edge, were positioned on a cover-glass. This cover-glass was mounted in a custom-made Plexiglas support. The slice was then secured with a 200 μm nylon mesh, ensuring that a hole in the mesh aligned directly over the slice. Another smaller

cover-glass was placed atop the nylon mesh to create a perfusion chamber, and the edges were sealed with silicone grease. A gravity perfusion system generated a perfusion flow rate of approximately 500 μl min$^{-1}$, controlled by a VC-6 six-channel valve controller (Warner Instruments Corp.). The sHBSS in the perfusion reservoir was exposed to atmospheric air (21% O$_2$/0.04% CO$_2$). All experiments were conducted within a custom-made Plexiglas chamber equipped with a custom-built objective heater and thermal controller to maintain a temperature of 27 °C. μManager software, operating a custom-built wide-field digital imaging system[59], was employed to record contractile activities and Ca$^{2+}$ signals in IAS slices. Fluorescence images of the entire IAS slice were captured using a 2× objective (Nikon, Tokyo, Japan) interfaced with an IX71 Olympus inverted microscope, and the camera (DU-885K-CSO-#VP EMCCD, Andor Technology) acquired images at an 11 Hz speed. Fluorescence excitation was provided by the 488 nm line of an argon-ion laser, with exposure duration controlled by a shutter; emission of the Ca$^{2+}$ indicator was detected at wavelengths >510 nm.

For quantifying changes in Ca$^{2+}$ within IAS slices, Ca$^{2+}$ fluorescence in the circular smooth muscle was measured. As increases in Ca$^{2+}$ induce contractions of IAS slices, manually outlining the inner and outer boundaries of the circular smooth muscle at each time point to track Ca$^{2+}$ changes could be laborious. Thus, we developed a semi-automatic method for extracting Ca$^{2+}$ fluorescence in the circular smooth muscle using ImageJ/Fiji (https://imagej.net/software/fiji/). To achieve this, for an image sequence, we initially marked the outer edge of the circular muscle during the frame showing maximal relaxation or minimal contraction (i.e., the frame with the largest lumen area in the presence of extracellular calcium or the last frame when extracellular calcium was removed) and the inner edge during the frame showing maximal contraction (i.e., the frame with the smallest lumen area). We then utilized these markers to define the area between the two edges as the circular smooth muscle. To evaluate the accuracy of this method, we compared its results to frame-by-frame manual extraction of fluorescence in the circular smooth muscle. Supplementary Fig. 1 illustrates that changes in Ca$^{2+}$ assessed through the semi-automatic extraction method were not significantly different from those obtained through manual extraction ($n = 9$, $p = 0.99$). This suggests that Ca$^{2+}$ signals predominantly occurred within the circular smooth muscle. As a result, we employed the semi-automatic extraction method for Ca$^{2+}$ fluorescence measurements.

Prior to analysis, a bleach correction using Exponential Fit (accessed through the ImageJ/Fiji menu Image>Adjust>Bleach Correction) was applied to the images. Changes in fluorescence intensity were represented as $(F_t - F_0)/F_0*100$, denoted as $\Delta F/F_0(\%)$, where $F_t$ indicated the fluorescence intensity at a specific time, and $F_0$ denoted the minimal fluorescence in the presence of 1.3 mM extracellular Ca$^{2+}$. Time courses of $\Delta F/F_0$ were exported to OriginPro 2022, and Ca$^{2+}$ events were identified using the local maximum method of the OriginPro Peak Analyzer. The values of local points and amplitude height % were adjusted manually to ensure detection of over 95% of peaks. Any missing or erroneously selected peaks were rectified or removed through visual verification of the original images. The frequency of Ca$^{2+}$ events was subsequently calculated by dividing the number of peaks by the total time. The full width at half maximum (FWHM) of Ca$^{2+}$ signals was determined using the Multiple Peaks Fit with the Lorentz Peak Function in OriginPro 2022.

A pacemaker was defined as an area of 100 μm radius centered at the pixel from which the largest number of Ca$^{2+}$ waves was initiated during a 2.5-min recording. This radius was selected because Ca$^{2+}$ waves propagate in multiple directions, and it was sufficiently large to capture the start of any Ca$^{2+}$ wave, given the

imaging acquisition rate of 11 Hz and $Ca^{2+}$ wave propagation speed of 1.11 mm s$^{-1}$ as measured (see "Results").

For the quantification of contractions, the lumen area of IAS slices was measured and analyzed with ImageJ/Fiji[20]. After selecting an appropriate grayscale threshold to distinguish the lumen from the surrounding area, the lumen area of the IAS slice was calculated by summing the pixels frame by frame. The summed area values were normalized to the maximal relaxation or minimal contraction during rhythmic contractions in the presence of 1.3 mM extracellular $Ca^{2+}$. The percent relaxation, corresponding to the basal tone, was calculated as $(Area_{0Ca} - Area_{Ca})/Area_{Ca}*100$, where $Area_{Ca}$ represented the mean lumen area of the last 30 s in the presence of 1.3 mM $Ca^{2+}$ before any treatment, and $Area_{0Ca}$ represented the mean lumen area of the last 30 s during calcium free treatment. The frequency and FWHM of rhythmic contractions were calculated in OriginPro using the same method described above for the $Ca^{2+}$ signal analysis. Calcium wave propagation distances were determined by manually tracing along the central axis of the waves using ImageJ/Fiji. In cases where calcium waves from a pacemaker propagated in two opposite directions, the distances from both directions were summed.

**Immunohistochemical analyses**. Anorectal tissues were isolated and immediately embedded in optimal cutting temperature compound (Bio-Tek). Cryosections with a thickness of 10 μm were fixed in pre-cooled acetone for 10 min and washed thrice with phosphate-buffered saline (PBS) at room temperature. To block non-specific antibody binding, cryosections were incubated with PBST (0.3% Triton in PBS) containing 1% bovine serum albumin for 1 h. They were then incubated overnight at 4 °C using a rabbit polyclonal antibody against TMEM16A (ab53212, 1:100; Abcam) and a goat polyclonal antibody against c-Kit (AF332, 1:20; R&D Systems) for TMEM16A$^{ICCKO}$ mouse samples and their isogenic controls. Alternatively, we used the same rabbit polyclonal antibody against TMEM16A (1:100) and a mouse monoclonal antibody against Myh11 (ab683, 1:400; Abcam) for TMEM16A$^{SMKO}$ mouse samples and their isogenic controls. The specificity of these antibodies had been verified by manufacturers and different researchers including us[20,31]. After washing with PBS, the sections were treated with a secondary antibody solution, containing Alexa Fluor 488-conjugated donkey anti-rabbit immunoglobulin G (IgG) H&L (ab150061, 1:500; Abcam) and Alexa Fluor 568-conjugated donkey anti-goat IgG (H+ L; ab175474, 1:500; Abcam) for TMEM16A$^{ICCKO}$ samples and their controls. Alternatively, the mixture included the same donkey anti-rabbit immunoglobulin G (IgG) H&L (1:500) and Alexa Fluor 594-conjugated donkey anti-mouse IgG (H + L; ab150112, 1:500; Abcam) for TMEM16A$^{SMKO}$ samples and their controls. Secondary antibodies were incubated for 2 h at room temperature. As a negative control, primary antibodies were omitted. Immunoreactivity was assessed using a Leica TCS SP8 confocal microscope (Leica Microsystems Inc.), with images captured at 20× magnifications and a Z-plane interval of 0.5 μm. We analyzed the images using ImageJ/Fiji, focusing on regions of circular smooth muscle and excluding the surrounding areas. To account for background signals, we measured the intensity from the respective negative control channel. The intensity average plus 2 standard deviations from the negative control was then subtracted from each image. We evaluated colocalization between TMEM16A with either c-Kit or Myh11 using thresholded Mander's colocalization coefficients through the JACoP image plugin[60]. A value of 0 indicated no colocalization, while 100% denoted complete colocalization.

**Statistics and reproducibility**. Data are presented as means ± s.e.m., with "n" denoting the number of animals used. Differences between groups were analyzed using either paired or unpaired Student's $t$ tests. For zonal quantification of TMEM16A-containing pixels, a one-way analysis of variance (ANOVA) followed by Tukey's post hoc test was employed to ascertain significant differences. Significance levels are represented as: N.S. (not significant) for $p > 0.05$, * for $p < 0.05$, ** for $p < 0.01$, *** for $p < 0.001$.

**Reporting summary**. Further information on research design is available in the Nature Portfolio Reporting Summary linked to this article.

## Data availability

All data supporting the findings of this study are included in the article and its Supplementary Information. Numerical source data for all charts and graphs can be found in Supplementary Data 1 and any remaining information can be obtained from the corresponding author upon reasonable request.

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

## Acknowledgements
This work was supported by the National Institute of Health (R01HL139686 and R21HD097458 to R.Z.).

## Author contributions
PL designed and performed the experiments. PL, KB, and LL analyzed the data, supervised instruments, or performed statistical analysis. RZ conceived the study, designed the experiments, and analyzed the data. PL and RZ wrote the manuscript with contributions by LL. All authors reviewed and approved of the final manuscript.

## Competing interests
The authors declare no competing interests.
