## [Peer Review File · Communications Biology]

Reviewers' comments:

Reviewer #1 (Remarks to the Author):

General comments:

The aim of this study was to characterize the molecular mechanisms underlying the generation of tone in the internal anal sphincter (IAS). The authors used calcium dye-loaded vibratome sections of the distal IAS to examine pacemaker activity and the spread of calcium activity in the circumferential direction. They also determined the contribution of TMEM16A channels in smooth muscle cells (SMCs) and interstitial cells of Cajal (ICCs) using two different mice in an effort to obtain a cell specific deletion of TMEM16A. This is similar to their previous study published in 2021. There are some concerns that need to be addressed particularly specifics from papers by another group pertaining to the studies in this current manuscript that have not been considered fully. These are outlined in the reviewer comments below.

Specific comments:

Abstract/introduction:

Please more clearly state the aim of the study and the hypothesis being tested. It is not immediately obvious how the aims of this study are different to the previously published study (Lu et al., 2021).

Methods and Materials:

p.4. lines 97-109. Please provide the number of animals used for each strain and also indicate this for each experiment throughout the study. Specifically, it hasn't been provided for TMEM16A SM-KO or TMEM16A ICC-KO mouse data.

p.4. lines 103-109. What tamoxifen treatment strategy was used for TMEM16A SM-KO mice? How much knockdown was observed in these and the TMEM16A ICC-KO mice? While this was characterized in previous studies

p.4. line 118. How were the rings cut into two halves? In the circular direction or the longitudinal direction? What were the dimensions of the muscle strips used for isometric tension recordings?

p. 5. line 126. Tension is spelled incorrectly three different times in the same sentence, also on line 128.

p. 5. Line 130. Was the mucosa removed? If so, how and were the rings imaged with the circular muscle innermost? The supplemental figure in the previous paper indicates that the ring was inverted but it's not clear as to whether it was then subsequently reinverted again. What procedures were used to ensure that there was no damage to the muscle during this process? Were the rings subject to an agonist such as carbachol or KCl to test viability?

p. 5. Lines 144-147. Was the agarose gel still in place during recording? Does the agarose expand the lumen and stretch the ring in any manner?

p. 5. Lines 144-147. This chamber is difficult to visualize. A picture is shown in the supplemental data section of the previous paper but it is small and blurry when zoomed in. Please provide a higher quality, larger size depiction of the recording chamber. Is the ring distorted in any way by the nylon mesh or coverslip? Where are the inflow and outflow located, i.e., to the sides of the ring or to the lumen? How close is the ring to the coverslip? Is there sufficient room for perfusion of the entire preparation? How do the authors ensure adequate flow of solution and added drugs to the entire preparation?

p. 5. Lines 148-150. Why was 27°C used? This is not a physiological temperature and is highly likely to affect the activity of the tissue (see Burdyga and Wray, 2002). The authors should note that the studies by Hall et al., 2014, Cobine et al., 2020 and Hannigan et al., 2020 were all done at

370C.

p. 5-6. Lines 161-164. What is meant by maximal relaxation and maximal contraction? How was this determined? Without addition of a substance that would cause relaxation or removal of extracellular calcium it would be difficult to establish maximal relaxation. In a similar manner, to determine maximal contraction, an agonist such as carbachol or high K⁺ would be required.

p. 6. Lines 164-167 and Fig. S1. There is a yellow line that depicts the "outer" edge but this is in the middle of the ring. What is located at the outer edge of the ring to the exterior of the yellow line? An image of the ring segment before and after removal of the extracellular calcium should be included in Supplementary Figure 1 to make it clearer what the ring looks like without extracellular calcium and to better evaluate the contractile data plotted in the graphs.

p. 6. Lines 187-191 and Figures 1,5 and 6. The ring is not maximally relaxed in normal extracellular calcium. The reviewer is not sure what the authors are trying to state here.

Results:

p. 7. Lines 213-214. How many hours? Was any bleaching of the signal observed? Typically, photobleaching is observed with continuous recording in tissues loaded with calcium dyes (Hennig et al., 2015 (Frontiers Cellular Neuroscience); Drumm et al., 2022 (J Physiol review)).

p. 7. Lines 213-215 and Figure 1A,B. How does the lumen diameter in normal extracellular calcium compare to the diameter of a ring contracted with an agonist or KCl? What is the unit used for lumen area and why is "maximal relaxation" depicted as "100"? The recording in Fig. 1A appears to show that lumen area doubles following removal of calcium (i.e., from 100-200). Is this the case? This differs significantly from the previous 2021 study in which removal of extracellular calcium increased lumen area to 145 from 100 (i.e., by 45% not 100%). These values suggest very different levels of basal tone between studies utilizing the same preparation and methodology.

p. 7. Lines 218-220 and 232-234. How did the authors confirm that the calcium oscillations arose from smooth muscle cells? With a dye loaded tissue all cell types are being visualized so it is possible that this signal may have arisen from an ICC and conducted to adjacent SMCs as has been suggested in other studies.

p. 7. Lines 242-245. What is the significance of the activity spreading both perpendicular and parallel to the circular muscle SMCs?

p. 7. Lines 259-268 and Figures 1 and 2D. How do the authors account for a frequency of ~24 contractions and calcium oscillations per minute when studies in whole muscle strips have revealed a frequency of ~50-60 contractions per minute (Duffy et al., 2012 and Figure 5 of present study) and studies in mice with GCaMP expressed specifically in SMCs (Cobine et al., 2020) have revealed a frequency of ~70 calcium transients per minute?

p. 8-9. Lines 290-294 and Figure 4. It should be noted that niflumic acid is not a specific CaCl channel inhibitor as it has been shown to have other off target effects including blocking non-selective cation channels. Did the authors test CaCCinh-A01 on contractions in isolated muscle strips or determine their effects on lumen area in these rings? With the perfusion system used are the authors confident that the entire ring was exposed to these drugs? A 2016 study by the same authors (Zhang et al., 2016) demonstrated that all IAS contractile activity was abolished by T16Ainh-A01, another CaCl channel blocker and that in fact it was more efficacious than niflumic acid (100% inhibition with T16Ainh-A01 compared to 80% with NFA). A 2017 study by Cobine et al. demonstrated that T16Ainh-A01 and the same concentration of CaCCinh-A01 used in the current study both completely abolished phasic and tonic contractile activity as well as electrical slow waves in IAS muscle strips. Additionally, a study by Drumm et al. (J Physiol 2022) demonstrated that a lower concentration of Ani9, i.e., 3 uM completely abolished tone in lower esophageal sphincter muscle strips. Thus, a lack of response to these antagonists in the present study is highly curious. It is possible that the lack of effect of these more specific CaCl channel

inhibitors has something to do with either mixing issues associated with drug delivery, inefficient access to the entire muscle ring, problems with the quality of the drugs or with precipitation of drug during dilution. The data with CaCCinh-A01 is also difficult to interpret as the control activity in the example provided seems less than in either the niflumic acid or Ani9 examples and with no removal of extracellular calcium shown it isn't clear how the change in calcium fluorescence is normalized. These experiments should be repeated to better characterize this. Also, authors need to include data demonstrating the effects of these drugs (specifically NFA since that had an effect) on contraction and lumen area in both the TMEM16A ICC-KO and the TMEM16A SM-KO as one would expect no effect of these drugs in the mouse where the TMEM16A has been knocked out of the pacemaker cell.

p. 9. Lines 312-317 and Figure 5. The unit for contraction on the summary graphs is incorrect, i.e., should be mN not N. It should be noted that in this Figure as well as throughout the manuscript the change in fluorescence has been plotted as a percentage. This does not allow the reader to see if the actual basal fluorescence has changed in any way. For example, if the basal fluorescence (F) has diminished then the percentage change will be affected as well as it may be close to zero (F_0) before any maneuver that would alter fluorescence is applied. Therefore, it is important to represent the actual F and F_0 numbers as important information may be hidden in representing it in this manner. In the same vein, the absolute lumen area should be provided and not just given an arbitrary value of 100. Some of the images in the manuscript show a smaller diameter lumen (e.g., Figure 1D) than others (e.g., Figure 1C).

It should also be noted that in the gastric antrum where TMEM16A in ICC has been shown to be responsible for pacemaker activity studies have utilized a similar model (Hwang et al., 2019) and while there was a significant reduction in contractile activity it was not completely abolished just like in the 2021 study by the authors of the current study. The study by Hwang et al. waited 50 days after the final tamoxifen injection to ensure maximal knockdown whereas the authors in this study waited only 21 days. Thus, perhaps this is not sufficient to deplete all or at least the majority of TMEM16A expression. Therefore, one must question if TMEM16A truly is absent/diminished from all ICC in these mice. Perhaps the authors can compare mice between these timepoints and examine if there are further changes in TMEM16A gene and protein expression. The authors have referred to their 2021 study to demonstrate the loss of TMEM16A in ICCs. However, these immunohistochemical studies don't show significant levels of TMEM16A in SMCs. Thus, these studies should be repeated in both TMEM16A ICC-KO and SM-KO mice as a direct comparison.

p. 10. Lines 340-343. Why are the parameters measured different to those stated for the TMEM16A ICC-KO, i.e., here phasic contraction amplitude is presented as a percentage whereas for the TMEM16A ICC-KO phasic contractions are not mentioned but rather tone as an absolute mN value. Why isn't both tone and phasic contractions addressed for each mouse so that it can be evaluated more clearly? Also, why are the values for the SM-KO mice presented as percentages in the text whereas for the ICC-KO mouse it is presented in the figure as a percentage but an absolute value in the text? What are these percentages of (e.g., amplitude 9.69 % in control). This needs to be explained and the data needs to be presented the same way between different mouse groups to better assess any differences present.

p. 10. Lines 346-350 and Figure 6. Was the lumen larger in these mice compared to control mice? That isn't clear from the text. If so, then this would explain a lesser change in lumen area with removal of extracellular calcium. Again, absolute lumen area should be provided. This should be compared in muscle strips to see if there is the same phenomenon. It is rather curious that removal of extracellular calcium should decrease the lumen area but that the area at the end of the period where extracellular calcium was removed is very similar to that at the beginning of the experiment in normal calcium when the other figures show an almost doubling of the lumen area under the same experimental conditions. This and the dramatic effect on calcium waves as compared to what is observed with TMEM16A inhibitors makes one question whether the tissue was even functional. To verify that it is, the authors need to do some experiments showing that they are able to elicit a decrease in lumen area (and increase in contraction) by addition of an agonist such as carbachol or ACh, an activator of Cav channels, e.g., Bay K 8644 or high K^+ (60-80mM). In fact, this should be routinely performed to address the issue of differences in the lumen size being noted. The differences in the lumen size (again, see Figure 1C vs. 1D) calls into question as to whether some rings have been subject to stretching during the removal of the

mucosa or the embedding process. In the author's 2016 study they showed that the tone in the TMEM16A SM-KO mouse was reduced by 50% but not entirely abolished. How does this fit with the complete abolishment of calcium waves in the same mouse within the present study?

p. 9-10. Lines 333-356. To further support the selective removal of pacemaker activity in the IAS of the TMEM16A SM-KO mouse a positive control is required. The authors could again like they did in their 2021 study use the gastric antrum to serve as a control to document the selectivity of block in the IAS by comparing the contractile activity of control and TMEM16A SM-KO antral muscle strips. Additionally, the present study concludes that phasic activity is generated by a unique mechanism in the IAS which differs from other GI regions. Have there been any published studies by other groups using a similar inducible TMEM16A SM-KO mouse that have reported a loss of phasic activity and/or tone when TMEM16A has been knocked down in SMCs (e.g., in blood vessels)? If so, please discuss, if not then indicate that this current study is the first.

Discussion:

p. 10. Lines 369-372. It should be noted that although the authors in this study (Cobine et al., 2020) were using a different preparation they were visualizing calcium transients in the circular muscle of the entire IAS and rectum. The strategy of that study was to determine whether the tonic and phasic behavior of intracellular calcium in the tone-generating IAS differs from that of more proximal regions where tone is absent. This study provided strong evidence that that is the case. Phasic calcium transients were greatest in amplitude and frequency at the distal IAS and conducted with decrement in the proximal direction. Even more relevant, basal calcium levels were also greatest in amplitude at the distal IAS again declining in the proximal direction. The authors of this current manuscript state that Cobine et al., 2020 conclude "calcium transients propagate preferentially in the oral (longitudinal) direction" however this was not stated in their paper.

p. 10-11. Lines 365-389. The authors argue that their 250 um thick ring segment has numerous advantages over muscle sheets pinned in a tissue bath. However, to create a muscle ring both surfaces are damaged and the segment must be disconnected from the remaining syncytium. Their technique also precludes inclusion of cells located at the distal edge of the IAS since sections were excluded until "a complete circular muscle ring" was obtained. The authors also comment that their rings are "free to contract isometrically" while muscle sheet are confined to isotonic contraction. However, since the internal anal sphincter is embedded within the external anal sphincter, mucosa/anoderm and glands along with a great deal of connective tissue, the ability of the IAS to change length in vivo is far less than what is permitted in these isolated ring experiments where length can change from "maximum contraction" to twice "maximum relaxation".

p. 11. Lines 379-381. The authors suggest that the Cobine et al., 2020 paper conflicts with their earlier study (Hall et al., 2014) in which "slow waves were coordinated over a much greater distance in the circumferential direction than in the oral direction". However, this not the case. The fact that there is a high degree of coordination in the circumferential direction (e.g., at least 0.5 mm, Hall et al., 2014) made it possible to confidently compare signals from the distal IAS to those of progressively more proximal regions during a single recording. The real take home message from the study was that large, rapid frequency calcium transients arise in the IAS which lead to an increase in basal calcium levels in SMCs and tone development. As the amplitude and frequency of calcium transients decline in the proximal direction, so does elevated basal calcium and tone. This conclusion was published more than two years ago but was not mentioned by the authors in their manuscript.

p. 11. Lines 407-410. The authors state the presence of the highest frequency slow waves in the most distal IAS and the lowest frequency next to the rectum suggests that the IAS only has pacemaker cells next to the skin. However, the 2014 study by Hall et al. recorded slow waves and phasic activity in isolated subsections of the IAS and showed that EACH segment has slow waves and phasic activity with the distal-most section always having the most rapid frequency slow waves and phasic activity, i.e., there is a progressive decline in slow wave frequency and amplitude (pacemaker activity) from distal IAS to rectum. This has also been shown in the rectoanal region of the monkey and dog (see Keef and Cobine, 2019 review J Neurogastroenterol Motil). This argues against pacemakers being confined to the region next to the skin. A surprising observation

of the Cobine et al., 2020 full muscle sheet imaging study was that the fastest pacemaker, i.e., the one located most distally, conducted not only circumferentially but also in the proximal direction. Thus, for some distance it predominated over the innate frequency of the more proximal region. This behavior is similar to the heart where pacemakers of decreasing frequency (i.e., SA node > AV node > Purkinje fibers) have been identified with the fastest pacemaker, i.e., the SA node setting the heart rate frequency. When the distal most section of the mouse IAS (i.e., most distal 300 μ m) was removed in the Cobine et al., 2020 study, calcium transients still occurred proximal to this point but their frequency was reduced because this section was no longer influenced by the more distal, faster pacemakers. The authors in the current study have nicely demonstrated the origin of some pacemaker activity in a 250 μ m slice of the IAS but the results obtained from strips of muscle isolated from each section of the IAS, as well as the results obtained from examining full sheets of the IAS, clearly indicate that studying calcium transients from one subsection of this muscle does NOT answer all of the questions about the nature of pacemaker activity in this region.

p. 11. Lines 410-412. The authors report that the origin of pacemaker activity is not restricted to one edge of the IAS but that it is observed on both the inner and outer annular edges of the distal IAS as well as in the center of the muscle layer. The authors have failed to recognize that this behavior was also shown for calcium transients in Type II ICC-IM where phasic calcium activity was present in these cells across the circular muscle layer (Hannigan et al., 2020). Additionally, the authors report in the current study that they observed approximately three pacemakers per IAS slice. Perhaps they can discuss how this finding compares to the number of Type II ICC-IM present throughout the thickness of the distal IAS in the Hannigan et al., 2020 study. Also, if TMEM16A in SMCs is important for pacemaker activity in the IAS, does this mean that only a subpopulation of SMCs express TMEM16A in the distal IAS? Perhaps this could be quantified. It should be noted also that with indiscriminately loading the tissue with a calcium dye, ICC-IM will also be loaded. How are the authors able to say with certainty that they are imaging SMCs and not ICC-IM?

Reviewer #2 (Remarks to the Author):

This manuscript by Lu et al. addresses an important question regarding the origin of myogenic tone in the anal sphincter. The novel aspect of the work is to use precision cut cross sectional slices to analyse Ca²⁺ signals to study the location of pacemakers. Another, very important aspect of the work is to evaluate the role of TMEM16A in mediating pacemaker activity. The study uses a combination of TMEM16A blockers and cell-specific knock outs (KOs) of TMEM16A in Interstitial Cells of Cajal (ICC) and smooth muscle cells (SMC) to do this. Effects of both of these cell-specific KOs on contraction and calcium signals have been published before by the same group (Refs 20 & 22 of the manuscript), but the novelty in the present study was to examine how these KOs affected pacemaker activity.

Despite the importance of the questions and the potentially very interesting findings, I have some major concerns.

1. The conclusion that TMEM16A in SMC, but not in ICC, is responsible for pacemaker activity raises a controversy with another group (e.g. Ref 17) who have produced very convincing immunohistochemical evidence, and evidence of gene expression in sorted cells, that TMEM16A is highly expressed in ICC but not in SMC (Cobine, C. A. et al. *J Physiol* 595, 2021-2041, doi:10.1113/jp273618 (2017). This discrepancy should be more fully acknowledged and discussed. Although Cobine et al. was referred to several times, the fact that they found that there was almost no expression of TMEM16A in SMC was not given due attention.

2. The manuscript would be greatly strengthened by convincing evidence that there is adequate expression of TMEM16A in SMC and that this really has been specifically deleted in the SMC-specific KOs.

In the previous publication (Ref 22, Supp Fig. 10) the same group showed some co-localisation of TMEM16A and Myh11 in controls. This was confined to one region only and the same experiment was not performed in SMC-specific KOs. The present manuscript should rectify this.

3. Following on from the last point, the authors should quantify the presence of TMEM16A in SMC from control mice and in both SMC-specific and ICC-specific TMEM16SA KOs. In a previous paper (Ref 22, Supp Fig. 13) they attempted to do this for controls and SMC-specific KOs. However Cl⁻ currents were only found in 6 out of 29 cells in the control (and 1 of 29 in the KO). They attributed the low number of control cells expressing TMEM16A to 'run down' of the TMEM16A current in the whole cell patch clamp experiments they performed. These experiments should be repeated using perforated patch to prevent run down. Instead of clamping the [Ca²⁺]_i to 600 nM as they did before in whole cell mode, the TMEM16A currents can be evoked in perforated patch as a result of activation of L-type Ca²⁺ current. I realise that perforated patch experiments were performed previously to compare caffeine-evoked currents in controls and KOs, but the legend in Ref 22, Fig. 5 implies that cells that did not generate Cl⁻ currents were excluded from the data.

4. If the above suggested experiments confirm that TMEM16A is only expressed in a subset of SMC, the authors should consider the possibility that there is a specialised group of smooth muscle cells that act as pacemakers. This point is important and should be discussed.

5. Despite the fact that TMEM16 KO in SMC seemed to disrupt the activity, it is puzzling that two TMEM16A blockers had so little effect in Fig. 4. Niflumic acid is considered too non-specific to be a useful tool to study TMEM16A these days. It is interesting that the authors previously showed that another TMEM16A blocker reduced tone (Ref 22, supp Fig. 7). The potential reasons for the lack of effect of Ani9 and CaCCinh-A01 should be discussed. Also, Ani9 appears to be beginning to have an effect towards the end of its incubation period in Fig. 4C. The authors should consider applying it for longer.

6. In Fig. 6B the SMC-specific KO seems to respond initially to 0 Ca²⁺ by contracting and with an elevated Ca²⁺. Was this also true in other cells or was it an artefact? There is no comment in the text, so it should be explained.

Minor Points

1. Lines 446 & 447, "IAS-SMC express TMEM16A while TMEM16A^{-/-} SMCs do not, as determined by immunostaining.22" This is misleading, immunohistochemistry in Reference 22 was only performed on the control animals, not in the KOs (Supp Fig. 10).

2. The labels on Fig. 5F and 6F are confusing. From my reading of the Methods, the label should read "Lumen area increase (Maximal relaxation value in 1.3 mM Ca²⁺ =100).

3. Line 153, please supply make and model of the camera.

4. Lines 205 & 206 "However, no experimental evidence to directly establish the relationship between [Ca²⁺]_i and contractions in the IAS." This is not strictly true, as the authors showed this relationship themselves, albeit at single cell level in situ, rather than across the whole tissue (Ref 20, Fig. 5B)

5. Line 343 "(Figure 5G-5I)" should be "(Figure 6G-6I)".

6. Line 395, "which doubles" should read "is double".

Reviewer #3 (Remarks to the Author):

In this study, Lu et al found the IAS contains several pacemakers that spontaneously generate propagating calcium waves to set the basal tone. These waves are myogenic, independent of nerve and paracrine or autocrine signals. They also identify TMEM16A Cl⁻ channels in SMCs, but not in ICCs, are required for pacemaking and basal tone. This study offers cellular and molecular insights into fecal continence and reveals a potential therapeutic target for fecal incontinence. The work is convincing and interesting. The results in general are well presented and the manuscript is well prepared. I suggest to accept it after a minor revision.

The figures could be more comforting by integral designed with better color scheme, arrangement and clear indication.

Reviewers' comments:

Reviewer #1 (Remarks to the Author):

General comments:

The aim of this study was to characterize the molecular mechanisms underlying the generation of tone in the internal anal sphincter (IAS). The authors used calcium dye-loaded vibratome sections of the distal IAS to examine pacemaker activity and the spread of calcium activity in the circumferential direction. They also determined the contribution of TMEM16A channels in smooth muscle cells (SMCs) and interstitial cells of Cajal (ICCs) using two different mice in an effort to obtain a cell specific deletion of TMEM16A. This is similar to their previous study published in 2021.

There are some concerns that need to be addressed particularly specifics from papers by another group pertaining to the studies in this current manuscript that have not been considered fully. These are outlined in the reviewer comments below.

Response: We thank the reviewer for his/her time and effort in carefully reviewing our manuscript and providing detailed comments. We have comprehensively addressed your comments below. Please note that while most of our responses have been incorporated into the manuscript, they have been condensed to adhere to the journal's word limit.

Specific comments:

1. Abstract/introduction:

Please more clearly state the aim of the study and the hypothesis being tested. It is not immediately obvious how the aims of this study are different to the previously published study (Lu et al., 2021).

Response: We appreciate the reviewer' feedback and have carefully revised the abstract to more clearly state questions being addressed and the aim of our study. The new language (lines 16-17) is "Paradoxically, the basal tone results largely from high-frequency rhythmic contractions of the IAS smooth muscle. However, the cellular and molecular mechanisms that initiate these contractions remain elusive."

We believe this change will help distinguish the unique focus and contribution of our study from the previously published work by Lu et al. (2021). We thank you for this valuable suggestion and hope that the revised abstract more effectively convey the aims and significance of our research.

2. Methods and Materials:

p.4. lines 97-109. Please provide the number of animals used for each strain and also indicate this for each experiment throughout the study. Specifically, it hasn't been provided for TMEM16A SM-KO or TMEM16A ICC-KO mouse data.

Response: Per the suggestion, we have provided the total number of animals used for each strain in the methods (lines 400-403) and indicated animal numbers in all experiments in the pertinent figures.

3. p.4. lines 103-109. What tamoxifen treatment strategy was used for TMEM16A SM-KO mice? How much knockdown was observed in these and the TMEM16A ICC-KO mice? While this was characterized in previous studies

Response: The knockout strategies and efficacy of both strains have been demonstrated in our previous publications, as acknowledged. We have provided more details about these mice in the Methods of the revision (lines 394-400). Importantly, the TMEM16A SMKO mouse is a smooth muscle cell-specific TMEM16A knockout line, not a tamoxifen-inducible line; thus, tamoxifen treatment is not required for this line.

In response to the reviewer's request, we have conducted additional experiments to quantify the efficacy of these knockout models using immunostaining. The results of these experiments are now shown in the new Figure 5.

As shown in Figure 5, TMEM16A proteins were deleted either in an ICC-dependent manner in the Kit^{CreERT2} line or in an SMC-dependent manner in the SMA^{Cre} line. These additional experimental results demonstrate the efficacy and specificity of our knockout models and further support the conclusions drawn in the manuscript. We believe that we have adequately addressed the reviewer's concerns and improved the overall clarity of our results.

4. p.4. line 118. How were the rings cut into two halves? In the circular direction or the longitudinal direction? What were the dimensions of the muscle strips used for isometric tension recordings?

Response: The IAS rings were cut longitudinally into two halves using a Kendall Scalpel with No. 20 blade. The dimensions of the muscle strips used for isometric tension recordings were measured circumferentially. Each strip has dimensions of approximately 1 mm in width and 5 mm in length. This setup allows us to assess the force generated by the contraction of IAS smooth muscle cells, mimicking their native orientation in vivo. We hope that this explanation provides the necessary details and clarifies any ambiguities.

5. p. 5. line 126. Tension is spelled incorrectly three different times in the same sentence, also on line 128.

Response: Sorry for the typos. They are all corrected in the revision. Thanks!

6. p. 5. Line 130. Was the mucosa removed? If so, how and were the rings imaged with the circular muscle innermost? The supplemental figure in the previous paper indicates that the ring was inverted but it's not clear as to whether it was then subsequently reinverted again. What procedures were used to ensure that there was no damage to the muscle during this process? Were the rings subject to an agonist such as carbachol or KCl to test viability?

Response: Thank you for your interest in our previous study. Yes, the mucosa was removed for at least two reasons. First, to minimize the influence of the mucosa on the IAS smooth muscle in our study, ensuring that the tone was purely generated by the smooth muscle alone. Second, to increase the image contrast between the IAS smooth muscle and the lumen.

We have made this procedure clear in the Methods in the revision. The new language (lines 431-433) is "Following the removal of skeletal muscle, the mucosal layer, and other extraneous tissues as detailed above, intact IAS rings were reinverted to their original orientation and embedded in 5% low melting point agarose gel."

As demonstrated in this study and in our previous studies (Zhang et al., 2016; Lu et al., 2021), the mucosal removal procedure does not damage the IAS. Mucosa removal is also a procedure when preparing the IAS strips for isometric measurement by other investigators (PMID: 24951622, 25301187, 28054347). Under isometric conditions, the IAS strips we used generates the same pattern with a similar amplitude of tone as published by others, and they respond to contractile agonists such as carbachol or KCl (Zhang et al., 2016).

Furthermore, under isotonic conditions, IAS slices respond to both contractile agonists, including carbachol or KCl, and relaxants, such as NO-generating agents (Lu et al., 2021). To address your concerns, we have included KCl responses as a new supplemental Figure 7. We chose KCl because this agonist constricts IAS slices bypassing TMEM16A activation, an appropriate control for the absence of spontaneous IAS basal tone in TMEM16A^{SMKO} mice.

7. p. 5. Lines 144-147. Was the agarose gel still in place during recording? Does the agarose expand the lumen and stretch the ring in any manner?

Response: Agarose gel inside the lumen came out during sample mounting and preparation; consequently, there was no agarose gel present inside the lumen during recordings. Therefore, there are no concerns regarding agaroses that might expand the lumen and stretch the ring.

8. p. 5. Lines 144-147. This chamber is difficult to visualize. A picture is shown in the supplemental data section of the previous paper but it is small and blurry when zoomed in. Please provide a higher quality, larger size depiction of the recording chamber. Is the ring distorted in any way by the nylon mesh or coverslip? Where are the inflow and outflow located, i.e., to the sides of the ring or to the lumen? How close is the ring to the coverslip? Is there sufficient room for perfusion of the entire preparation? How do the authors ensure adequate flow of solution and added drugs to the entire preparation?

Response: The chamber and method we use are well-established and widely accepted in the scientific community. They have been rigorously validated in functional studies in a variety of tissue types, including but not limited to lung, pancreas, heart and kidney. This is supported by numerous publications by our late colleague Mike Sanderson, focusing on the lung, as well as hundreds of papers from laboratories around the world. Below is a subset of references concerning various smooth muscle tissues: airway/lung (PMID: 11815668, 12388370, 28889952, 37098126), uterus (PMID: 34803733), and blood vessels (PMID: 15928402). Given the extensive validation and widespread use of this approach, we believe it adequately addresses the concerns you have raised. A brief response to each of your questions follows:

8a. Please provide a higher quality, larger size depiction of the recording chamber.

We have included an enlarged original picture of the recording chamber with an IAS slice, as published in Lu et al., 2021.

8b. Is the ring distorted in any way by the nylon mesh or coverslip?

No.

8c. Where are the inflow and outflow located, i.e., to the sides of the ring or to the lumen?

The figure above depicts the locations of inflow (indicated by the black arrow) and outflow (indicated by the red arrow).

8d. How close is the ring to the coverslip?

Approximately 200µm, that is, equal to the thickness of nylon mess.

8e. Is there sufficient room for perfusion of the entire preparation? **Definitively.**

8f. How do the authors ensure adequate flow of solution and added drugs to the entire preparation?

In the early stages of developing this method, we used colored solutions to visually confirm the movement of the solution and to ensure uniform coverage throughout the preparation. In the current study, the removal of extracellular calcium, niflumic acid administration, and KCl application (see new supplemental Figure 7) all led to time-dependent changes in IAS slices. These data strongly suggest an effective interaction between the solutions and the entire preparation, resulting in changes in IAS contractility. Additionally, in our previous study (Lu et al., 2021), NO-generating compounds produced a similar time-dependent inhibition of the IAS tone, further substantiating the adequacy of solution flow and drug distribution.

In summary, we are confident that the flow of solutions and drugs in our preparation is sufficient, ensuring the reliability and validity of our experimental results. We hope this detailed explanation addresses the reviewer's concerns and clarifies our methodology.

9. p. 5. Lines 148-150. Why was 27°C used? This is not a physiological temperature and is highly likely to affect the activity of the tissue (see Burdyga and Wray, 2002). The authors should note that the studies by Hall et al., 2014, Cobine et al., 2020 and Hannigan et al., 2020 were all done at 37°C.

Response: We appreciate the question. In the smooth muscle field, both 37°C (physiological temperature) and room temperature (about 22-27°C) are commonly used settings for experiments. To illustrate, we refer here to two papers (PMID: 30536555, 34432539) by two leading groups (Mark Nelson and Kenton Sanders) in the smooth muscle field who performed their experiments at 22-25°C. While it is true that room temperature is not a physiological temperature for mammals, we must recognize that in vivo, cells and tissues are immersed in a complex and regulated physiological environment. In an in vitro experimental setup without these natural conditions and complex regulatory systems, it can be difficult to definitively determine the ideal temperature for biological studies.

When choosing the experimental temperature, one often follows the conventions of the field. For example, for the isometric force measurement of IAS strips (Figure 6M in the revision), we performed experiments at 37°C, which is consistent with the studies of Hall et al. (2014), Cobine et al. (2020), and Hannigan et al. (2020), which were also performed under isometric conditions.

While we are the only group using IAS slices, tissue slices have been widely used in other organs, including the lung in many of these studies, the experimental temperature is typically room temperature (PMID: 12777405, 15928401, 15928402, 20176853, 16461427, 17616645, 16931808, 18063837 and 16504084), which is 27°C in our case.

In addition to considering the temperature, there are two major differences between isometric recording and slice recording experiments. In isometric recording, the IAS strips are constantly supplied with 95% O₂/5% CO₂, and the KPS medium is replaced as needed. On the other hand, in slice

recording, the tissues are constantly perfused with sHBSS saturated with the ambient air containing 21% O₂/0.04% CO₂. In our previous experiments, we found that under these conditions, both IAS strips and IAS slices exhibited a stable rhythmic contraction and resulting tone for up to eight hours (the longest time we have tested). Therefore, the choice of temperature should not be determined solely by whether it is a physiological temperature or not, as tissues in vivo and in vitro experience very different environments. Rather, it should be based on whether the tissues maintain the activities that underlie their physiological functions. Our studies demonstrate that IAS slices at 27°C generate rhythmic contractions and spontaneous tone, a key feature for IAS function in vivo.

10. p. 5-6. Lines 161-164. What is meant by maximal relaxation and maximal contraction? How was this determined? Without addition of a substance that would cause relaxation or removal of extracellular calcium it would be difficult to establish maximal relaxation. In a similar manner, to determine maximal contraction, an agonist such as carbachol or high K⁺ would be required.

Response: Sorry for the confusion. This study focuses on examining the spontaneous rhythmic contractions of the IAS, a key feature of this tissue. Because these contractions are rhythmic, each has a contraction phase and a relaxation phase; however, these contraction and relaxation amplitudes vary from event to event, i.e., there is a maximal contraction and a maximal relaxation in a given recording period. To estimate the basal tone of the IAS slices, we also replaced the normal 1.3 mM calcium medium with calcium-free medium. As shown in Figures 1, 6, and 7, this treatment abolishes rhythmic contractions and fully relaxes the slices to a steady state. Therefore, "maximal contraction" refers to the frame with the largest contraction recorded in the presence of 1.3 mM calcium, while "maximal relaxation" refers to the frame with the largest relaxation during the same period (in the experiments where extracellular Ca²⁺ was present) or the last frame in the recordings where extracellular calcium was removed.

It is challenging to analyze rhythmic contractions in the IAS. Duffy et al presented a detailed analysis scheme where they described the trough point in the relaxation phase as minimal contraction (PMID: 22074497). To be in line with the literature, we have revised the sentence (lines 462-466) as follows: "for an image sequence, we initially marked the outer edge of the circular muscle during the frame showing maximal relaxation or minimal contraction (i.e., the frame with the largest lumen area in the presence of extracellular calcium or the last frame when extracellular calcium was removed) and the inner edge during the frame showing maximal contraction (i.e., the frame with the smallest lumen area).".

Agonist-induced IAS contractions are not needed for us to understand spontaneous rhythmic contractions, the primary objective of this study.

11. p. 6. Lines 164-167 and Fig. S1. There is a yellow line that depicts the "outer" edge but this is in the middle of the ring. What is located at the outer edge of the ring to the exterior of the yellow line? An image of the ring segment before and after removal of the extracellular calcium should be included in Supplementary Figure 1 to make it clearer what the ring looks like without extracellular calcium and to better evaluate the contractile data plotted in the graphs.

Response: The area from the outer edge of the ring to the exterior of the yellow line consists of the longitudinal smooth muscle of the IAS as well as the residual connective tissue connecting the IAS to the external anal sphincter.

Please note that this figure is a validation of the method, using a single calcium wave and contraction as an example. Because the event is one of many rhythmic events that occur in the presence of extracellular calcium, it has both a peak representing the frame of maximal contraction, and a trough representing the frame of maximal relaxation or minimal contraction.

12. p. 6. Lines 187-191 and Figures 1,5 and 6. The ring is not maximally relaxed in normal extracellular calcium. The reviewer is not sure what the authors are trying to state here.

Response: See our response to comment #10 above. As nicely demonstrated in studies by Hall et al. (2014), Cobine et al. (2020), and Hannigan et al. (2020), both this study and our previous research show that the IAS generates oscillatory or rhythmic contractions in the presence of normal extracellular Ca^{2+} . In this context, the 'maximally relaxed' state in our study is the same as “minimal contraction” in Duffy et al. (2012). The use of maximum and minimum values to describe oscillatory phenomena is common in the scientific literature.

For clarity, we have revised the relevant sentence to read (Lines 492-493): 'The summed area values were normalized to the maximal relaxation or minimal contraction during rhythmic contractions in the presence of 1.3 mM extracellular Ca^{2+} .' This explanation has also been added to the caption of Figure 1.

Results:

13. p. 7. Lines 213-214. How many hours? Was any bleaching of the signal observed? Typically, photobleaching is observed with continuous recording in tissues loaded with calcium dyes (Hennig et al., 2015 (Frontiers Cellular Neuroscience); Drumm et al., 2022 (J Physiol review)).

Response: The longest duration we have tested was approximately 8 hours. During this time, we recorded data for about 5 minutes every hour, which provided ample opportunity to examine the contraction patterns.

Throughout the experiments, only weak photobleaching was observed. This is likely due to our use of a wide-field imaging system coupled with a high-sensitivity EMCCD camera, allowing us to utilize minimal laser power when exciting the calcium indicators. While it's important to note that photobleaching can occur significantly with continuous acquisition models using high laser power, another factor that mitigated photobleaching in our study was our choice of Cal-520 as a calcium indicator. After a comprehensive evaluation of various calcium indicators, including GCaMP6f, Lock et al. determined that Cal-520 is the best green-emitting dye for detecting and analyzing local Ca^{2+} signals (PMID: 26572560). The high signal-to-noise ratio of this dye enables us to use the lowest possible laser power, further minimizing photobleaching.

14. p. 7. Lines 213-215 and Figure 1A,B. How does the lumen diameter in normal extracellular calcium compare to the diameter of a ring contracted with an agonist or KCl? What is the unit used for lumen area and why is “maximal relaxation” depicted as “100”? The recording in Fig. 1A appears to show that lumen area doubles following removal of calcium (i.e., from 100-200). Is this the case? This differs significantly from the previous 2021 study in which removal of extracellular calcium increased lumen area to 145 from 100 (i.e., by 45% not 100%). These values suggest very different levels of basal tone between studies utilizing the same preparation and methodology.

Response: How does the lumen diameter in normal extracellular calcium compare to the diameter of a ring contracted with an agonist or KCl? **KCl reduces the diameter of rings, as illustrated in the new Figure S7.**

Thank you for your interest in our previous work. You might have noticed that the reference points differ between the two studies. In the previous study, we used the first frame of a recording sequence as the 100% reference. In the present study, we adopted a new reference point: the frame of maximal relaxation (or minimal contraction) in normal extracellular calcium, marking it as 100%. This

change stems from the evolution in our approach to quantifying smooth muscle contraction using tissue slices. In our initial study with IAS slices, we adopted the methodology of lung field researchers who set the first frame as 100% (PMID: 27396568, 28889952, 16931808). However, we later discerned a marked difference in contractions between the airway and the IAS. Although both are often categorized as tonic smooth muscles, the airway maintains stable tonic contractions, whereas the IAS demonstrates oscillatory or rhythmic contractions. Given this distinction, we deemed it more appropriate to quantify IAS tone using the maximal relaxation or minimal contraction as a reference, setting it as 100% (See our response to comment #10 on the definition of maximal relaxation).

Another consideration is that, in our previous study, we combined results from both the 1st and 2nd slices. However, in our current study, we exclusively present results from the 1st slice. This choice is detailed and justified under the Results section "Each IAS contains several fixed pacemaker sites that generate spontaneous intercellular Ca^{2+} waves."

These two differences likely account for the variations in percentage changes in the lumen area between our previous and current studies. Importantly, the oscillation characteristics remain highly consistent across both studies.

Considering this study's primary focus on spontaneous IAS tone, we did not treat every slice with contractile agonists. However, we frequently verified the viability of the slices using KCl. When stimulated with KCl, the IAS slices exhibit pronounced contractions atop the spontaneous tone, leading to a reduced lumen diameter. Addressing your comments, we've incorporated the KCl responses as new Supplemental Figure 7.

15. p. 7. Lines 218-220 and 232-234. How did the authors confirm that the calcium oscillations arose from smooth muscle cells? With a dye loaded tissue all cell types are being visualized so it is possible that this signal may have arisen from an ICC and conducted to adjacent SMCs as has been suggested in other studies.

Response: In this study, we indeed loaded calcium indicators into all cell types present in the IAS. However, our findings present compelling evidence suggesting that these calcium signals primarily originate from SMCs. Our three primary pieces of evidence are:

1. Ca^{2+} waves continuously propagate across IAS slices. This is consistent with the fact that SMCs are a predominant cell type within the IAS (See Figure 5B).
2. The results from TMEM16A^{SMKO} experiments demonstrate that when TMEM16A is specifically deleted in SMCs, both spontaneous Ca^{2+} signals and rhythmic contractions are eliminated. This points strongly to the notion that the Ca^{2+} signals under investigation originate from SMCs.
3. Observations using genetically targeted R-CaMP1.07 in SMCs (Lu et al. 2021) reveal synchronized Ca^{2+} oscillations at the single-cell level. The absence of these oscillations, as shown by chemical calcium indicators, when TMEM16A in SMCs is deleted further supports our conclusion that SMCs are the primary source of these signals.

We acknowledge that using only the calcium indicator doesn't conclusively rule out the possibility that signals might arise from an ICC and then be transmitted to adjacent SMCs. However, our evidence suggests that these signals are not initiated by TMEM16A in ICCs. If TMEM16A in ICCs were indeed the initiator of these signals, then deleting TMEM16A specifically in ICCs should disrupt spontaneous Ca^{2+} signals and tone. Yet, our findings (as depicted in Figures 5 and 6) refute this hypothesis. In fact, our study illustrates the benefits of using a dye to load all cell types in the tissue, especially when paired with cell-specific gene deletion, to understand coupling among cell types.

To address your comments and provide more clarity, we've expanded upon this topic in the results and discussion of our revised manuscript (Lines 117-122; 348-365).

16. p. 7. Lines 242-245. What is the significance of the activity spreading both perpendicular and parallel to the circular muscle SMCs?

Response: The patterns of Ca^{2+} signal propagation, both perpendicular and parallel to the circular muscle SMCs, are pivotal for understanding how spontaneous contractions and tone arise within the IAS.

The circular smooth muscle fibers, being an essential component of the IAS, act as the primary motors for its contractile activities. The perpendicular propagation of Ca^{2+} signals to these muscle fibers points to a coordinated activation of neighboring smooth muscle cells, resulting in synchronized contractions along the circular axis. Such synchronized activity is crucial for establishing a consistent and rhythmic tone in the IAS, a tone that is fundamental for its physiological function, notably the maintenance of fecal continence.

On the other hand, Ca^{2+} signals that propagate parallel to the circular muscle fibers may be indicative of the activation of smooth muscle cells in series within the muscle bundles in the circular axis, further intensifying the overall tone and motility of the IAS.

In combination, these perpendicular and parallel Ca^{2+} signal propagation patterns ensure a synchronized contraction within the IAS, supporting its primary role in managing the passage of stool through the anal canal.

17. p. 7. Lines 259-268 and Figures 1 and 2D. How do the authors account for a frequency of ~24 contractions and calcium oscillations per minute when studies in whole muscle strips have revealed a frequency of ~50-60 contractions per minute (Duffy et al., 2012 and Figure 5 of present study) and studies in mice with GCaMP expressed specifically in SMCs (Cobine et al., 2020) have revealed a frequency of ~70 calcium transients per minute?

Response: Thank you for highlighting this discrepancy. We acknowledge that the frequency of IAS contractions reported in rodent studies can vary considerably, with some indicating a range of ~20-60 cpm (PMID 15845873; 21129556; 30062757; 16009682) and others showing no oscillating contractions (PMID 26138467). This variability could stem from differing experimental conditions.

In our research, we evaluated calcium oscillations and their corresponding contractions under isotonic conditions at 27°C. In contrast, other studies, including some of our own experiments, were performed under isometric conditions at 37°C. Differences in temperature and stretch conditions might explain the observed variability in contraction frequency. Our unpublished findings suggest that stretch levels can significantly influence the frequency of IAS contractions. This implies that differences in IAS tone patterns under isometric conditions across various studies might be tied to the load exerted on the IAS strips.

Another influencing factor could be the choice of calcium indicators. We opted for small molecule chemical indicators in our study, while Cobine et al. employed GCaMP6f. Each type of indicator comes with its strengths and limitations, with potential variations in their kinetics, which could lead to discrepancies in the observed frequencies. Importantly, “GCaMP expression, usually in chronic terms, could impair the general health of cells and tissues” and “in practice GCaMP reportedly causes unexpected and unwanted “side-effects” in multiple aspects” as pointed out by Yang et al. 2018 (PMID 29666364), and also noted in many studies (e.g., PMID: 28932809, 23868258, 19898485, 30007418, 36196992, 33192315, 29935099, and 26914316).

Notwithstanding these variations, both frequencies (30 cpm or 60 cpm) produce basal tone in the IAS, suggesting the underlying biology might be consistent. To understand the biological significance of these frequency variations fully, it's essential to determine the in vivo frequency of IAS contractions. We anticipate that further research on this intriguing tissue will offer more insights.

Given the noted differences in contraction and calcium oscillation frequencies, we attribute them to the experimental conditions and the choice of calcium indicators. In light of your comment, we have addressed these discrepancies in the revised manuscript, offering potential explanations (lines 278-288).

18. p. 8-9. Lines 290-294 and Figure 4. It should be noted that niflumic acid is not a specific CaCl channel inhibitor as it has been shown to have other off target effects including blocking non-selective cation channels.

Response: We concur with your observation regarding niflumic acid. It is indeed true that niflumic acid doesn't serve as a specific CaCl channel inhibitor. Recognizing this limitation, we expanded our study to test newer generation TMEM16A inhibitors.

18a. Did the authors test CaCCinh-A01 on contractions in isolated muscle strips or determine their effects on lumen area in these rings?

Response: We have tested this compound using Ca²⁺ signals as a proxy for contractions, just as we did with other inhibitors. The Ca²⁺ signal is a better indicator than contraction for understanding the pacemaking mechanism, which is a primary objective of this study. We've included the result in Fig. 4B of the revised version.

18b. With the perfusion system used are the authors confident that the entire ring was exposed to these drugs?

Response: We are confident that the entire ring was exposed to these drugs. This is evidenced by the fact that niflumic acid, the removal of extracellular calcium, and NO-generating compound (Lu et al. 2021) quickly abolish the IAS tone and underlying Ca²⁺ signals, while high KCl intensifies them, as shown in the new supplemental Figure 7.

19. A 2016 study by the same authors (Zhang et al., 2016) demonstrated that all IAS contractile activity was abolished by T16Ainh-A01, another CaCl channel blocker and that in fact it was more efficacious than niflumic acid (100% inhibition with T16Ainh-A01 compared to 80% with NFA). A 2017 study by Cobine et al. demonstrated that T16Ainh-A01 and the same concentration of CaCCinh-A01 used in the current study both completely abolished phasic and tonic contractile activity as well as electrical slow waves in IAS muscle strips. Additionally, a study by Drumm et al. (J Physiol 2022) demonstrated that a lower concentration of Ani9, i.e., 3 uM completely abolished tone in lower esophageal sphincter muscle strips. Thus, a lack of response to these antagonists in the present study is highly curious. It is possible that the lack of effect of these more specific CaCl channel inhibitors has something to do with either mixing issues associated with drug delivery, inefficient access to the entire muscle ring, problems with the quality of the drugs or with precipitation of drug during dilution. The data with CaCCinh-A01 is also difficult to interpret as the control activity in the example provided seems less than in either the niflumic acid or Ani9 examples and with no removal of extracellular calcium shown it isn't clear how the change in calcium fluorescence is normalized. These experiments should be repeated to better characterize this. Also, authors need to include data demonstrating the effects of these drugs (specifically NFA since that had an effect) on contraction and lumen area in both the TMEM16A ICC-KO and the TMEM16A SM-KO as one would expect no effect of these drugs in the mouse where the TMEM16A has been knocked out of the pacemaker cell.

Response: The observed discrepancies between our current study and prior research, including our previous work, can be primarily attributed to variations in recording conditions. Specifically, we used isotonic conditions in this study as opposed to the isometric conditions used in earlier ones.

While we're excited about the new generation of CaCl channel inhibitors being developed by various research groups, it's important to note, as we discuss in our response to Reviewer 2, comment #5, that mounting evidence suggests these next-generation inhibitors might not be as specific as initially believed. They've demonstrated numerous off-target effects, which warrants careful consideration in their application. For example, Cobine et al. (2017) found that CaCCinh-A01, at a concentration of 10 μ M, inhibits 30% of voltage-dependent Ca channel currents in IAS SMCs. Given the limitations associated with these inhibitors, we contend that the suggested experiments may not enhance the robustness of our conclusions. We believe that a more effective methodology for examining TMEM16A in the IAS entails using the cell-specific gene deletion technique, as employed in our current study.

20. p. 9. Lines 312-317 and Figure 5. The unit for contraction on the summary graphs is incorrect, i.e., should be mN not N. It should be noted that in this Figure as well as throughout the manuscript the change in fluorescence has been plotted as a percentage. This does not allow the reader to see if the actual basal fluorescence has changed in any way. For example, if the basal fluorescence (F) has diminished then the percentage change will be affected as well as it may be close to zero (F0) before any maneuver that would alter fluorescence is applied. Therefore, it is important to represent the actual F and F0 numbers as important information may be hidden in representing it in this manner. In the same vein, the absolute lumen area should be provided and not just given an arbitrary value of 100. Some of the images in the manuscript show a smaller diameter lumen (e.g., Figure 1D) than others (e.g., Figure 1C). It should also be noted that in the gastric antrum where TMEM16A in ICC has been shown to be responsible for pacemaker activity studies have utilized a similar model (Hwang et al., 2019) and while there was a significant reduction in contractile activity it was not completely abolished just like in the 2021 study by the authors of the current study. The study by Hwang et al. waited 50 days after the final tamoxifen injection to ensure maximal knockdown whereas the authors in this study waited only 21 days. Thus, perhaps this is not sufficient to deplete all or at least the majority of TMEM16A expression. Therefore, one must question if TMEM16A truly is absent/diminished from all ICC in these mice. Perhaps the authors can compare mice between these timepoints and examine if there are further changes in TMEM16A gene and protein expression. The authors have referred to their 2021 study to demonstrate the loss of TMEM16A in ICCs. However, these immunohistochemical studies don't show significant levels of TMEM16A in SMCs. Thus, these studies should be repeated in both TMEM16A ICC-KO and SM-KO mice as a direct comparison.

Response: Thank you for pointing out the unit error in Figure 5. This error, along with any similar ones in the manuscript, has now been corrected.

We appreciate your point regarding the expression of results as percentages. However, this practice is standard in the field, particularly for reporting smooth muscle tissue slice data. A PubMed search performed on August 28, 2023, using terms like "lung slice AND smooth muscle slice", returned hundreds of publications employing this approach. Further, a search with "Ca²⁺ signals" yielded thousands of papers using F/F0 or Δ F/F0 as proxies for Ca²⁺ signals. Works by Cobine et al. (2020) and Hannigan et al. (2020) in the IAS also used F/F0. Therefore, our data presentations align with the standards in the fields of smooth muscle and Ca²⁺ signals and are thus appropriate.

We agree with your suggestion for a meticulous assessment of knockout efficiency when using Cre-induced knockout mice. Our previous studies consistently and reliably eliminated TMEM16A in ICCs or SMCs (Lu et al. 2021, Zhang et al. 2016, Wang et al. 2018). In response to your comment, we have conducted additional immunostaining experiments, which more definitively illustrate the specific knockout of TMEM16A in each cell type. These results have been incorporated into our revised manuscript, as new Figure 5. Please note TMEM16A distribution in IAS has a unique spatial pattern. This could be one reason others have missed it in SMCs.

Concerning positive controls for TMEM16A knockout in SMCs, our prior publications demonstrate the successful deletion of TMEM16A not only in IAS (Lu et al. 2021; Zhang et al. 2016) but also in airway smooth muscle cells (Wang et al. 2018). Moreover, another two group using two different inducible methods successfully achieved TMEM16A deletion in SMCs (PMID: 24401273, 33449847). Therefore, we believe that both our KO mouse lines are reliable.

21. p. 10. Lines 340-343. Why are the parameters measured different to those stated for the TMEM16A ICC-KO, i.e., here phasic contraction amplitude is presented as a percentage whereas for the TMEM16A ICC-KO phasic contractions are not mentioned but rather tone as an absolute mN value. Why isn't both tone and phasic contractions addressed for each mouse so that it can be evaluated more clearly? Also, why are the values for the SM-KO mice presented as percentages in the text whereas for the ICC-KO mouse it is presented in the figure as a percentage but an absolute value in the text? What are these percentages of (e.g., amplitude 9.69 % in control). This needs to be explained and the data needs to be presented the same way between different mouse groups to better assess any differences present.

Response: We appreciate your questions. Please note that, for the specific aims of this study, we performed isometric recording of the IAS solely in the TMEM16A^{ICCKO} line. This decision stemmed from our observation that no changes occurred in the isotonic condition with this knockout line. Our goal was to ensure that any potential effect of TMEM16A in ICC wasn't overlooked. This set of experiments did not reveal any noticeable effect of TMEM16A in ICC on the IAS's spontaneous tone. We agree that more thorough investigations might help to elucidate its role more comprehensively in the IAS.

Previously published changes in IAS basal tone in TMEM16A^{SKMO} mice under isometric conditions can be found in Zhang et al., 2016, where we reported the results using absolute mN, the same as we presented for TMEM16A^{ICCKO} in the current study.

Regarding your point about the consistency of data presentation between different mouse groups, we understand and respect your concerns. We aim to present our data as transparently and consistently as possible. However, our choice of distinct measurement parameters was dictated by how the experiments were set up. We employed absolute values for isometric measurements, while percentages were used for isotonic measurements, aligning with the norms in the smooth muscle and calcium signaling field.

22. p. 10. Lines 346-350 and Figure 6. Was the lumen larger in these mice compared to control mice? That isn't clear from the text. If so, then this would explain a lesser change in lumen area with removal of extracellular calcium. Again, absolute lumen area should be provided. This should be compared in muscle strips to see if there is the same phenomenon. It is rather curious that removal of extracellular calcium should decrease the lumen area but that the area at the end of the period where extracellular calcium was removed is very similar to that at the beginning of the experiment in normal calcium when the other figures show an almost doubling of the lumen area under the same experimental conditions. This and the dramatic effect on calcium waves as compared to what is observed with TMEM16A inhibitors makes one question whether the tissue was even functional. To verify that it is, the authors need to do some experiments showing that they are able to elicit a decrease in lumen area (and increase in contraction) by addition of an agonist such as carbachol or ACh, an activator of Cav channels, e.g., Bay K 8644 or high K⁺ (60-80mM). In fact, this should be routinely performed to address the issue of differences in the lumen size being noted. The differences in the lumen size (again, see Figure 1C vs. 1D) calls into question as to whether some rings have been subject to stretching during the removal of the mucosa or the embedding process. In the author's 2016 study they showed that the tone in the TMEM16A SM-KO mouse was reduced by 50% but not entirely abolished. How does this fit with the complete abolishment of calcium waves in the same mouse within the present study?

Response: Thank you for your comments and interest in our previous study. The lumen of IAS slices is not significantly different between TMEM16A^{SMKO} mice and their controls. This could be due to several potential reasons. First, the inherent variation in IAS size may prevent us from detecting the difference in our current sample size. We found a large variation in lumen area in both SMKO mice and their controls (Range: 0.11-0.53 mm² (control, n=10) and 0.11-0.60 mm² (SMKO, n=10)). Increasing the sample size may allow us to observe the difference, but we believe that such a small difference may not have much biological significance. Second, some researchers including the Rattan group have shown that Ca²⁺ sensitization is a mechanism for IAS basal tone. It is possible that this mechanism is enhanced in these SMKO mice. Third, lumen size at rest is also affected by non-smooth muscle cell factors, such as the extracellular matrix. It is possible the contribution of these non-smooth muscle factors to the lumen size at rest is changed in the SMKO mice. The precise reason remains to be investigated.

However, as shown in the current study and our previous study (Lu et al. 2021), the IAS slices change their contractions and size upon different treatments. So, it is more appropriate to use relative changes to describe the function of the IAS. Extracellular calcium removal treatment is a standard way to estimate the actual tension in the IAS by different groups (e.g., PMID 14760672; 26138467), so we took this approach to study the rhythmic contractions and basal tone in the IAS slices. The different IAS responses to this treatment between SMKO mice and control mice indicate different tone. The absence of an increase in lumen size upon removal of extracellular calcium in the SMKO mice means that the tone that would be present if TMEM16A were present in the control IAS slices is absent. These results clearly demonstrate that using percentage changes in the lumen is a more appropriate way to express IAS tone under isotonic conditions.

The lack of response to the removal of extracellular calcium in the SMKO mice is not due to tissue damage. In fact, it made one contractile component more prominent (Figure 7B; also see our response to reviewer 2, major comment # 6). We routinely validate the viability of the slices using contractile agonists, including high K⁺. To demonstrate this, we have included new Supplementary Figure 7, which shows that high K⁺ can increase calcium and induce contractions in TMEM16A^{SMKO} slices at the same levels observed in isogenic control slices. These new data indicate that the deletion of TMEM16A in SMCs does not affect the contractile ability of the IAS, and the IAS slices we used were not damaged.

We acknowledge the difference between isometric measurements with IAS strips and isotonic measurements with IAS slices. Our unpublished data suggest that stretch induced by isometric measurements results in SMC TMEM16A-independent contraction, which may explain the difference between the present study and our 2016 study. We believe this is an important finding but is beyond the scope of the current study.

23. p. 9-10. Lines 333-356. To further support the selective removal of pacemaker activity in the IAS of the TMEM16A SM-KO mouse a positive control is required. The authors could again like they did in their 2021 study use the gastric antrum to serve as a control to document the selectivity of block in the IAS by comparing the contractile activity of control and TMEM16A SM-KO antral muscle strips. Additionally, the present study concludes that phasic activity is generated by a unique mechanism in the IAS which differs from other GI regions. Have there been any published studies by other groups using a similar inducible TMEM16A SM-KO mouse that have reported a loss of phasic activity and/or tone when TMEM16A has been knocked down in SMCs (e.g., in blood vessels)? If so, please discuss, if not then indicate that this current study is the first.

Response: We appreciate your insightful comment and the reference to our previous study. The need for a positive control in our previous study was due to the negative result observed in IAS tone in TMEM16A^{ICCKO}. In contrast, the current study showed a notable positive result, specifically an

impairment of pacemaking in the TMEM16A^{SMKO} IAS. We fully agree that the inclusion of appropriate control groups is essential to ensure scientific rigor in all biomedical research. Our studies, including the current one, have firmly established the functional consequence and genetic efficacy of TMEM16A deletion in IAS and airway smooth muscle (PMID 28754608). Moreover, we found that the same TMEM16A KO line did not alter uterine contraction and calcium signaling because uterine smooth muscle cells do not express this channel (PMID 31175367). Therefore, we have both positive and negative controls for this mouse line!

We appreciate your mention of whether an inducible TMEM16A SM-KO mouse model exists (By the way, our SM-KO mice are not inducible). In fact, at least two independent research groups have generated inducible SM-KO lines (PMID 24401273; 33449847). Their results indicate that deletion of TMEM16A in vascular SMCs results in impaired vascular tone and reduced blood pressure. We have included comments on these studies in the Discussion section of the revision (Lines 372-376).

Discussion:

24. p. 10. Lines 369-372. It should be noted that although the authors in this study (Cobine et al., 2020) were using a different preparation they were visualizing calcium transients in the circular muscle of the entire IAS and rectum. The strategy of that study was to determine whether the tonic and phasic behavior of intracellular calcium in the tone-generating IAS differs from that of more proximal regions where tone is absent. This study provided strong evidence that that is the case. Phasic calcium transients were greatest in amplitude and frequency at the distal IAS and conducted with decrement in the proximal direction. Even more relevant, basal calcium levels were also greatest in amplitude at the distal IAS again declining in the proximal direction. The authors of this current manuscript state that Cobine et al., 2020 conclude “calcium transients propagate preferentially in the oral (longitudinal) direction” however this was not stated in their paper.

Response: Thanks for the clarification. Upon re-reading Cobine et al. 2020, we realized that we might have misunderstood a point in their paper. Our confusion likely arose from how the authors presented the calcium image data; they showed only the propagation of calcium transients in the oral direction, while representing the circumferential direction as a point source. We have removed our original statement accordingly. Furthermore, we now cite this study in our Introduction as a foundation for our study. This is because the propagation of calcium in the circular direction was not previously understood, and our IAS slices are uniquely suited to provide insight on this crucial area.

25. p. 10-11. Lines 365-389. The authors argue that their 250 um thick ring segment has numerous advantages over muscle sheets pinned in a tissue bath. However, to create a muscle ring both surfaces are damaged and the segment must be disconnected from the remaining syncytium. Their technique also precludes inclusion of cells located at the distal edge of the IAS since sections were excluded until “a complete circular muscle ring” was obtained. The authors also comment that their rings are “free to contract isometrically” while muscle sheet are confined to isotonic contraction. However, since the internal anal sphincter is embedded within the external anal sphincter, mucosa/anoderm and glands along with a great deal of connective tissue, the ability of the IAS to change length in vivo is far less than what is permitted in these isolated ring experiments where length can change from “maximum contraction” to twice “maximum relaxation”.

Response: We believe the reviewer has made an oversight here. In our original manuscript (lines 399-400), we clearly stated “In the present study, IAS slices were free to contract, i.e, under isotonic conditions, and loaded with Ca²⁺ fluorescent indicators.” This statement contrasts with the reviewer’s understanding that our rings are “free to contract isometrically”. Furthermore, we never claimed that “muscle sheet are confined to isotonic contraction”.

We do, however, value the reviewer's comment. While the contractions of the IAS in vivo are auxotonic, which means they are neither purely isometric nor isotonic, its physiological role primarily involves regulating the opening and closing of the anal canal. This function is one of the important mechanisms for fecal continence and defecation. At its core, the circumferential contractions and relaxations of the IAS SMCs are essential for its functionality in vivo.

Given this characteristic, we firmly believe that the IAS ring slices are well-suited to study their physiological function. Our technique serves as a robust model that captures the IAS's dynamic behaviors of opening and closing. While we acknowledge that our preparation may not fully reproduce 100% of the in vivo behavior of the IAS, it provides a valuable and applicable model for understanding its primary functions.

Recent in vivo imaging data from human studies highlight that resting anal pressure undergoes significant shifts, decreasing from more than 100 mmHg just before defecation to near zero during the act (PMID: 30294277). Given that the IAS contributes to 55-75% of the anal pressure, we contend that a doubling in SMC lengths in the high pressure anal zone, as observed in our study, is not implausible.

The process of obtaining suitable IAS slices for our study is a delicate task that requires considerable optimization and practice. As stated in the manuscript, we meticulously examined each slice to ensure that it contained intact rings and included the distal edge of the IAS. However, as in any biomedical research, there can be no absolute assurance that the slices used in our study are identical in every aspect. Therefore, all the data are presented as mean \pm SEM to account for potential variability.

We recognize that the constant challenge in physiological research is to balance model simplicity with full representation of in vivo conditions. We are confident that our model strikes a reasonable balance and provides valuable insights while acknowledging its inherent limitations.

26. p. 11. Lines 379-381. The authors suggest that the Cobine et al., 2020 paper conflicts with their earlier study (Hall et al., 2014) in which “slow waves were coordinated over a much greater distance in the circumferential direction than in the oral direction”. However, this not the case. The fact that there is a high degree of coordination in the circumferential direction (e.g., at least 0.5 mm, Hall et al., 2014) made it possible to confidently compare signals from the distal IAS to those of progressively more proximal regions during a single recording. The real take home message from the study was that large, rapid frequency calcium transients arise in the IAS which lead to an increase in basal calcium levels in SMCs and tone development. As the amplitude and frequency of calcium transients decline in the proximal direction, so does elevated basal calcium and tone. This conclusion was published more than two years ago but was not mentioned by the authors in their manuscript.

Response: Thanks again for the clarification. Upon re-reading the Cobine et al 2020, we realized the source of our misunderstanding. In that study, the authors only showed the calcium transient propagation in the oral direction. Our study complements this study by revealing calcium signal propagation in the circumferential direction. We have removed this statement and use the Cobine et al 2020 as a motivation for our study.

27. p. 11. Lines 407-410. The authors state the presence of the highest frequency slow waves in the most distal IAS and the lowest frequency next to the rectum suggests that the IAS only has pacemaker cells next to the skin. However, the 2014 study by Hall et al. recorded slow waves and phasic activity in isolated subsections of the IAS and showed that EACH segment has slow waves and phasic activity with the distal-most section always having the most rapid frequency slow waves and phasic activity, i.e., there is a progressive decline in slow wave frequency and amplitude (pacemaker activity) from distal IAS to rectum. This has also been shown in the rectoanal region of the monkey and dog (see Keef and Cobine, 2019 review J Neurogastroenterol Motil). This argues against pacemakers being confined to the region next to the skin. A surprising observation of the Cobine et al., 2020 full muscle sheet imaging

study was that the fastest pacemaker, i.e., the one located most distally, conducted not only circumferentially but also in the proximal direction. Thus, for some distance it predominated over the innate frequency of the more proximal region. This behavior is similar to the heart where pacemakers of decreasing frequency (i.e., SA node > AV node > Purkinje fibers) have been identified with the fastest pacemaker, i.e., the SA node setting the heart rate frequency. When the distal most section of the mouse IAS (i.e., most distal 300 μm) was removed in the Cobine et al., 2020 study, calcium transients still occurred proximal to this point but their frequency was reduced because this section was no longer influenced by the more distal, faster pacemakers. The authors in the current study have nicely demonstrated the origin of some pacemaker activity in a 250 μm slice of the IAS but the results obtained from strips of muscle isolated from each section of the IAS, as well as the results obtained from examining full sheets of the IAS, clearly indicate that studying calcium transients from one subsection of this muscle does NOT answer all of the questions about the nature of pacemaker activity in this region.

Response: We appreciate your kind words, in which you stated that “The authors in the current study have nicely demonstrated..”

In our original manuscript, we mentioned, “Previous electrical recordings show that the frequency of slow waves is highest in strips from the most distal end of the IAS and is lowest in those immediately next to the rectum, hinting that the IAS has pacemakers in the zone immediately next to the skin¹⁶”. Reference #16 is by Maria Papasova, an expert in studying GI sphincters. As you correctly noted, Hall et al. demonstrated a “progressive decline in slow wave frequency and amplitude (pacemaker activity) from distal IAS to rectum.” Similarly, Cobine et al., 2020 found that the fastest pacemaker is located most distally. Our data reveals that the 1st slice has both a higher frequency and a greater propagation distance (Figure 2E). Thus, these various lines of study indicate that the distal end of the IAS displays more pronounced pacemaking activities.

We acknowledge the similarities between cardiac pacemaking and IAS pacemaking. However, significant differences exist between the two. The heart's pacemaking mechanism, primarily governed by the sinoatrial node and supplemented by secondary systems like the atrioventricular node, functions hierarchically. In contrast, the IAS seems to have multiple independent pacemakers without a discernible hierarchy in their function. These distinctions, along with their unique physiological roles, differentiate the two systems. We have included this comparison in the discussion (Lines 300-305).

Pacemaking is a fundamental aspect of IAS rhythmic contractions. More research is undeniably required to gain a comprehensive understanding of its organization and molecular identities.

28. p. 11. Lines 410-412. The authors report that the origin of pacemaker activity is not restricted to one edge of the IAS but that it is observed on both the inner and outer annular edges of the distal IAS as well as in the center of the muscle layer. The authors have failed to recognize that this behavior was also shown for calcium transients in Type II ICC-IM where phasic calcium activity was present in these cells across the circular muscle layer (Hannigan et al., 2020). Additionally, the authors report in the current study that they observed approximately three pacemakers per IAS slice. Perhaps they can discuss how this finding compares to the number of Type II ICC-IM present throughout the thickness of the distal IAS in the Hannigan et al., 2020 study. Also, if TMEM16A in SMCs is important for pacemaker activity in the IAS, does this mean that only a subpopulation of SMCs express TMEM16A in the distal IAS? It should be noted also that with indiscriminately loading the tissue with a calcium dye, ICC-IM will also be loaded. How are the authors able to say with certainty that they are imaging SMCs and not ICC-IM?

Response: We appreciate your comment. We have included a discussion (Lines 356-365) of the potential relevance of type II ICC-IM and their associated calcium signals as highlighted in our current study. Hannigan et al. (2020) are to be commended for using ICC GCaMP to study Ca^{2+} signals in the ICC. In our study, we used a calcium dye that could be loaded into all cells within the IAS slices.

Technically, the Ca^{2+} signals we detected could have come from any cell in the IAS, including the ICC-IM. However, as shown in Hannigan et al (Fig. 6Aa) and in our present study (Fig. 5), ICCs are sparsely distributed in the IAS. Cobine et al. (2011) estimated that ICCs occupy ~5% of the IAS volume, whereas the Ca^{2+} signals we observed propagated continuously across over a large area of the slice or even the entire slice. Moreover, the deletion of TMEM16A in the ICC did not affect any parameters of Ca^{2+} signals, arguing against these signals originating from the ICC-IM. In contrast, the deletion of TMEM16A in SMCs abolished these Ca^{2+} signals, suggesting that they originate from SMCs. Again, because the Ca^{2+} signals in Type II ICC-IM were studied under isometric conditions, they are likely to be different from the Ca^{2+} signals examined in our current study.

Given that (1) pacemaker sites occupy less than 1% of the IAS, and (2) TMEM16A immunostaining signals are distributed as the bands at the edges and the distal end of the IAS, we speculate that only a subset of TMEM16A+ SMCs serve as pacemaker cells. However, further quantification and investigation are needed to draw this conclusion.

Finally, we would like to emphasize that “indiscriminately loading the tissue with a calcium dye”, in fact, is an advantage for dissecting calcium signal coupling between different cell types when coupled with cell-specific genetic manipulation, as demonstrated in our study.

Reviewer #2 (Remarks to the Author):

This manuscript by Lu et al. addresses an important question regarding the origin of myogenic tone in the anal sphincter. The novel aspect of the work is to use precision cut cross sectional slices to analyse Ca²⁺ signals to study the location of pacemakers. Another, very important aspect of the work is to evaluate the role of TMEM16A in mediating pacemaker activity. The study uses a combination of TMEM16A blockers and cell-specific knock outs (KOs) of TMEM16A in Interstitial Cells of Cajal (ICC) and smooth muscle cells (SMC) to do this. Effects of both of these cell-specific KOs on contraction and calcium signals have been published before by the same group (Refs 20 & 22 of the manuscript), but the novelty in the present study was to examine how these KOs affected pacemaker activity.

Despite the importance of the questions and the potentially very interesting findings, I have some major concerns.

Response: We sincerely appreciate your recognition of the importance of our study. We also appreciate your recognition of our innovative approach to investigating the role of TMEM16A in mediating pacemaker activity. The experimental design involving TMEM16A blockers and cell-specific knockouts (KOs) of TMEM16A in Interstitial Cells of Cajal (ICC) and smooth muscle cells (SMC) was certainly a challenge that we felt it was essential to undertake. Below, we address your concerns point by point.

1. The conclusion that TMEM16A in SMC, but not in ICC, is responsible for pacemaker activity raises a controversy with another group (e.g. Ref 17) who have produced very convincing immunohistochemical evidence, and evidence of gene expression in sorted cells, that TMEM16A is highly expressed in ICC but not in SMC (Cobine, C. A. et al. *J Physiol* 595, 2021-2041, doi:10.1113/jp273618 (2017)). This discrepancy should be more fully acknowledged and discussed. Although Cobine et al. was referred to several times, the fact that they found that there was almost no expression of TMEM16A in SMC was not given due attention.

Response: We thank the reviewer for raising this important point about the discrepancy in TMEM16A expression between our study and that of Cobine et al. (2017). This difference certainly warrants careful attention. In response, we have added a paragraph addressing this issue (lines 348-356).

In our immunostaining of TMEM16A in the IAS (new Figure 5), as well as in our previous studies (Zhang et al. 2016; Lu et al. 2021), we observed a pattern similar to that of Cobine et al. (2017). However, we are puzzled by Cobine et al. (2017)'s display of the co-immunostaining of smMHC (a SMC marker) and TMEM16A in their Figure 10A. Notably, Figure 10A depicts a much smaller area of the IAS compared to the coimmunostaining of c-Kit and TMEM16A in their Figure 9A. When viewing Figure 10A online or on a computer screen, it is evident that large patches of TMEM16A-egfp signals clearly colocalize with smMHC from the 3 o'clock to 7 o'clock position on the edge of the IAS. It is possible that Cobine et al. (2017) might have overlooked the TMEM16A expression in SMCs for reasons unknown to us. Our results indicate that TMEM16A is expressed only in a subset of SMCs, primarily in the distal end and peripheral regions of the IAS tissues. As such, displaying the entire IAS section is essential to accurately depict the distribution of TMEM16A within the IAS.

Another possible explanation for the discrepancy may lie in the potential effects of the cell isolation and sorting process on gene expression. The process of cell isolation and the subsequent waiting period before sorting can alter gene expression profiles, a fact supported by many studies (e.g., van den Brink et al., 2017; Adam et al., 2017; Sheng et al., 2017). During isolation, cells are subjected to various stressors, such as enzymatic digestion, mechanical disruption, and temperature changes. These stressors can induce dynamic changes in gene expression. The time lag between cell isolation and sorting could further exacerbate these changes, especially if cells are maintained under suboptimal

conditions. In the case of the study by Cobine et al. (2017) it is plausible that the conditions during cell isolation and the subsequent waiting period before sorting could have affected the gene expression profiles of the isolated ICC and SMCs. The TMEM16A expression they detected in ICC but not in SMCs could have been influenced by these factors. It is also possible that the TMEM16A expression in SMCs is more susceptible to changes due to isolation stress than in the ICC. In addition, as TMEM16A is predominantly expressed in the extremities and peripheral regions of the IAS tissues. It is quite possible that the distal end of the IAS was omitted when samples were harvested for smooth muscle isolation and sorting. If the cell populations sampled by Cobine et al. (2017) were more centrally located, this could potentially explain the observed differences in TMEM16A expression.

Finally, mRNA levels do not always correlate with function. Strikingly, to the best of our knowledge, there is no direct evidence that TMEM16A in ICC is functional in the IAS. All the evidence from mRNA and protein expression, biophysical recordings or pharmacological manipulations is indirect or merely associative. Our cell-specific genetic deletion experiments show no essential role of TMEM16A in ICC in spontaneous rhythmic contractions and basal tone.

Taken together, immunostaining data presentation and the issue of gene expression changes due to the cell isolation process, time lag before sorting, and sample heterogeneity may lead to apparently contradictory results between different studies.

For your information, here are some references we used for this response.

van den Brink, S. C., et al. (2017). "Single-cell sequencing reveals dissociation-induced gene expression in tissue subpopulations." *Nat Methods* 14(10): 935-936. DOI: 10.1038/nmeth.4437.

This study demonstrates that the process of dissociating cells from tissues can induce changes in gene expression, particularly in certain subpopulations of cells.

Adam, M., Potter, A. S., & Potter, S. S. (2017). "Psychrophilic proteases dramatically reduce single-cell RNA-seq artifacts: a molecular atlas of kidney development." *Development*, 144(19), 3625–3632. DOI: 10.1242/dev.150433.

This paper shows that the choice of protease used to dissociate cells can impact the gene expression profiles obtained in single-cell RNA-seq experiments.

Sheng, K., Cao, W., Niu, Y., Deng, Q., & Zong, C. (2017). "Effective detection of variation in single-cell transcriptomes using MATQ-seq." *Nature methods*, 14(3), 267–270. DOI: 10.1038/nmeth.4145.

This paper presents a method to effectively detect gene expression variation in single-cell transcriptomes and discusses the potential impact of cell isolation and handling.

2. The manuscript would be greatly strengthened by convincing evidence that there is adequate expression of TMEM16A in SMC and that this really has been specifically deleted in the SMC-specific KOs. In the previous publication (Ref 22, Supp Fig. 10) the same group showed some co-localisation of TMEM16A and Myh11 in controls. This was confined to one region only and the same experiment was not performed in SMC-specific KOs. The present manuscript should rectify this.

Response: Following your suggestion, we have performed additional experiments to further demonstrate that 1) the TMEM16A proteins in SMCs indeed are restricted to the distal and peripheral IAS in the control IAS tissue, and 2) they are specifically deleted in the SMCs from TMEM16A^{SMKO} mice. We also performed the same experiment in TMEM16A^{ICCKO} mice and their controls, showing that TMEM16A is expressed and can be specifically deleted in the ICC. Finally, we provide a comparison of TMEM16A expression in different regions of the IAS. We have presented these new results in a new figure 5 in the revision.

3. Following on from the last point, the authors should quantify the presence of TMEM16A in SMC from control mice and in both SMC-specific and ICC-specific TMEM16A KOs. In a previous paper

(Ref 22, Supp Fig. 13) they attempted to do this for controls and SMC-specific KOs. However Cl⁻ currents were only found in 6 out of 29 cells in the control (and 1 of 29 in the KO). They attributed the low number of control cells expressing TMEM16A to 'run down' of the TMEM16A current in the whole cell patch clamp experiments they performed. These experiments should be repeated using perforated patch to prevent run down. Instead of clamping the [Ca²⁺]_i to 600 nM as they did before in whole cell mode, the TMEM16A currents can be evoked in perforated patch as a result of activation of L-type Ca²⁺ current. I realise that perforated patch experiments were performed previously to compare caffeine-evoked currents in controls and KOs, but the legend in Ref 22, Fig. 5 implies that cells that did not generate Cl⁻ currents were excluded from the data.

Response: Our previous study suggested that the IAS contains at least two different types of smooth muscle cells in terms of TMEM16A expression, which we have confirmed with new immunostaining data. The present study provides further evidence to support this notion. However, this feature complicates the channel recording of TMEM16A Cl⁻ currents in IAS SMCs. For example, the absence of Cl⁻ currents in TMEM16A^{-/-} SMCs could be due to at least three factors: 1) cells from the population that do not express TMEM16A, 2) cells from the population that do express TMEM16A but where it has been deleted, and 3) the experimental procedure itself inhibits recording, as patch-clamp recording is an invasive technique that could potentially damage the cells. As a result, quantifying TMEM16A function by patch-clamp, although a gold standard method one would consider for ion channels, is problematic and will yield ambiguous results in the IAS.

One possible way to address the issue is by generating TMEM16A^{SMKO} mice on a TMEM16A GFP+ genetic background, where GFP protein is expressed under the control of the endogenous *Tmem16a* regulatory elements. We have made persistent efforts to obtain TMEM16A-GFP knock-in mice, but to no avail. In collaboration with the investigator who developed this mouse line (PMID: 22988107), we attempted to generate these mice using the sperm cells provided by this investigator from TMEM16A-GFP knock-in mice. Unfortunately, due to the poor quality of the cells, our transgenic facility was unable to produce TMEM16A GFP+ breeders. Subsequently, we contacted an investigator at the University of Missouri about the mice (PMID: 30862712), only to learn that this group no longer maintained the line. We also reached out to a professor at the University of Nevada. Despite seeming receptive during each interaction, this professor never sent us the mice and did not provide a reason. Our efforts to secure these mice are ongoing, but this will likely require more time. While we value your suggestions and insights, conducting these experiments is extremely challenging at this time. The corresponding author is maintaining all communication records with these investigators and will be happy to share them if requested.

4. If the above suggested experiments confirm that TMEM16A is only expressed in a subset of SMC, the authors should consider the possibility that there is a specialised group of smooth muscle cells that act as pacemakers. This point is important and should be discussed.

Response: We appreciate the reviewer's insightful comment and agree that the possibility of a specialized subset of smooth muscle cells acting as pacemakers is indeed an interesting and important point to consider. To address your suggestion, we have expanded the Discussion (lines 342-347) to consider this possibility. We suggest that a TMEM16A+ SMC subset may be functionally distinct and could potentially be the source of pacemaker activity in the anal sphincter. We believe that these additions significantly enhance the depth of our manuscript and sincerely thank the reviewer for the valuable suggestions.

We would like to note that further studies, involving extensive characterization of the functional and molecular properties of this SMC subset, will be required to confirm this hypothesis. We are intrigued by this line of investigation and look forward to exploring this possibility in our future work.

5. Despite the fact that TMEM16 KO in SMC seemed to disrupt the activity, it is puzzling that two TMEM16A blockers had so little effect in Fig. 4. Niflumic acid is considered too non-specific to be a useful tool to study TMEM16A these days. It is interesting that the authors previously showed that another TMEM16A blocker reduced tone (Ref 22, supp Fig. 7). The potential reasons for the lack of effect of Ani9 and CaCCinh-A01 should be discussed. Also, Ani9 appears to be beginning to have an effect towards the end of its incubation period in Fig. 4C. The authors should consider applying it for longer.

Response. We appreciate your expert comments and insights. We have increased the number of experiments and observed marginal but statistically significant inhibition by Ani9 and CaCCinh-A01 on Ca²⁺ waves in the IAS slices (Figure 4). In our tests with T16Ainh-01, we observed marked inhibition of the tone under isometric conditions (Zhang et al. 2016). To our surprise, it marginally inhibited rhythmic contractions under isotonic conditions. We suspect that this difference arises from the variance in mechanical stretch between the two recording conditions. Mechanical stretch has been shown to impact TMEM16A activity, as evidenced by others (PMID: 22872152, 23104560, 30862712). This could, in turn, potentially alter the channel's response to inhibitors. More experiments are needed to resolve this intriguing observation. On the other hand, we would like to draw your attention to the fact that these data align with accumulating evidence casting doubt on the usefulness of the new generation TMEM16A inhibitors.

We advocate for the use of novel TMEM16A inhibitors and were among the first groups to use this class of inhibitors (Zhang et al., 2013). However, as the field has advanced and these inhibitors have been used by more investigators, the limitations of these compounds have become apparent and of concern. A key limitation relates to their effectiveness in revealing the function of TMEM16A in a given cell or tissue.

Indeed, there is no doubt that this new generation of TMEM16A inhibitors can inhibit TMEM16A currents, as demonstrated in the original studies of these compounds. However, the use of these inhibitors to study the function of these channels has yielded mixed results. For example, some studies have shown the efficacy of Ani9 in different cell types and conditions (PMID: 32272686, 36967079, 32821878, 35080921, 32039824). However, others have shown that it has little or no effect on specific functions, such as the ATP-induced Ca²⁺ increase in certain cell types or on intracellular calcium (PMID 32272686). In addition, Ani9 has been reported not to affect carbachol-induced contractions (PMID 36967079) and not to relax smooth muscle tissue (PMID: 32821878).

Regarding CaCCinh-A01, its efficacy also appears to be cell type and context dependent (PMID: 30165368, 28733893, 30562749, 31008669, 26013995, 36521670). While it reduced intracellular Cl⁻ concentration ([Cl⁻]_i) in some cell types, it only partially reduced Ca²⁺ waves in others (PMID 28733893) and failed to block pacemaker potentials in certain tissues (PMID: 30562749). These studies suggest that its mechanism of action may be more complex than previously thought, perhaps involving indirect effects mediated by alterations in intracellular Ca²⁺ handling (PMID: 36444690).

These observations lead to another major limitation of the new generation of TMEM16A inhibitors, namely, their numerous off-target effects (PMID: 28893247). Cobine et al. 2017 showed that CaCCinh-A01 at 10 μM reduced 30% of voltage-dependent Ca²⁺ currents in IAS SMCs. Other studies have shown that TMEM16A inhibitors have a number of off-target effects, including reducing intracellular Ca²⁺ elevation induced by stimuli acting on intracellular Ca²⁺ stores (PMID: 36444690, 26013995), relaxing rodent resistance arteries independent of the transmembrane chloride gradient (PMID: 26013995), blocking TMEM16F, a cation channel of the TMEM16 family (PMID: 30165368, 31008669), and inhibiting CFTR chloride channels in adult mouse trachea (PMID: 27063443), bestrophin-1 chloride channels stably expressed in CHO 9 (PMID: 25078708), and Ca²⁺-activated K⁺ channel (KCa_{3.1}) activity in human erythrocytes (PMID: 23430221).

Given these limitations, we have been cautious in drawing conclusions based on the effects of these inhibitors, a position also taken by Boedtkjer et al. (PMID: 26013995) and Dwivedi et al. (PMID: 36967079). Instead, we have emphasized a genetic approach in our study. As we show, a genetic approach allows us to dissect the function of TMEM16A in the IAS in a cell-specific manner, which is necessary since this channel is expressed in different cell types in this tissue.

In response to your comment, and in light of the recent advancements in TMEM16A inhibitors, we have added a paragraph discussing these points in Lines 366-376.

6. In Fig. 6B the SMC-specific KO seems to respond initially to 0 Ca²⁺ by contracting and with an elevated Ca²⁺. Was this also true in other cells or was it an artefact? There is no comment in the text, so it should be explained.

Response: We appreciate the reviewer's attention to detail regarding the response observed in SMC-specific KO in Figure 7B. Indeed, the initial contraction and elevated Ca²⁺ in response to 0 Ca²⁺ is a consistent observation in the SMKO tissues, and not an artifact.

Your observation leads to an intriguing point about the potential role of TMEM16A in IAS SMCs. We hypothesize that there may be a mechanism in the IAS whereby the removal of extracellular calcium induces a Ca²⁺ increase and contraction, and that this mechanism is somehow suppressed by the presence of TMEM16A in SMCs.

This is an area that we have yet to fully explore. Given its complexity, we were initially hesitant to comment on it, concerning it might introduce potential confusion to our current findings. However, based on your suggestion, we have added a note to the caption of Figure 7 that emphasize our current understanding and points out the intriguing questions this observation poses for future studies. Additionally, we have mentioned this observation as evidence that SMKO slices are functional in lines 252-253. We sincerely appreciate the reviewer for bringing attention to this interesting aspect.

Minor Points

1. Lines 446 & 447, "IAS-SMC express TMEM16A while TMEM16A^{-/-} SMCs do not, as determined by immunostaining.22" This is misleading, immunohistochemistry in Reference 22 was only performed on the control animals, not in the KOs (Supp Fig. 10).

Response. We thank the reviewer for the careful reading and we apologize for the confusion. The data from our previous work showing the absence of TMEM16A in SMCs in TMEM16A knockout mice was indeed inferred from Supplementary Figure 12, rather than directly shown.

Upon reflection, we agree that we should make this point more explicit in the current manuscript. To this end, we have now added new data clearly demonstrating that TMEM16A is expressed in IAS-SMCs from control mice but not in TMEM16A^{-/-} SMCs. We have revised the relevant lines to reflect these changes and to remove any ambiguity. We believe that these additional data provide a more accurate picture of the expression pattern of TMEM16A in our study.

2. The labels on Fig. 5F and 6F are confusing. From my reading of the Methods, the label should read "Lumen area increase (Maximal relaxation value in 1.3 mM Ca²⁺ =100).

Response: We appreciate your suggestion. The correct label is 'basal tone,' as indicated in the caption. The calculations for this are described in the methods (Lines 493-497). We have updated the labels in both figures accordingly. Please note that Fig. 5F and 6F in the original version are now represented as Fig. 6G and 6J, respectively, due to the introduction of a new Fig. 5 and rearrangement of the data description.

3. Line 153, please supply make and model of the camera.

Response: We have added this information in line 454.

4. Lines 205 & 206 “However, no experimental evidence to directly establish the relationship between $[Ca^{2+}]_i$ and contractions in the IAS.” This is not strictly true, as the authors showed this relationship themselves, albeit at single cell level in situ, rather than across the whole tissue (Ref 20, Fig. 5B).

Response: We appreciate your attention to detail and your reference to our previous work. You are correct; we have shown this relationship at the single cell level. In this sentence, our intent was to highlight the lack of evidence at the tissue level. We have now revised the sentence on lines 96-97 to: “To directly establish the relationship between $[Ca^{2+}]_i$ and contractions at the IAS tissue level, we imaged Ca^{2+} indicator Cal-520-loaded IAS slices at low magnification.”

5. Line 343 “(Figure 5G-5I)” should be “(Figure 6G-6I)”.

Response: Corrected as suggested.

6. Line 395, “which doubles” should read “is double”.

Response: Corrected as appropriate.

Reviewer #3 (Remarks to the Author):

In this study, Lu et al found the IAS contains several pacemakers that spontaneously generate propagating calcium waves to set the basal tone. These waves are myogenic, independent of nerve and paracrine or autocrine signals. They also identify TMEM16A Cl⁻ channels in SMCs, but not in ICCs, are required for pacemaking and basal tone. This study offers cellular and molecular insights into fecal continence and reveals a potential therapeutic target for fecal incontinence. The work is convincing and interesting. The results in general are well presented and the manuscript is well prepared. I suggest to accept it after a minor revision.

The figures could be more comforting by integral designed with better color scheme, arrangement and clear indication.

Response: We are grateful for your expert evaluation of our manuscript and for finding our work convincing and interesting. We also appreciate your suggestions for improvement. In response, we have revised our figures to include a more consistent color scheme (Figures 3, 4, 5, 6, 7, S4, S7), rearranged them for better coherence (add new panel in Figure 4, new Figure 5, Figures S3 and S7), and added more annotations for clarity (Figures 1, 2, 3, 6, 7, S2, S5).

Reviewers' comments:

Reviewer #1 (Remarks to the Author):

General comments:

While the authors have made extensive revisions to the manuscript, performed some additional experiments and provided lengthy comments back to the reviewers several issues remain to be addressed. Some of the comments/rebuttal and revisions do not adequately satisfy the reviewers comments. These are listed below.

Specific comments:

Introduction:

p.3. line 36. I believe the authors mean "historically" rather than "histologically"?

p.3. lines 59-61. The authors state that no Cl⁻ currents have been recorded from ICC in the IAS. While this is true, Cl⁻ currents have been recorded from ICC in other GI regions (PMID: 19703958, PMID: 25631870, PMID: 27742704) and in these studies as well as the 2017 study by Cobine et al. ANO1 protein and gene expression far exceeds that in other cell types including smooth muscle cells. It should be recognized that this is a difficult task to complete. Even in the 2016 study by the ZhuGe group, Cl⁻ currents were not able to be recorded from all smooth muscle cells. Additionally, since the ICC in the IAS occupy just 5% of the cell volume (PMID: 21337122; as referred to by the authors of the current study), this added to the complexity of these experiments. So perhaps it is due to the difficulty of the technique rather than ICC not having active ANO1 channels. Have the authors themselves tried to record currents from ICC? Perhaps these studies could be added to strengthen their argument that ICC do not exhibit Cl⁻ currents in either control or SM KO mice.

p.3. lines 67-68. Please note that the study by Hall et al. (PMID: 24951622) uses a dual microelectrode approach rather than use of just a single microelectrode to assess slow wave conduction and coordination in both the circumferential and oral directions. Therefore, these studies do provide spatial information further investigated by subsequent calcium imaging studies by the same group (PMID: 31625250, PMID: 32587396).

p.3. lines 75-76. It should be noted that the Cobine et al. study (PMID: 31625250) utilized a preparation whereby a cut in the circumferential direction was made to isolate the distal edge of the IAS (300um) from the more proximal aspect. They found that the frequency was greater in the distal segment of this preparation than in the same region of an intact tissue preparation. Therefore, the authors should be careful with stating that it mimics what occurs in vivo as per the Cobine study, activity is altered compared to what occurs in an intact tissue (recognizing that they too used a preparation that does not perfectly represent what occurs in vivo).

Results

p.5. lines 162-174. These ANO1 antagonist experiments should also be evaluated on contractile responses. Given the suggestion that ANO1 activity in SM is more relevant for contractile activity then one would anticipate no effect on SM ANO1 KO tissues and still seeing an effect in ICC ANO1 KO and control tissues. Therefore, this is an important experiment to better evaluate the KO models. It is also important to note that studies utilizing inducible ANO1 knockdown in ICC (same Cre mouse as used in the present study) have stated that there was only partial knockdown which coincided with a persistence in slow waves and calcium transients albeit somewhat disrupted (PMID: 27979828). A complete knockout of ANO1 in ICC was deemed to be required to completely eliminate these activities. This is not too dissimilar to studies using Kit mutant animals such as the W/W^v mouse. It has been demonstrated that some ICC populations are eliminated in these mice whereas others persist (PMID: 18079585). Additionally, it has been shown that ANO1 expression is present when Kit is reduced or eliminated (PMID: 25039457) suggesting that although these proteins are both expressed in ICC, their expression is not dependent on one another.

Methods and Materials:

p.10. lines 399-402. While you have provided the total number of mice, you have not provided breakdown into male versus female mice or commented as to whether or not sex differences were evaluated.

p.10. lines 405-413. The reviewer wants to clarify here. Are the authors saying that they removed the distal most 1mm and then cut that into two strips longitudinally, i.e., two strips per IAS? This is a very different preparation to the one used for calcium measurements. Why was the IAS cut into two and not left intact? Also, similar experiments are not included for the SM KO mice. It is important to repeat these experiments in both mice to better evaluate this question. As it stands, conclusions regarding the lack of ICC involvement are being drawn from a different approach than what has been used from SM mice. It is also important to include something to quantify tone with, e.g., as a percent of maximum contraction to KCl.

p. 11. Lines 448-450. While often experiments examining currents or calcium transients on isolated cells are performed at room temperature (such as the Sanders and Nelson studies referred to in the comments to reviewers) it should be emphasized, that temperature can significantly affect the conductance through calcium-activated chloride channels and voltage-dependent calcium channels (PMID: 12967945, PMID: 25366238, PMID: 11773241). Therefore, this and the fact that a different saline solution was used may contribute to the stark conclusions drawn between the authors and the Keef and Cobine group. Notably, the frequency of calcium oscillations was much greater in the Cobine study than in the current study, i.e., ~75-80 cpm when the distal most 300um was recorded from in a preparation where the distal IAS was separated from the remainder of the IAS as compared to ~60-65 cpm in the same region when the preparation was left intact as compared to ~30 cpm in the distal most 250um segment in the current study. Therefore, some of these calcium experiments should be repeated at a more physiological temperature just as the isometric tension recordings were done at 37oC to assess whether this can account for such a large discrepancy. It also important to note that stretch cannot entirely explain differences between the studies as Cobine et al. also addressed the question of stretch in their study. They performed length-tension experiments to determine optimal length and then evaluated changes in calcium transient amplitude and frequency with stretch. At the most distal part of the IAS, stretch did not significantly increase or decrease either parameter however, there was a rightward shift in the amplitude but not frequency in the proximal direction. Thus, the discrepancies noted here are more likely due to differences in the temperature than differences in stretch. Furthermore, if the authors are trying to establish whether the SM or the ICC are involved, then the same experiments need to be performed in both KO models, i.e., tension recordings at 37oC in SM KO tissues as well as in ICC KO tissues.

Figures:

Figure 5 is performed in preparations cut in the perpendicular direction to those rings used for calcium measurements yet the functional experiments were conducted in ring preparations, why was labeling not performed in the same preparation type? The results are interesting because as mentioned by Reviewer #2 an ANO1-eGFP mouse used in Cobine et al., 2017 (PMID: 28054347) showed that there was no GFP expression within the smooth muscle of the distal IAS. The authors should note that in the 2017 Cobine study, the orientation of the image in Fig 9 and 10 is different, the distal IAS being on the right in Fig.9 but on the bottom in Fig. 10. Though the images in Fig. 10 are higher in magnification, it still shows the distal most edge of the IAS and allows one to see the expression of ANO1 and smMHC and that ANO1 is not expressed in SMCs in this region. It should also be noted that often there is autofluorescence at a tissue edge where there is an abundance of connective tissue. This is especially true when a fixative such as acetone is used and may therefore account for the higher intensity of the ANO1 fluorescence at this edge (as well as the edge moving more proximal) than that seen in the middle of the tissue section. It is interesting that the authors had to use such a high concentration of Kit antibody (1:20) compared to the Cobine study where the same antibody was used at 1:1000 (though they used 4% paraformaldehyde as a fixative). While it is true that mRNA levels do not always correspond to protein expression, performing gene expression analysis would enhance the study by addressing this issue with another technique thus avoiding any issues with antibodies or autofluorescence (no negative control images have been included to verify specificity of the antibody labeling). Even if the methods utilized by Cobine et al. cannot be used, evaluation of gene expression in the whole of the distal tip of the IAS where the greatest fluorescence intensity is observed versus more

proximal where ICC ANO1 expression predominates should be performed. This is particularly important for the KO models.

Minor comment:

Please use a method other than "track changes" to show the changes made to a manuscript. All of the changes to formatting are unnecessary to show and it makes the document much less readable. It also makes the figures much harder to follow.

Reviewer #2 (Remarks to the Author):

1. The conclusion that TMEM16A in SMC, but not in ICC, is responsible for pacemaker activity raises a controversy with another group (e.g. Ref 17) who have produced very convincing immunohistochemical evidence, and evidence of gene expression in sorted cells, that TMEM16A is highly expressed in ICC but not in SMC (Cobine, C. A. et al. J Physiol 595, 2021-2041, doi:10.1113/jp273618 (2017). This discrepancy should be more fully acknowledged and discussed. Although Cobine et al. was referred to several times, the fact that they found that there was almost no expression of TMEM16A in SMC was not given due attention.

Thank you for inserting the paragraph beginning line 348. This at least admits that there is discrepancy between the present manuscript and the previous paper published by Cobine et. al. 2017.

2. The manuscript would be greatly strengthened by convincing evidence that there is adequate expression of TMEM16A in SMC and that this really has been specifically deleted in the SMC-specific KOs.

In the previous publication (Ref 22, Supp Fig. 10) the same group showed some co-localisation of TMEM16A and Myh11 in controls. This was confined to one region only and the same experiment was not performed in SMC-specific KOs. The present manuscript should rectify this.

This has been addressed by inclusion of new data in new Fig. 5.

3. Following on from the last point, the authors should quantify the presence of TMEM16A in SMC from control mice and in both SMC-specific and ICC-specific TMEM16A KOs. In a previous paper (Ref 22, Supp Fig. 13) they attempted to do this for controls and SMC-specific KOs. However Cl⁻ currents were only found in 6 out of 29 cells in the control (and 1 of 29 in the KO). They attributed the low number of control cells expressing TMEM16A to 'run down' of the TMEM16A current in the whole cell patch clamp experiments they performed. These experiments should be repeated using perforated patch to prevent run down. Instead of clamping the [Ca²⁺]_i to 600 nM as they did before in whole cell mode, the TMEM16A currents can be evoked in perforated patch as a result of activation of L-type Ca²⁺ current. I realise that perforated patch experiments were performed previously to compare caffeine-evoked currents in controls and KOs, but the legend in Ref 22, Fig. 5 implies that cells that did not generate Cl⁻ currents were excluded from the data.

I cannot agree with the following excerpt from the Authors' response:

"3) the experimental procedure itself inhibits recording, as patch-clamp recording is an invasive technique that could potentially damage the cells. As a result, quantifying TMEM16A function by patch-clamp, although a gold standard method one would consider for ion channels, is problematic and will yield ambiguous results in the IAS. "

However, I take the other two points and that it would mean sampling many cells to get a definitive answer to this question.

4. If the above suggested experiments confirm that TMEM16A is only expressed in a subset of SMC, the authors should consider the possibility that there is a specialised group of smooth muscle

cells that act as pacemakers. This point is important and should be discussed.

Thank you for the additional discussion, lines 342-347.

5. Despite the fact that TMEM16 KO in SMC seemed to disrupt the activity, it is puzzling that two TMEM16A blockers had so little effect in Fig. 4. Niflumic acid is considered too non-specific to be a useful tool to study TMEM16A these days. It is interesting that the authors previously showed that another TMEM16A blocker reduced tone (Ref 22, supp Fig. 7). The potential reasons for the lack of effect of Ani9 and CaCCinh-A01 should be discussed. Also, Ani9 appears to be beginning to have an effect towards the end of its incubation period in Fig. 4C. The authors should consider applying it for longer.

Despite the Authors' attempt to answer this question, and inclusion of additional discussion in lines 366-376, I do not think that this anomaly has been fully resolved. Firstly, all of the blockers in question, although prone to non-specific effects, are actually quite good blockers of TMEM16A, and Ani9 is particularly potent. Therefore, how could potentially non-specific effects prevent the drugs from blocking the responses?

Also, in their response, I think that the Authors have somewhat misrepresented the conclusions of Dwivedi et al. (PMID 36967079) with regard to Ani9 not inhibiting carbachol contractions. My understanding is that Dwivedi et. al. concluded that the lack of effect of Ani9 in their experiments indicated that TMEM16A was not involved in the carbachol contractures. Moreover, Dwivedi et al and Centeio et al (PMID 32272686) both concluded that Ani9 was relatively 'clean' compared to a range of other blockers.

However, the new paragraph beginning line 366, at least acknowledges the discrepancy with the previous work by the authors, namely that T16Ainh-A01 was effective. It is helpful that the authors have now included traces using T16Ainh-A01, which is the drug they used previously and so is comparable. As a suggestion, on revision the Authors should consider showing Figs 4Aa, Ba, Ca, Da, on an expanded vertical scale – why does the scale have to go down to -50? This does not make sense.

6. In Fig. 6B the SMC-specific KO seems to respond initially to 0 Ca²⁺ by contracting and with an elevated Ca²⁺. Was this also true in other cells or was it an artefact? There is no comment in the text, so it should be explained.

Thank you for your response.

Reviewers' comments:

Reviewer #1 (Remarks to the Author):

General comments:

While the authors have made extensive revisions to the manuscript, performed some additional experiments and provided lengthy comments back to the reviewers several issues remain to be addressed. Some of the comments/rebuttal and revisions do not adequately satisfy the reviewers comments. These are listed below.

Response: We appreciate the reviewer's time in reviewing our revision and rebuttal. We acknowledge that our study introduces a novel IAS preparation with new results and we understand that this requires a thorough review. Because of the new approaches used in this study, it is not correct to require that some parameters in this study be the same as those reported by Cobine et al. to which the reviewer consistently refers. As detailed in our manuscript, in our previous rebuttal, and below, there are fundamental differences in preparations, experimental conditions, data collection, and analysis between our study and the work of Cobine et al. We urge the reviewer to keep these differences in mind when evaluating our study.

We would like to emphasize that we meticulously detailed our procedures and performed all experiments in a controlled manner, including pre- and post-treatment comparisons and between wild-type and knockout models. It is under these controlled conditions that we have drawn our main conclusions: TMEM16A in SMCs is essential for calcium pacemaking, calcium waves and basal tone in IAS, whereas the presence of this channel in ICCs does not serve these functions. For the sake of objectivity, we also acknowledge the possibility that TMEM16A in ICCs may serve as-yet-undetermined functions in IAS (p. 9. lines 332-334). While no study is perfect, we believe it is clear that our study and its presentation would be considered solid by any reasonable standard. To the best of our knowledge, no study other than our own (Zhang et al.; 2016) has used two cell-specific knockout mice to elucidate the key function of the IAS - the generation of spontaneous basal tone.

Introduction:

p.3. line 36. I believe the authors mean “historically” rather than “histologically”?

Response: Thank you for pointing out this error. We have corrected the “typo” and replaced “histologically” with “historically”.

p.3. lines 59-61. The authors state that no Cl⁻ currents have been recorded from ICC in the IAS. While this is true, Cl⁻ currents have been recorded from ICC in other GI regions (PMID: 19703958, PMID: 25631870, PMID: 27742704) and in these studies as well as the 2017 study by Cobine et al. ANO1 protein and gene expression far exceeds that in other cell types including smooth muscle cells. It should be recognized that this is a difficult task to complete. Even in the 2016 study by the ZhuGe group, Cl⁻ currents were not able to be recorded from all smooth muscle cells. Additionally, since the ICC in the IAS occupy just 5% of the cell volume (PMID: 21337122; as referred to by the authors of the current study), this added to the complexity of these experiments. So perhaps it is due to the difficulty of the technique rather than ICC not having active ANO1 channels. Have the authors themselves tried to record currents from ICC? Perhaps these studies could be added to strengthen their argument that ICC do not exhibit Cl⁻ currents in either control or SM KO mice.

Response: Our current study provides a possible explanation for why Ano1 gene expression is higher in ICC than in smooth muscle cells in the study by Cobine (2017). The reason is that only a subpopulation of smooth muscle cells expresses ANO1. Given the numerous negative results regarding the functionality of ICC ANO1 in the IAS, we question the necessity of recording Cl⁻ currents in IAS ICC in our study. It might be more productive for those who support the importance of ICC in the IAS to perform these recordings. For example, the Cobine group, who used *Kit^{copGFP/+}* mice to sort ICC in their 2014 and 2017 studies, would be well positioned to attempt recordings from copGFP⁺ cells in the IAS. The lack of reported Cl⁻ currents from these cells in their work is puzzling, especially given their claim that ICC are the predominant cell type expressing ANO1 in the IAS.

If the reviewer is implying that studying the IAS is challenging, this only underscores the uniqueness and significance of our findings in the IAS. Our new data provide additional insight and explain why Cl⁻ currents were not able to be recorded from all smooth muscle cells: only a subset of SMCs express this channel. In addition, we have uncovered a novel role for some SMCs expressing TMEM16A in acting as pacemakers in the IAS. We believe that this discovery is both exciting and of significant value to the field.

p.3. lines 67-68. Please note that the study by Hall et al. (PMID: 24951622) uses a dual microelectrode approach rather than use of just a single microelectrode to assess slow wave conduction and coordination in both the circumferential and oral directions. Therefore, these studies do provide spatial information further investigated by subsequent calcium imaging studies by the same group (PMID: 31625250, PMID: 32587396).

Response: Your comment raises a question: if the dual microelectrode approach yields valuable spatial information, what prompted the same research group to further investigate using the calcium imaging approach? Probably because calcium imaging provides improved spatial resolution and other additional information. Following this logic, our imaging of IAS slices, especially when examined in circumferential orientations, should provide comprehensive insights into IAS calcium signaling and function. We believe that it is necessary for the scientific community to recognize diverse methodologies and findings. Our work contributes to the field, and we hope that it will be considered fairly and objectively, thereby fostering progress and a deeper understanding of IAS function.

p.3. lines 75-76. It should be noted that the Cobine et al. study (PMID: 31625250) utilized a preparation whereby a cut in the circumferential direction was made to isolate the distal edge of the IAS (300um) from the more proximal aspect. They found that the frequency was greater in the distal segment of this preparation than in the same region of an intact tissue preparation. Therefore, the authors should be careful with stating that it mimics what occurs in vivo as per the Cobine study, activity is altered compared to what occurs in an intact tissue (recognizing that they too used a preparation that does not perfectly represent what occurs in vivo).

Response: We appreciate your attention to the details of both our study and the Cobine et al. study. However, there are two key differences that need to be clarified and kept in mind. In the Cobine study, the IAS was cut, opened, and pinned with or without additional stretching, a preparation that alters its natural state regardless of the location of the cut. In contrast, our current study preserves the IAS as a ring, maintaining a condition more similar to its in vivo state. Moreover, Cobine et al. studied Ca²⁺ signals propagating in the longitudinal direction and did not provide circumferential information. Our

study focuses on the circumferential aspect, which is crucial because IAS tone results from smooth muscle contraction in this orientation. Therefore, our results provide more relevant insights for understanding the physiological functions of the IAS.

We believe that these distinctions are critical and support the validity and relevance of our findings. We trust that the scientific community will recognize the value of diverse methods and perspectives in advancing our understanding of the IAS function.

Results

p.5. lines 162-174. These ANO1 antagonist experiments should also be evaluated on contractile responses. Given the suggestion that ANO1 activity in SM is more relevant for contractile activity than one would anticipate no effect on SM ANO1 KO tissues and still seeing an effect in ICC ANO1 KO and control tissues. Therefore, this is an important experiment to better evaluate the KO models. It is also important to note that studies utilizing inducible ANO1 knockdown in ICC (same Cre mouse as used in the present study) have stated that there was only partial knockdown which coincided with a persistence in slow waves and calcium transients albeit somewhat disrupted (PMID: 27979828). A complete knockout of ANO1 in ICC was deemed to be required to completely eliminate these activities. This is not too dissimilar to studies using Kit mutant animals such as the W/W^v mouse. It has been demonstrated that some ICC populations are eliminated in these mice whereas others persist (PMID: 18079585). Additionally, it has been shown that ANO1 expression is present when Kit is reduced or eliminated (PMID: 25039457) suggesting that although these proteins are both expressed in ICC, their expression is not dependent on one another.

Response: We would like to draw the reviewer's attention to the growing concerns regarding the specificity and potential off-target effects of ANO1 antagonists, as demonstrated in many studies, including that Cobine et al. (PMID: 28054347), where CaCCinh-A01 at 10 μ M was shown to inhibit 30% of L-type Ca²⁺ currents in IAS SMCs. This ambiguity surrounding antagonists underscores the importance and reliability of our cell-specific ANO1 knockout mouse models in elucidating the role of ANO1 in the IAS.

We believe that our knockout models provide compelling evidence to support our conclusions. Performing additional experiments with ANO1 antagonists on contractile responses in these models is unlikely to provide additional insights that would strengthen our existing conclusions. Worse, the results of these proposed experiments with inhibitors are more difficult to interpret, as many studies have suggested. For example, Dwivedi et al (PMID: 36967079) conclude that "previous studies that have used these and similar compounds as tools to determine the role of TMEM16A in whole tissue experiments may have come to misleading conclusions and should be re-evaluated." Boedtkjer et al even titled their paper "New selective inhibitors of calcium-activated chloride channels-T16Ainh-A01, CaCCinh-A01 and MONNA- what do they inhibit?" (PMID: 26013995)

Regarding the characterization of our ICC ANO1 [TMEM16A] knockout mice, we reiterate that our results show approximately 95% deletion of TMEM16A in ICC, as shown in Figure 5D. Describing this as an "only partial knockdown" does not accurately represent our findings.

We do not understand why studies in the non-IAS parts of the GI tract (stomach, small intestine and colon in PMID 18079585, and colon in PMID 25039457) are used to criticize our data from the IAS, since the IAS has different physiological characteristics, and findings from these non-IAS parts may not be directly applicable. One reason we speculate is that Cobine et al. contradicted themselves regarding ICC

in the IAS. Cobine et al. reported that only ICC-SM are present in the IAS of W/W^V mice (PMID: 21337122) and suggested that these ICC-SM contribute to the comparable slow wave (SW) and tone of the IAS between W/W^V and wild type (PMID: 22074497). However, a subsequent study (PMID: 24951622) by Cobine et al. showed that although all three types of ICCs are present in the rectum, only ICC-IM (not ICC-SM or ICC-My) is present in the distal IAS where the highest frequency Ca²⁺ signals and slow waves were recorded. If ICC-IM is the only ICC population in the wild-type IAS, it is very surprising that they previously found ICC-SM present in the IAS of W/W^V mice. In fact, the consensus view is that ICC do not contribute to the spontaneous tone in the IAS, based on the studies in W/W^V mice (PMID: 16009682; 33452672; 22074497), including one from the Cobine and Keef group (PMID: 22074497).

We note that in light of the unexplained results of the slow wave, tone, and ICC characterizations in wild-type and W/W^V mice by Cobine et al, they suggested that "finally, SWs may be generated by SMC" (PMID 24951622). Our current study provides compelling genetic evidence to support this idea. We are puzzled as to why the reviewer is resistant to our new evidence.

Your comment regarding the relationship between ANO1 and Kit is not relevant to our data and conclusion.

Methods and Materials:

p.10. lines 399-402. While you have provided the total number of mice, you have not provided breakdown into male versus female mice or commented as to whether or not sex differences were evaluated.

Response: No sex difference was observed, so we pooled the results from both males and females as indicated on line 392.

Since the reviewer has consistently referred to the Cobine group's data to criticize our study, we feel it is appropriate to further discuss the issue of data reporting. Reporting the total number of mice is not a standard reporting method because statistical analyses are not performed on these numbers. Instead, the number of animals should be reported for each experiment in which statistical analyses are performed and results are presented. A review of the papers by Cobine et al reveals a lack of consistency in their reporting. For example, while they reported animal numbers in PMID 28054347 and 31625250 (both sexes were used), they failed to do so in PMID 21337122, 25301187, and 32587396. In particular, none of the papers cited by the Cobine group evaluated sex differences. It appears that the reviewer is applying a double standard in the evaluation, the reason for which remains unclear to us.

p.10. lines 405-413. The reviewer wants to clarify here. Are the authors saying that they removed the distal most 1mm and then cut that into two strips longitudinally, i.e., two strips per IAS? This is a very different preparation to the one used for calcium measurements. Why was the IAS cut into two and not left intact? Also, similar experiments are not included for the SM KO mice. It is important to repeat these experiments in both mice to better evaluate this question. As it stands, conclusions regarding the lack of ICC involvement are being drawn from a different approach than what has been used from SM mice. It is also important to include something to quantify tone with, e.g., as a percent of maximum contraction to KCl.

Response: Yes, for isometric experiments, the IAS was indeed split into two, resulting in two strips per IAS. This preparation differs from the slice used for calcium measurements. Please note in our study we measured contraction and calcium signals simultaneously under isotonic conditions. We did not use the isometric force measurement to explain the calcium signals obtained under isotonic conditions. Furthermore, it is important to note that the direction of contraction of the IAS SMC in both setups is aligned with the long axis of the SMC, which reflects the in vivo behavior. The use of two strips per IAS allows more efficient use of animal resources without compromising IAS behavior, as the contraction pattern of these strips is similar to that of Cobine et al. (2017), in which they used intact IAS after a cut for their experiments. Note that in Cobine et al. (PMID: 31625250), the IAS tissues were cut in half longitudinally for calcium transient measurements. So we are not the only ones using halves of the IAS for experiments.

The disrupted tone characterized by isometric measurement in SM KO mice was reported in our previous study (Zhang et al. 2016). There is no need to repeat it in this study.

As discussed in our previous rebuttal, in addition to isotonic measurement, we also performed isometric measurements in ICC KO mice to ensure that potential effects of TMEM16A in ICC KO mice were not missed. Our results were negative, which is consistent with isometric measurements of IAS strips in W/W^v mice (PMID: 22074497; 33452672; 16009682).

Therefore, our conclusions about TMEM16A in SMCs and ICC were derived from the same set of experiments, i.e., both isotonic and isometric conditions

Please note that in this study we focus on the spontaneous basal tone. We used extracellular calcium removal to reveal this tone in both wild-type and KO mice. Normalization of basal tone to KCl response would not provide additional insight into spontaneous tone generation. Nevertheless, we used KCl to verify that the absence of tone in SM KO mice was not due to a loss of contractile ability in the IAS. As shown in Figure S7, deletion of ANO1 in the SMC does not affect the contractile capacity of the IAS. Furthermore, the use of high KCl is not a standard way to study spontaneous IAS contractions, e.g., Cobine et al. did not use high KCl to stimulate IAS tissue in PMID 25301187.

p. 11. Lines 448-450. While often experiments examining currents or calcium transients on isolated cells are performed at room temperature (such as the Sanders and Nelson studies referred to in the comments to reviewers) it should be emphasized, that temperature can significantly affect the conductance through calcium-activated chloride channels and voltage-dependent calcium channels (PMID: 12967945, PMID: 25366238, PMID: 11773241). Therefore, this and the fact that a different saline solution was used may contribute to the stark conclusions drawn between the authors and the Keef and Cobine group. Notably, the frequency of calcium oscillations was much greater in the Cobine study than in the current study, i.e., ~75-80 cpm when the distal most 300um was recorded from in a preparation where the distal IAS was separated from the remainder of the IAS as compared to ~60-65 cpm in the same region when the preparation was left intact as compared to ~30 cpm in the distal most 250um segment in the current study. Therefore, some of these calcium experiments should be repeated at a more physiological temperature just as the isometric tension recordings were done at 37oC to assess whether this can account for such a large discrepancy. It also important to note that stretch cannot entirely explain differences between the studies as Cobine et al. also addressed the question of stretch in their study. They performed length-tension experiments to determine optimal length and then evaluated changes in calcium transient amplitude and frequency with stretch. At the most distal part of

the IAS, stretch did not significantly increase or decrease either parameter however, there was a rightward shift in the amplitude but not frequency in the proximal direction. Thus, the discrepancies noted here are more likely due to differences in the temperature than differences in stretch. Furthermore, if the authors are trying to establish whether the SM or the ICC are involved, then the same experiments need to be performed in both KO models, i.e., tension recordings at 37°C in SM KO tissues as well as in ICC KO tissues.

Response: We have previously performed Cl⁻ current recordings in IAS SMCs at room temperature in Zhang et al. (2016). Given this, we believe it is more consistent to perform both the Cl⁻ current recordings in IAS SMCs and the Ca²⁺ measurement in IAS slices at the same temperature. More importantly, in our previous rebuttal (9. p. 5. Lines 148-150), we thoroughly justified our decision to perform the slice experiments at 27°C. This choice was consistent with field conventions and took into account the differences between isometric recording and slice recording experiments.

The reviewer is concerned about the discrepancies between our results and those of Cobine et al., especially regarding the frequency of calcium oscillations. However, please note that the frequency in the two studies cannot be compared because there are fundamental differences in the methods and presentation of calcium signals between the two studies. Cobine et al. used a confocal microscope, whereas we used a wide-field microscope to acquire Ca²⁺ signals. Cobine et al. analyzed and presented Ca²⁺ signals from regions of interest (ROIs), whereas we measured and presented calcium signals from entire IAS slices. Because the size of the ROIs was not clearly defined in the method or in all 8 figures except Figure 6 in Cobine et al., one cannot know the size of the regions from which Ca²⁺ signals were derived. In Figure 6, the size of the ROI was reported to be 100 x 200 μm. Based on their Figure 1D, the smallest IAS sheet (i.e. the one cut 300 μm from the distal end) is 4000 x 300 μm. Thus, an ROI of 100 x 200 μm would be approximately 1.7% of the smallest IAS sheet they examined. The use of such a small ROI to measure Ca²⁺ oscillations is a major concern when comparing the results of Cobine et al. and ours because, as clearly shown in our study, calcium signals vary greatly in different areas of IAS slices. Furthermore, in the study by Cobine et al, the calcium transients propagated from the distal extremity to the proximal, i.e., oral direction, whereas in our study the calcium signals propagated circumferentially across the slices. One would not expect the calcium oscillations measured in completely different ways in our study and the study of Cobine et al. to be the same.

Assuming for the sake of argument that there are indeed differences in calcium oscillation frequency between our study and Cobine et al, it's premature to attribute these differences to temperature without direct experimentation. In our approach, IAS tissues were not subjected to external stretching, in contrast to the method of Cobine et al. This fundamental difference may play a role in the observed differences. Both our previous work (Lu et al., 2021, Figure S9) and Cobine et al (2017, Figure 1) highlighted significant differences in contraction between stretched and unstretched IAS strips.

It's important to emphasize that these differences in methodology and experimental conditions do not undermine our conclusions. At the frequency we measured, Ca²⁺ signals can generate the basal tone. Furthermore, our conclusions on TMEM16A's role in each cell type are derived from consistent parameters in both wild-type and KO IAS tissues.

Regarding the suggestion of "tension recordings at 37°C in both SM KO and ICC KO tissues," we reiterate that we have already performed these experiments in SM KO tissues, as documented in Zhang et al. (2016). Reproducing previously published results would be redundant. For this study, we performed

isometric measurements in ICC KO mice to ensure that we didn't miss any potential effects of TMEM16A on the IAS tone in these mice. We believe that it's important to document negative results as well.

Figures:

Figure 5 is performed in preparations cut in the perpendicular direction to those rings used for calcium measurements yet the functional experiments were conducted in ring preparations, why was labeling not performed in the same preparation type? The results are interesting because as mentioned by Reviewer #2 an ANO1-eGFP mouse used in Cobine et al., 2017 (PMID: 28054347) showed that there was no GFP expression within the smooth muscle of the distal IAS. The authors should note that in the 2017 Cobine study, the orientation of the image in Fig 9 and 10 is different, the distal IAS being on the right in Fig.9 but on the bottom in Fig. 10. Though the images in Fig. 10 are higher in magnification, it still shows the distal most edge of the IAS and allows one to see the expression of ANO1 and smMHC and that ANO1 is not expressed in SMCs in this region. It should also be noted that often there is autofluorescence at a tissue edge where there is an abundance of connective tissue. This is especially true when a fixative such as acetone is used and may therefore account for the higher intensity of the ANO1 fluorescence at this edge (as well as the edge moving more proximal) than that seen in the middle of the tissue section. It is interesting that the authors had to use such a high concentration of Kit antibody (1:20) compared to the Cobine study where the same antibody was used at 1:1000 (though they used 4% paraformaldehyde as a fixative). While it is true that mRNA levels do not always correspond to protein expression, performing gene expression analysis would enhance the study by addressing this issue with another technique thus avoiding any issues with antibodies or autofluorescence (no negative control images have been included to verify specificity of the antibody labeling). Even if the methods utilized by Cobine et al. cannot be used, evaluation of gene expression in the whole of the distal tip of the IAS where the greatest fluorescence intensity is observed versus more proximal where ICC ANO1 expression predominates should be performed. This is particularly important for the KO models.

Response: The orientation of our immunostaining was carefully chosen to provide a comprehensive and unbiased view of TMEM16A expression heterogeneity across the IAS. This result provides a good explanation for our functional results. If we performed the immunostaining in a circumferential orientation, we would have to show a series of images to reveal the heterogeneity in TMEM16A expression and distribution. Therefore, our choice is well justified.

We followed the manufacturer's guidelines for diluting the Kit antibody to 1:20 (https://www.rndsystems.com/products/human-cd117-c-kit-antibody_af332). We reconstituted the antibody at 200 µg/mL and diluted it to 10 µg/mL (1:20 dilution, the manual recommends dilution to 5-15 µg/mL). We have provided detailed information about our antibody use, whereas Cobine et al. did not provide such specifics. It is important to note that R&D Systems offers three different Kit antibodies (Catalog #: AF332, AF1356, MAB1356) suitable for the detection of mouse Kit by immunohistochemistry. Therefore, it is premature to assume that the same antibody was used in both our study and that of Cobine et al.

We performed appropriate negative controls as detailed in our methodology (page 13, lines 519-520; 523-525). While autofluorescence can indeed occur, it was not a problem in our study. This is evident from our knockout experiments: when TMEM16A was knocked out in SMCs, no fluorescent signal was detectable at the distal end and the edges of the IAS (as shown in Figures 5A and 5B). This observation clearly indicates that the fluorescent signals at the distal end and the edges in the wild-type mice (as

well as in the TMEM16A^{ICCKO} and its isogenic control) result directly from TMEM16A expression in SMCs and not from autofluorescence or some experimental error.

For your convenience, we have included Figures 9 and 10 from Cobine et al. (2017) below. In Figure 10A, the smooth muscle bundles are positioned at an approximate angle of 75 degrees. We are puzzled as to how the distal IAS ends up at the bottom in this orientation. It is also puzzling why Figures 9 and 10 have been cropped to different extents and presented at different spatial scales. Perhaps it would have been better to show the entire IAS tissue as in Figure 5 in our current study. Furthermore, Cobine et al. (2017) did not provide any information about the processing of these two images, such as the determination of background levels or negative controls. In the spirit of thorough scientific discussion, we have adjusted the intensity scaling of Figure 10A for the comparison below. As can be seen, fluorescent signals are present near the edge of the IAS and colocalize with smMHC. Curiously, the strongest signal in the SMCs is almost obscured by the ANO1-egfp label in the middle panel.

(Cobine et al. 2017; Figure 9)

(Cobine et al. 2017. Figure 10)

(Cobine et al. 2017. Figure 10A (Intensity rescaled for better visualization of ANO1-egfp signal))

Minor comment:

Please use a method other than "track changes" to show the changes made to a manuscript. All of the

changes to formatting are unnecessary to show and it makes the document much less readable. It also makes the figures much harder to follow.

Response: We apologize if the previous format made the document difficult to read and the figures difficult to follow. We intended to provide a clean version for evaluation and review, while the “track changes” feature was only meant to highlight the areas where changes were made.

Reviewer #2 (Remarks to the Author):

1. The conclusion that TMEM16A in SMC, but not in ICC, is responsible for pacemaker activity raises a controversy with another group (e.g. Ref 17) who have produced very convincing immunohistochemical evidence, and evidence of gene expression in sorted cells, that TMEM16A is highly expressed in ICC but not in SMC (Cobine, C. A. et al. J Physiol 595, 2021-2041, doi:10.1113/jp273618 (2017). This discrepancy should be more fully acknowledged and discussed. Although Cobine et al. was referred to several times, the fact that they found that there was almost no expression of TMEM16A in SMC was not given due attention.

Thank you for inserting the paragraph beginning line 348. This at least admits that there is discrepancy between the present manuscript and the previous paper published by Cobine et. al. 2017.

2. The manuscript would be greatly strengthened by convincing evidence that there is adequate expression of TMEM16A in SMC and that this really has been specifically deleted in the SMC-specific KOs.

In the previous publication (Ref 22, Supp Fig. 10) the same group showed some co-localisation of TMEM16A and Myh11 in controls. This was confined to one region only and the same experiment was not performed in SMC-specific KOs. The present manuscript should rectify this.

This has been addressed by inclusion of new data in new Fig. 5.

3. Following on from the last point, the authors should quantify the presence of TMEM16A in SMC from control mice and in both SMC-specific and ICC-specific TMEM16SA KOs. In a previous paper (Ref 22, Supp Fig. 13) they attempted to do this for controls and SMC-specific KOs. However Cl⁻ currents were only found in 6 out of 29 cells in the control (and 1 of 29 in the KO). They attributed the low number of control cells expressing TMEM16A to ‘run down’ of the TMEM16A current in the whole cell patch clamp experiments they performed. These experiments should be repeated using perforated patch to prevent run down. Instead of clamping the [Ca²⁺]_i to 600 nM as they did before in whole cell mode, the TMEM16A currents can be evoked in perforated patch as a result of activation of L-type Ca²⁺ current. I realise that perforated patch experiments were performed previously to compare caffeine-evoked currents in controls and KOs, but the legend in Ref 22, Fig. 5 implies that cells that did not generate Cl⁻ currents were excluded from the data.

I cannot agree with the following excerpt from the Authors’ response:

“3) the experimental procedure itself inhibits recording, as patch-clamp recording is an invasive technique that could potentially damage the cells. As a result, quantifying TMEM16A function by patch-

clamp, although a gold standard method one would consider for ion channels, is problematic and will yield ambiguous results in the IAS. "

However, I take the other two points and that it would mean sampling many cells to get a definitive answer to this question.

Response: In response to this comment, we would like to further clarify our approach and results. We understand your emphasis on quantifying the presence of TMEM16A in both SMC- and ICC-specific TMEM16A knockouts (KOs), as well as in control mice.

To address this concern, we have performed an extensive quantification of TMEM16A expression in both lines of TMEM16A KOs and their respective controls, as shown in Figure 5. Our analysis includes the entire IAS longitudinal section, and to account for the variation in Ca²⁺ signals, we have further divided the data into three evenly spaced segments. Our results clearly show that Kit^{CreERT2} mediated an approximate 95% reduction in TMEM16A expression in TMEM16A^{ICCKO} mice, and SMA^{Cre} led to an almost complete deletion of TMEM16A in TMEM16A^{SMKO} mice.

These findings have led to two critical and distinct results concerning intercellular Ca²⁺ waves and basal tone in the IAS: the deletion of TMEM16A in SMCs results in the elimination of intercellular Ca²⁺ waves and basal tone, whereas its deletion in the ICC has no significant effect on either parameter. We believe that these results provide robust and compelling evidence to elucidate the specific role of TMEM16A in the IAS. They underscore the critical role of TMEM16A in SMCs, while highlighting its apparent lack of functional involvement in ICCs, particularly with respect to intercellular Ca²⁺ waves and the maintenance of basal tone.

Furthermore, these results provide a more coherent interpretation of our previous findings, where "Cl⁻ currents were only found in 6 out of 29 cells in the control (and 1 of 29 in the KO)." This can now be understood to mean that TMEM16A is expressed in only a subset of SMCs in the control group.

Given that only a small percentage of IAS SMCs express TMEM16A, we believe that 'sampling many cells' in our current mouse models (i.e., without the TMEM16A-GFP marker) may not yield a definitive answer. Similarly, while recording Cl⁻ currents from dissociated single cells provides valuable data, it may not directly reveal how intercellular Ca²⁺ waves and resultant contractions are generated in IAS tissue slices, the primary focus of our study. Nevertheless, we recognize that this information has its own significance, especially in the context of the ongoing debate surrounding TMEM16A in the IAS. We are interested in further exploring these issues in future research using TMEM16A-GFP mouse lines for a more comprehensive investigation.

We hope that we have adequately addressed the reviewer's concerns and demonstrated the validity and strength of our experimental approach and results.

4. If the above suggested experiments confirm that TMEM16A is only expressed in a subset of SMC, the authors should consider the possibility that there is a specialised group of smooth muscle cells that act as pacemakers. This point is important and should be discussed.

Thank you for the additional discussion, lines 342-347.

5. Despite the fact that TMEM16 KO in SMC seemed to disrupt the activity, it is puzzling that two

TMEM16A blockers had so little effect in Fig. 4. Niflumic acid is considered too non-specific to be a useful tool to study TMEM16A these days. It is interesting that the authors previously showed that another TMEM16A blocker reduced tone (Ref 22, supp Fig. 7). The potential reasons for the lack of effect of Ani9 and CaCCinh-A01 should be discussed. Also, Ani9 appears to be beginning to have an effect towards the end of its incubation period in Fig. 4C. The authors should consider applying it for longer.

Despite the Authors' attempt to answer this question, and inclusion of additional discussion in lines 366-376, I do not think that this anomaly has been fully resolved. Firstly, all of the blockers in question, although prone to non-specific effects, are actually quite good blockers of TMEM16A, and Ani9 is particularly potent. Therefore, how could potentially non-specific effects prevent the drugs from blocking the responses?

Response: We understand the reviewer's concern regarding the apparent anomaly in the effects of TMEM16A blockers, and we acknowledge that our previous explanations may not have fully addressed this issue. Indeed, the TMEM16A blockers in question are well-documented as potent inhibitors. However, their efficacy in elucidating TMEM16A's function may be subject to modulation by a variety of factors, including potential non-specific effects. For example, the induction of calcium release from internal stores, as demonstrated in several studies including Dwivedi et al. (PMID 36967079), could theoretically negate the inhibitory influence of TMEM16A blockers. In addition, our current isotonic measurement, which differs from the isometric measurement used previously, may also contribute to the lack of efficacy of the blockers. Nevertheless, we would like to emphasize that we are not attributing the marginal effects of these inhibitors in the IAS solely to these non-specific effects. The complexity of this situation warrants a more thorough investigation to uncover the underlying mechanisms at play.

Also, in their response, I think that the Authors have somewhat misrepresented the conclusions of Dwivedi et al. (PMID 36967079) with regard to Ani9 not inhibiting carbachol contractions. My understanding is that Dwivedi et. al. concluded that the lack of effect of Ani9 in their experiments indicated that TMEM16A was not involved in the carbachol contractures. Moreover, Dwivedi et al and Centeio et al (PMID 32272686) both concluded that Ani9 was relatively 'clean' compared to a range of other blockers.

Response: We agree with your understanding of Dwivedi et al. and Centeio et al. However, we believe that the conclusion drawn by Dwivedi et al. suggesting that TMEM16A is not involved in carbachol-induced airway constriction may not be entirely accurate. It appears that Dwivedi et al. may have overlooked some crucial observations made by Wang et al. (a study to which Ping Lu and R ZhuGe contributed), which, as stated in their Introduction, served in part as the basis for the study by Dwivedi et al. Using the same TMEM16A KO mouse line as in the present study, Wang et al. found that deletion of TMEM16A abolished methacholine-induced Cl⁻ currents in airway smooth muscle cells (Fig. 1D). Although TMEM16A deletion does not alter Mch-induced contractions at high concentrations (>3 μ M), "when KO muscle was treated with MCh at a concentration as low as 300 nmol/L, the evoked force was significantly reduced (KO: 2.58 ± 0.68 mN vs. CTR: 5.66 ± 0.95 mN)" (Fig. 2D in Wang et al.). This result suggests that TMEM16A may be necessary for the contractile response to mild cholinergic stimulation.

The point here is not to argue which study is right or wrong, especially since their preparation is airway, not IAS. Instead, we would like to emphasize that results obtained from TMEM16A inhibitors are often

difficult to interpret. Both Dwivedi et al. and Centeio et al. advise caution in the use of these inhibitors. After discussing the specificity of TMEM16A inhibitors, Centeio et al. concluded, “This somewhat convoluted situation calls for attention when interpreting physiological effects of inhibitors and activators of TMEM16A.” Dwivedi et al concluded that “previous studies that have used these and similar compounds as tools to determine the role of TMEM16A in whole tissue experiments may have come to misleading conclusions and should be re-evaluated.” Boedtkjer et al even titled their paper “New selective inhibitors of calcium-activated chloride channels-T16Ainh-A01, CaCCinh-A01 and MONNA- what do they inhibit?” Our study echoes this sentiment. We understand that this is a complicated question and that each inhibitor has its off-targets, and each tissue or cell may respond differently to these inhibitors. For this reason, we discussed this issue in very general terms (lines 366-376) and cautioned the field to be cautious when using these inhibitors.

In addition to specificity and off-target issues, the pharmacological approach using these inhibitors is more problematic in tissues such as the IAS, where TMEM16A is expressed in multiple cell types and the inhibitors cannot differentially target TMEM16A in a given cell type. Therefore, we had to use a cell-specific TMEM16A deletion approach to delineate its role in the IAS. As a result, our primary conclusions are not dependent on the effects of these TMEM16A inhibitors but are essentially based on the findings from two cell-specific TMEM16A KO mice.

However, the new paragraph beginning line 366, at least acknowledges the discrepancy with the previous work by the authors, namely that T16Ainh-A01 was effective. It is helpful that the authors have now included a traces using T16Ainh-A01, which is the drug they used previously and so is comparable. As a suggestion, on revision the Authors should consider showing Figs 4Aa, Ba, Ca, Da, on an expanded vertical scale – why does the scale have to go down to -50? This does not make sense.

Response: The decrease in Ca²⁺ caused by niflumic acid in Figure 4A was approximately -25. To maintain consistency among the inhibitors, we uniformly set the scale to -50. We also reasoned that scaling to -50 does not alter the actual data but only changes the graphical presentation for aesthetic purposes. Rescaling for each inhibitor can easily be done to adjust this appearance. The rescaled Figure 4 is shown below for reference.

Fig. 4. Ca^{2+} waves from pacemakers may require Cl_{Ca} channels.

6. In Fig. 6B the SMC-specific KO seems to respond initially to 0 Ca^{2+} by contracting and with an elevated Ca^{2+} . Was this also true in other cells or was it an artefact? There is no comment in the text, so it should be explained.

Thank you for your response.

Reviewer #3 (Remarks to the Author):

The authors had addressed all my requests. The manuscript is much better after revision. I suggest accepting it.

This email has been sent through the Springer Nature Tracking System NY-610A-NPG&MTS

Reviewers' comments:

Reviewer #2 (Remarks to the Author):

I have no further comments.

Reviewer #4 (Remarks to the Author):

General Comments

This manuscript describes experiments investigating the molecular and cellular basis of pacemaker activity in the mouse internal anal sphincter (IAS) and its impact on maintenance of basal smooth muscle tone. The authors used cell-specific transgenic mice to determine the role of TMEM16A or Anoctamin-1 (ANO1) in evoking Ca²⁺ waves and smooth muscle tone in thin slice preparations of IAS. The authors report that ANO1 is expressed in both c-Kit-positive interstitial cells of Cajal (ICCs) and smooth muscle cells. Whereas ANO1-expressing ICCs displayed a punctate distribution throughout the IAS, smooth muscle cells exhibiting the highest expression of ANO1 were primarily localized in the distal portion of the IAS. In slices loaded with the Ca²⁺ indicator Cal-Bryte 520, the authors identified an average of three pacemaker sites per slice in the distal portion of the IAS that generated Ca²⁺ transients that propagated randomly and partially through the IAS tissue. These Ca²⁺ transients were hypothesized to be the basis of generating basal tone in the IAS. While knocking out ANO1 specifically in smooth muscle cells had a profound inhibitory effect on Ca²⁺ transients and basal tone, knocking out ANO1 in c-Kit positive ICCs had no effect. The authors conclude that ANO1 expressed in a subset of smooth muscle cells located distally in the IAS is required to generate pacemaker activity to support intercellular Ca²⁺ signaling, rhythmic contractions and fecal continence.

This is a well-written and well-designed study that provide convincing evidence that "in this" particular preparation, ANO1 in smooth muscle cells, but not in ICCs is responsible for supporting pacemaker activity. Knocking down gene expression in a cell-specific manner is the hallmark approach to determine the function of a protein in a tissue. The authors provided compelling evidence that knocking down ANO1 in ICCs or smooth muscle cells was specific, and yet only suppressing ANO1 expression produced significant effects on pacemaker activity and tone. These findings are diametrically opposed to those reported by Cobine et al. (Cobine et al. (Cobine, C.A., E.E. Hannah, M.H. Zhu, H.E. Lyle, J.R. Rock, K.M. Sanders, S.M. Ward, and K.D. Keef. 2017. ANO1 in intramuscular interstitial cells of Cajal plays a key role in the generation of slow waves and tone in the internal anal sphincter. *J Physiol.* 595:2021-2041) and Hannigan et al. (Hannigan, K.I., A.P. Bossey, H.J.L. Foulkes, B.T. Drumm, S.A. Baker, S.M. Ward, K.M. Sanders, K.D. Keef, and C.A. Cobine. 2020. A novel intramuscular Interstitial Cell of Cajal is a candidate for generating pacemaker activity in the mouse internal anal sphincter. *Sci Rep.* 10:10378). As highlighted extensively by Reviewers 1 and 2, the studies from the two groups were performed using two different preparations under very different experimental conditions. The authors of the current manuscript used primarily a thin slice preparation of IAS embedded in an agarose gel, whereas those of the Cobine group used intact IAS tissue strips. Whereas the tissue strip was studied under various levels of stretch, the thin slice preparation was not, a factor that could be important in determining the functional response of the preparation, in particular when examining the effects of ANO1 inhibitors on wild-type IAS. Another major difference also discussed at length by Reviewers 1 and 2 is the widely different temperature of the experiments used by the two groups, 37°C for the tissue strips and 27°C for the thin slice preparation. One argument in support of the main conclusion of the current study was the fact that IAS activity of mutant W/W^v c-Kit-deficient mice was similar to wild-type mice. Cobine et al. (2017) argued in their Discussion that W/W^v are not a quantitative Kit knockout as one allele is null, while the other one carries a mutation leading to a protein with reduced function. Despite this argument, de Lorijn et al. (de Lorijn, F., W.J. de Jonge, T. Wedel, J.M. Vanderwinden, M.A. Benninga, and G.E. Boeckxstaens. 2005. Interstitial cells of Cajal are involved in the afferent limb of the rectoanal inhibitory reflex. *Gut.* 54:1107-1113) showed that Kit-positive cells were absent in the same mutant mice and yet the basal IAS pressure by manometry was similar to wild-type mice.

One major concern with the current manuscript is the poor correlation between the genetic and pharmacological approaches to inhibit ANO1 activity. With the exception of the classical CaCC

inhibitor niflumic acid, all other inhibitors produced a very modest inhibition of Ca²⁺ transients despite being used at concentrations that exceeded their reported IC₅₀ by 1 to 2 log units. This contrasts with the studies of the Cobine group who showed that the newer generation of compounds dose-dependently inhibited IAS motility at concentrations expected to inhibit CaCC/ANO1 activity. The authors argue that perhaps the fact that slices, as opposed to strips, were not subjected to stretch might explain why the ANO1 inhibitors were less potent, this on the basis that ANO1 channels were shown in some studies to display some mechanosensitivity. This is a very weak argument. Patch clamp experiments performed to determine the pharmacological profile of expressed or native ANO1 showed potent activity by these compounds in cells whose membranes (or the entire cell) were not stretched (e.g., Bradley, E., S. Fedigan, T. Webb, M.A. Hollywood, K.D. Thornbury, N.G. McHale, and G.P. Sergeant. 2014. Pharmacological characterization of TMEM16A currents. *Channels (Austin)*. 8:308-320, and Seo, Y., H.K. Lee, J. Park, D.K. Jeon, S. Jo, M. Jo, and W. Namkung. 2016. Ani9, A Novel Potent Small-Molecule ANO1 Inhibitor with Negligible Effect on ANO2. *PLoS One*. 11:e0155771).

The authors also argue that pharmacological agents may exert off-target effects. Although this is always a concern when using small molecule inhibitors, usually their effects are more potent than expected, not less as reported here. This is in fact the case for the paper by Boedtkjer et al. (Boedtkjer, D.M., S. Kim, A.B. Jensen, V.M. Matchkov, and K.E. Andersson. 2015. New selective inhibitors of calcium-activated chloride channels- T16AInh-A01, CaCCInh-A01, and MONNA- What do they inhibit? *Br J Pharmacol*. 172:4158-4172), which the authors used as an argument to justify that CaCC inhibitors have off-target effects and their activity should be interpreted with caution. However, in contrast to this study, they found "profound" inhibition that they attributed to blocking other targets unrelated to CaCC/ANO1. Although there are a few studies suggesting problematic activity of using some of the ANO1 blockers, there are many more showing a profile consistent with the role of ANO1 in specific tissues, including some where an excellent correlation was drawn between pharmacological inhibition and genetic knockdown (Zawieja, S.D., J.A. Castorena, P. Gui, M. Li, S.A. Bulley, J.H. Jaggar, J.R. Rock, and M.J. Davis. 2019. Ano1 mediates pressure-sensitive contraction frequency changes in mouse lymphatic collecting vessels. *J Gen Physiol*. 151:532-554; Papp, R., C. Nagaraj, D. Zabini, B.M. Nagy, M. Lengyel, D. Skofic Maurer, N. Sharma, B. Egemnazarov, G. Kovacs, G. Kwapiszewska, L.M. Marsh, A. Hrzenjak, G. Hofler, M. Didiysova, M. Wygrecka, L.K. Sievers, P. Szucs, P. Enyedi, B. Ghanim, W. Klepetko, H. Olschewski, and A. Olschewski. 2019. Targeting TMEM16A to reverse vasoconstriction and remodelling in idiopathic pulmonary arterial hypertension. *Eur Respir J*. 53; Akin, E.J., J. Aoun, C. Jimenez, K. Mayne, J. Baeck, M.D. Young, B. Sullivan, K.M. Sanders, S.M. Ward, S. Bulley, J.H. Jaggar, S. Earley, I.A. Greenwood, and N. Leblanc. 2023. ANO1, CaV1.2, and IP3R form a localized unit of EC-coupling in mouse pulmonary arterial smooth muscle. *J Gen Physiol*. 155). The authors should be less unequivocal in the Discussion about only siding their interpretation with studies proposing that ANO1 inhibitors are inefficient and/or non-specific.

Taken together, it is clear that the conflicting results and conclusions from these studies cannot be easily reconciliated. Either one group is right and the other is wrong, or both groups are partially right. Although attempting to manipulate stretch in the thin slice preparation would not be a trivial task (perhaps by fabricating an expandable matrix supporting the slice), performing experiments at 37°C, which is the main differing variable, should not be. Reviewer 1 also made this point but surprisingly, the authors were reluctant to perform these experiments, arguing that they had done this in one of their previous papers (Zhang, C.H., P. Wang, D.H. Liu, C.P. Chen, W. Zhao, X. Chen, C. Chen, W.Q. He, Y.N. Qiao, T. Tao, J. Sun, Y.J. Peng, P. Lu, K. Zheng, S.M. Craige, L.M. Lifshitz, J.F. Keaney, Jr., K.E. Fogarty, R. ZhuGe, and M.S. Zhu. 2016. The molecular basis of the genesis of basal tone in internal anal sphincter. *Nat Commun*. 7:11358), and this would be redundant. Actually, this is not an unreasonable request. The study by Zhang et al. was done in IAS strips only, not in slices. In view of the conflicting results obtained with the two types of preparations, it is only reasonable to at least eliminate the temperature variable in an attempt to reconcile these apparent contradictory results. Regardless of the results obtained at 37°C, the authors should present a more objective overview of the limitations of their study in the Discussion, acknowledging that it is possible that the creation of the thin slice preparation may have changed the physiology of this tissue and that one cannot rule out that gene deletion might have led to compensatory changes in gene expression that may have contributed to the results obtained.

Response to reviewer 4:

Thank you for taking the time and effort to review our manuscript. We sincerely appreciate your diligence in conducting a comprehensive assessment of the literature and providing thoughtful and well-balanced comments. We have taken your comments to heart and revised the manuscript accordingly.

We acknowledge your suggestion to conduct experiments at 37°C to eliminate the variable of temperature. However, we wish to clarify that experiments at this temperature in the IAS slices cannot resolve the discrepancy between the results from Cobine et al. and our group because in addition to differences in temperature and stretch, there are at least three other distinctions.

First, in our manuscript, we emphasize that Cobine et al. used wortmannin, a compound that inhibits muscle contraction for stable electrical recording, while we did not employ this method in our study. Additionally, Cobine et al. used various drugs in different studies. For instance, in PMID 32587396, atropine was utilized; in PMID 24951622, both guanethidine (1 μ M) and atropine (1 μ M) were used; and in PMID 28054347, a combination of atropine (1 μ M), guanethidine (1 μ M), N ω -nitro-L-arginine (L-NNA) (100 μ M), and MRS2500 (1 μ M) was employed. These details were not included in our previous discussion for two reasons: (1) due to space limitations, and (2) the ambiguity in Cobine et al.'s papers regarding the presence of these drugs in their slow wave recording experiments. For example, in PMID 24951622 under 'Tissue Preparation' in the methods section, it is stated that 'Guanethidine (1 μ M) and atropine (1 μ M) were included in all bathing solutions to eliminate the influence of adrenergic and cholinergic neural inputs respectively,' yet under 'Measurement of Electrical Activity,' the method only mentions 'To maintain impalements, tissues were initially bathed for 20 min in 20 μ M wortmannin (a MLCK inhibitor) followed by a 45-minute washout period in regular KRBS before beginning recordings.'

Second, we have discussed another factor in our manuscript: our use of chemical calcium indicators, which were not employed by Cobine et al. in measuring slow wave frequency.

Third, as we addressed in our previous response to Reviewer 1, in isometric recordings such as those in Cobine et al. and in our own work (Fig. 6 in current study and Fig. 2 in Zhang et al., 2016), the IAS strips are continuously supplied with 95% O₂/5% CO₂, and the KPS medium is replaced as needed. In contrast, the IAS slice study was conducted under constant perfusion with the sHBSS solution exposed to atmospheric air (21% O₂/0.04% CO₂).

To further elaborate, we have attached an enlarged original picture of the recording chamber with an IAS slice, as published in Lu et al., 2021. In our IAS slice experiments, the chamber is sealed at the long end with silicone grease and semi-sealed with a Kimwipe tissue (white rectangular object near the red arrow) at the short end to allow solution perfusion. The black and red arrows indicate the locations of inflow and outflow, respectively. We constantly perfused the IAS-mounted chamber with sHBSS (the ingredients of which are detailed in our manuscript) exposed to atmospheric air containing 21% O₂/0.04% CO₂.

We used this recording condition as the chamber and method are well-established and widely accepted in the scientific community. They have been rigorously validated in functional studies in a variety of tissue types, including but not limited to lung, pancreas, heart, and kidney. We learned this technique from the laboratory of our late colleague, Mike Sanderson, who pioneered lung slices (i.e., PMID: 11815668, and 12388370).

Therefore, as you recognized, the differences between our study and Cobine et al. are complicated. As described above, there are at least five major variable differences between the two groups: stretch, temperature, treatment with wortmannin and other drugs, chemical calcium indicators, and O_2/CO_2 .

We recognize the need to understand the reasons for different results observed between Cobine et al and our group. To facilitate this, we have initiated a collaboration with a biomedical engineering group at the University of Connecticut to develop a new tissue slice recording chamber. This new chamber will feature an open design and will be equipped with essential components, including mounting apparatus and a force transducer. These enhancements will enable us to conduct a more comprehensive investigation of the IAS slice. We plan to explore the influence of various factors, such as stretch,

temperature, O₂/CO₂ concentration, as well as other variables like drug treatments and calcium indicators in the future.

In response to your comments, we have revised our manuscript to provide a more objective overview of the study's limitations in the Discussion section. We thank you for directing our attention to three pivotal studies demonstrating “an excellent correlation between pharmacological inhibition and genetic knockdown,” which we have now referenced in our discussion (page 10, line 371). We acknowledge that the stretch may not be a strong explanation for the inconsistency in inhibitor results between the two conditions and have therefore removed it. Instead, we acknowledge that the underlying reasons remain unknown (page 10, lines 377-380). Additionally, we have recognized that the creation of the thin slice preparation might have altered tissue physiology (page 8, lines 292-294). Moreover, we considered the possibility of compensatory changes in gene expression that could have influenced our findings (page 10, lines 382-383). Finally, we included a detailed description of the O₂/CO₂ concentration in our methods (page 11, lines 455-456) and discussed its role as a significant compounding factor, potentially explaining some of the discrepancies between our study and previous research (page 8, lines 287-290; page 10, lines 377-380).

Finally, we would like to stress that the apparent differences between the two groups do not undermine the conclusions we obtained in this study. We appreciate your recognition that a strength of our study is “knocking down gene expression in a cell-specific manner,” which “is the hallmark approach to determine the function of a protein in a tissue.” Indeed, performing experiments in a precisely controlled manner—KO vs. their isogenic control—allowed us to reach all the main conclusions in this study. However, we acknowledge that no experiment is perfect. We intend to optimize our method to further gain insight into the pathophysiology of the IAS, a clinically important area, yet with limited studies performed by a small number of basic science groups worldwide.

We hope that our revision has addressed your concerns and improved the manuscript within the current experimental constraints. Once again, we appreciate your time and valuable feedback.

REVIEWERS' COMMENTS:

Reviewer #4 (Remarks to the Author):

Thank you for your thoughtful response.

Reviewer Comments: Thank you for your thoughtful response.

Response: We are pleased that Reviewer 4 is satisfied with our response.